Resource

# A topographic atlas defines developmental origins of cell heterogeneity in the human embryonic lung

Alexandros Sountoulidis [1,2,12], Sergio Marco Salas[1,3,12], Emelie Braun[4], Christophe Avenel[5,6], Joseph Bergenstråhle[7], Jonas Theelke [1,2], Marco Vicari [7], Paulo Czarnewski [7], Andreas Liontos[1,2], Xesus Abalo [7], Žaneta Andrusivová[7], Reza Mirzazadeh[7], Michaela Asp[7], Xiaofei Li[8], Lijuan Hu[4], Sanem Sariyar[9], Anna Martinez Casals[9], Burcu Ayoglu[9], Alexandra Firsova [1,2], Jakob Michaëlsson[10], Emma Lundberg [9], Carolina Wählby[5,6], Erik Sundström[8], Sten Linnarsson [4], Joakim Lundeberg [7], Mats Nilsson [1,3] ✉ & Christos Samakovlis [1,2,11] ✉

The lung contains numerous specialized cell types with distinct roles in tissue function and integrity. To clarify the origins and mechanisms generating cell heterogeneity, we created a comprehensive topographic atlas of early human lung development. Here we report 83 cell states and several spatially resolved developmental trajectories and predict cell interactions within defined tissue niches. We integrated single-cell RNA sequencing and spatially resolved transcriptomics into a web-based, open platform for interactive exploration. We show distinct gene expression programmes, accompanying sequential events of cell differentiation and maturation of the secretory and neuroendocrine cell types in proximal epithelium. We define the origin of airway fibroblasts associated with airway smooth muscle in bronchovascular bundles and describe a trajectory of Schwann cell progenitors to intrinsic parasympathetic neurons controlling bronchoconstriction. Our atlas provides a rich resource for further research and a reference for defining deviations from homeostatic and repair mechanisms leading to pulmonary diseases.

The traditional account of cellular heterogeneity in the lung based on meticulous histology and expression of few characteristic markers suggests more than 40 cell types in the adult human lung[1]. The lung cell-type repertoire has been further expanded by recent developments in single-cell genomics allowing the interrogation of hundreds of thousand cells from adult healthy and diseased human lungs[2–5]. So far, 58 distinct cell types and states can be categorized into the five major cell classes of epithelial, stromal, immune endothelial and neuronal cells.

Our knowledge of human lung development derives largely from animal models and simplified organoid cultures[6,7] underscoring the lack of systematic studies of intact embryonic tissues. In this Resource, we focused on the first trimester of gestation and applied state-of-the-art technologies to capture and map the gene expression profiles of human embryonic lung in time and space. We first defined six main cell categories: mesenchymal, epithelial, endothelial, neuronal and immune cells, and erythroblasts/erythrocytes. Higher-resolution analysis of each of these categories suggested 83 cell identities, corresponding to cell types and transitional states. Next, we defined topological neighbourhoods of spatially related cell identities and used interactome analyses to describe communication niches and

tissue-design rules driven by spatial factors and cell interactions. We present an online platform integrating single-cell RNA sequencing (scRNA-seq) with the spatial analyses to facilitate interactive exploration of our data on whole lung tissue sections at different ages.

## Results

### Overview of cell heterogeneity in the embryonic lung

We dissected lungs from 17 embryos, ranging from 5 to 14 weeks post conception (PCW) at approximately weekly intervals (Supplementary Table 1 (1) and Extended Data Fig. 1a–c). Assuming that the two lungs are bilaterally symmetric, we regularly used the right lobes for scRNA-seq and processed the left lobes for spatial analyses. For in situ mapping, we aimed to analyse consecutive sections of the same tissues to independently validate the cell-state topologies. A first clustering and differential expression analysis of 163,236, high-quality complementary DNA libraries (Extended Data Fig. 1d–h) revealed six main cell categories: the mesoderm-derived (1) mesenchymal, (2) endothelial, (3) immune cells and (4) erythroblasts/erythrocytes, as well as (5) the ectoderm-derived neuronal and (6) the endoderm-derived epithelial cells (Extended Data Fig. 2a–g and Supplementary Table 1 (3) and (13)). Next, we dived deeper into each of them by re-clustering the corresponding cells, to expose additional cell states that were hidden in the whole dataset analysis. This revealed an unexpectedly high heterogeneity of 83 distinct cell states (Fig. 1a and Extended Data Fig. 3a).

To further explore the proposed cell-states and map them back to the tissue, we monitored gene expression patterns on tissue sections with spatial transcriptomics (ST) in nine different stages (the interactive viewer[8] contains representative sections of 6, 8.5, 10 and 11.5 PCW lungs). Probabilistic analysis of the ST data[9] largely validated the scRNA-seq results and spatially mapped the suggested clusters (example in Fig. 1b). The probability estimation of each cluster in every ST spot allowed definition of possible cluster pairs, located consistently in the same 'niche' (55-µm-diameter ST spot). We defined four distinct cell neighbourhoods, in characteristic anatomical positions, including proximal and distal airway compartments, vessels and parenchyma (Fig. 1c and Methods). To explore the communication code among cell states in each neighbourhood, we used interactome analyses with CellChat[10] and Nichenet[11] (interactive viewer and example in Fig. 1d).

To achieve higher resolution, we targeted 177 cell-state markers and selected NOTCH, HH, WNT and RTK/FGF signalling components to validate cell communication events by multiplex HybISS[12,13] (Fig. 1e and Extended Data Fig. 2h) and SCRINSHOT[14]. To facilitate accessibility and easy data exploration, we constructed an interactive viewer combining all modules of our analyses (https://hdca-sweden.scilifelab.se/tissues-overview/lung/). Below, we present the analyses of mesenchymal, epithelial and neuronal cell states and their interactions. Immune and endothelial cells are described in Supplementary Note 1.

### Distinct positions of mesenchymal cell states

The largest cluster in our dataset consisted of mesenchymal cells (Extended Data Fig. 2a). Subclustering revealed six distinct cell types expressing specific markers for known fibroblast, mesothelial, chondroblast and smooth muscle cell types and several immature states, characterized by the general mesenchymal markers *COL1A2* (ref. 2) and *TBX4* (ref. 15) and the lack of specific cell-type markers (Fig. 2a, Extended Data Fig. 4a and Supplementary Table 1 (4)). Annotation was also based on the spatial mapping of clusters at different timepoints (Fig. 2b and Extended Data Fig. 4b), the relative cluster positioning in the uniform manifold approximation and projection (UMAP) plot[16], partition-based graph abstraction (PAGA plot)[17] (Fig. 2a) and scVelo[18] analyses (Extended Data Fig. 4c) positioning immature cell states in the UMAP-plot centre and the more mature ones at the periphery. We spatially detected: (1) mesothelial cells (cluster (cl)-19), expressing *WT1*, *MSLN*, *KRT18* and *KRT19* at the tissue margins (Extended Data Fig. 4d), (2) pericytes/vascular smooth muscle (cl-14) associated with endothelium (Fig. 1c) and marked by *PDGFRB* and moderate levels of *ACTA2* and *TAGLN*, (3) *SOX9*[pos] *COL2A1*[pos] chondroblasts (cl-18) surrounding proximal airways, (4) *MYH11*[pos] *DACH2*[pos] airway smooth muscle (ASM, cl-13) close to airway epithelium, (5) *SERPINF1*[pos] *SRFP2*[pos] adventitial fibroblasts (AdvFs, cl-10) and (6) *ASPN*[pos] *TNC*[pos] airway fibroblasts (AFs, cl-16). AdvF and AF occupied distinct positions in the bronchovascular bundles[19], with the AFs being localized closer to airways than AdvF (Fig. 2b (5), (6)). Immature cell states (cl-0, cl-2 and cl-6) showed scattered distribution (Extended Data Fig. 4b). Lastly, 5 of the 21 mesenchymal clusters contained proliferating cells, which were widely distributed at early stages and became more localized around distal airways over time (Fig. 2a and Extended Data Fig. 4e).

### ASM maturation states coincide with distinct topologies

A prominent PAGA-plot trajectory suggested a differentiation path of immature mesenchyme towards ASM. It connected three immature clusters (cl-0, cl-2 and cl-6) to a proliferating ASM cluster (cl-20) and three ASM clusters (cl-8, cl-12 and cl-13) (Fig. 2a). This proposed that the trajectory stems from the immature mesenchyme connects to the immature ASM cl-8 and cl-12, leading to the more mature ASM cl-13 (Fig. 2c,d and Extended Data Fig. 4f). Proliferating ASM cells showed high expression of smooth muscle markers, such as *ACTA2* and *TAGLN*, implying that they represent a more mature state than cl-0 (Extended Data Fig. 4a). Interestingly, cl-20 also selectively expressed genes encoding extracellular matrix (ECM) proteins (Extended Data Fig. 4g), suggesting that proliferating ASM progenitors are transcriptionally distinct and locally contribute to ECM composition. Using pseudotime analysis[20,21], we defined differentially expressed gene-modules that might contribute to differentiation along the ASM trajectory (Extended Data Fig. 5a). Characteristic regulators

**Fig. 1 | Overview of the study. a,** UMAP plot of the 83 identified cell clusters by the analyses of the main cell categories (mesenchyme, epithelium, endothelium, immune and neuronal cells) from all 17 analysed donors. The two insets (dotted lines) at the right side of the plot correspond to clusters of doublets (top) and epithelial ciliated cells (bottom), which have been re-arranged in the original UMAP plot. Their initial locations are shown in Extended Data Fig. 2a. imm, immature; endo, endothelial; macroph, macrophage; fibro, fibroblast; prol, proliferating; mesench, mesenchymal; ASM, airway smooth muscle; prog, progenitor; SCP, schwann precursor cell; megakaryo, megakaryocyte; epith, epithelial. **b,** Example of an analysed 6 PCW lung section with ST, showing the cluster positional predictions for 75 out of the 83 identified cell clusters, as pie charts, according to stereoscope analysis. The missing eight clusters correspond to the cell states in parasympathetic ganglia, which were detected as one neuronal cell state. Insert: magnification of an ST spot, showing its cluster composition. epi, epithelial; prox, proximal; pcw, post conception week. **c,** Co-localization graph based on cluster co-occurrence in ST spots, according to stereoscope. Neuronal clusters are grouped in a single group (neuronal), and immune cell types are excluded. Lines indicate the strongest connections (Pearson's *r* > 0.04) between two clusters in the 55-µm-diameter ST spots. Distal and proximal airways, vessels and parenchyma are the four identified 'cell neighbourhoods'. Colours as in **a.** epi, epithelial; mes, mesenchymal; endo, endothelial; erythro, erythrocytes. **d,** Cartoon of predicted WNT-signalling communication patterns between spatially related clusters, showing its effect on target cells, based on previous knowledge. Interactome analyses with (1) CellChat[10], based on expression of ligands, receptors and co-factors and (2) Nichenet[11], which that predicts target-gene activation in response to cell communications. Clusters represented by each drawn cell are indicated in **a. e,** Experimental validation of *WNT7B* communication pattern, between WNT7B[pos] epithelium and the surrounding mesenchyme, using HybISS (individual-gene images in Extended Data Fig. 2h). Interactive visualization of (1) scRNA-seq analyses with (2) cell-type distributions on whole sections, (3) spatial gene expression patterns (experimentally detected and imputed) and (4) cellular interactions, focusing on distinct tissue neighbourhoods is available in https://hdca-sweden.scilifelab.se/tissues-overview/lung/.

include the myogenic transcription factor (TF) DACH2 (ref. 22), which was detected mainly in intermediate states (cl-8 and c-12) (Extended Data Fig. 5a,b, module 5). LEF1 was expressed in cl-8 but not earlier, in agreement with the published role of WNT signalling in smooth muscle development[23,24] and SSRP1, a FACT complex component, which modifies the chromatin structure at the promoters of muscle-specific genes, activating them[25] (Extended Data Fig. 5b). The expression of the NOTCH ligand *JAG1* was also increased in cl-6 and cl-8, in agreement with previous in vitro analysis[26] (Extended Data Fig. 5c). Differentiation

into mature ASM states seems to occur in cl-12 and cl-13 and is illustrated by increased expression of *ACTA2*, *TAGLN* and *MYH11* (ref. 2) (Extended Data Fig. 5a, module 7). *NR4A1*, a negative regulator of vascular smooth muscle[27] proliferation, was among the most highly upregulated TFs in the mature ASM cells (cl-13) (Extended Data Fig. 5b). *HHIP*, a target and inhibitor of HH-signalling[28], and the secreted BMP-inhibitor GREM2 (ref. 29) were enriched in the more mature ASM cluster (Extended Data Figs. 4a and 5d: modules −7 and −9), implicating regulation of these pathways during ASM differentiation.

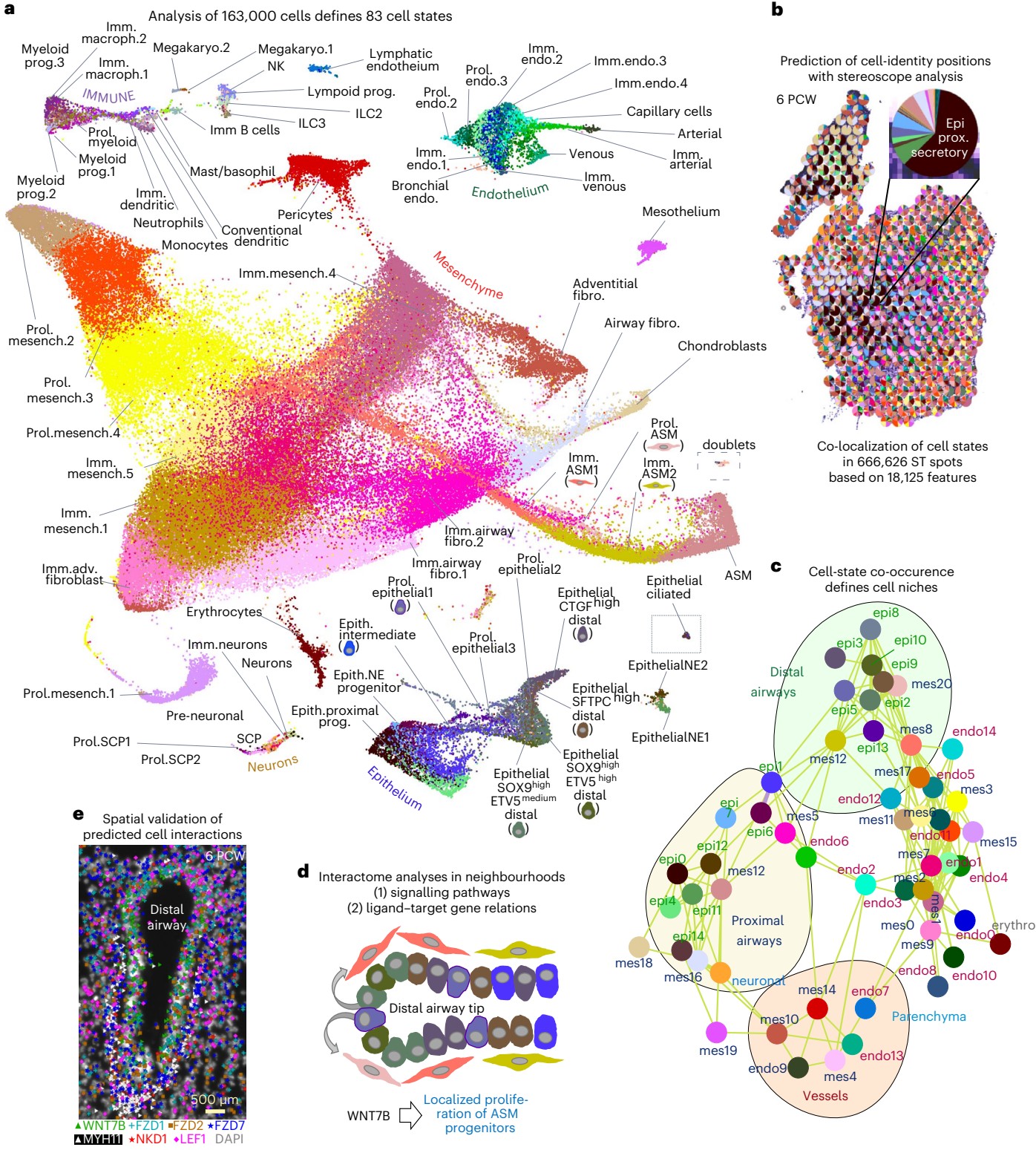

**a** Analysis of 163,000 cells defines 83 cell states

**b** Prediction of cell-identity positions with stereoscope analysis

Co-localization of cell states in 666,626 ST spots based on 18,125 features

**c** Cell-state co-occurence defines cell niches

**e** Spatial validation of predicted cell interactions

**d** Interactome analyses in neighbourhoods
(1) signalling pathways
(2) ligand–target gene relations

WNT7B → Localized proliferation of ASM progenitors

Spatial analysis localized most clusters of this trajectory in distinct positions along the developing airways (Fig. 2d,e), indicating a link between the ASM maturation states and their topology, with most immature states located peripherally and the mature ones being closer to proximal airways, as in mouse lung[15]. Mesenchymal cl-0 and cl-2 were dispersed in the parenchyma (Fig. 1d and Extended Data Fig. 4b) and highly expressed *WNT2* and *RSPO2* (Extended Data Fig. 5a,d). This is consistent with defects in ASM differentiation caused by *WNT2* inactivation in mice[30]. This suggests that precursors are evenly distributed in the peripheral parenchyma and begin to differentiate close to the bud tips.

## Two differentiation trajectories of lung fibroblasts

To complement the mesenchymal cell analysis, we focused on the two suggested fibroblast trajectories, based on the relation of the involved clusters (cl-4, cl-5, cl-16, cl-9 and cl-10) in PAGA plot (Fig. 2a and Extended Data Fig. 5e,f). ST analysis showed that cl-16 is localized around the airways, as early as 6 PCW (Fig. 2b (6)). This cluster is negative for *ACTA2* but expresses markers of other adult stromal cell types, such as *ASPN* for myofibroblasts, *SERPINF1* for AdvFs[2] and *COL13A1* characterizing a recently described lung fibroblast type found in human and mouse[31–33] (Extended Data Fig. 4a). Its unique profile and close proximity to the ASM layer (Fig. 2e,f) argued that cl-16 corresponds to an undescribed mesenchymal cell type, which we named 'airway fibroblast (AF)'. On the other hand, AdvFs were localized in bronchovascular bundles, at greater distance from the airways than AFs (Fig. 2b (5)).

scVelo and Slingshot analyses (Extended Data Fig. 5e,f) indicated that the immature fibroblasts of cl-4 either transit to immature AF2 (cl-5) and then to the mature AFs (cl-16) or produce the immature AdvFs (cl-9), which mature to the cl-10. *WNT2* and *FGF10* were expressed in the immature fibroblasts, similarly to the other immature mesenchymal clusters (Extended Data Fig. 5d) but the Netrin-receptor DCC is more selective for all three immature mesenchymal clusters and especially cl-4, suggesting a decline as differentiation proceeds (Extended Data Fig. 5g and Supplementary Table 1 (5)). Similarly, immature cells expressed *DACH1* and *ZBTB16*, whereas *MECOM* was gradually increased along the AF trajectory and the BMP-signalling targets *ID1* and *ID3* (ref. 34) along the adventitial one (Extended Data Fig. 5h). Different secreted ECM proteins such as *TNC*, *ASPN* and collagens were differentially expressed along the trajectories (Extended Data Fig. 5i). This suggests distinct roles of the embryonic lung fibroblast types in the creation of the 'scaffolding' substrates for resident lung cells.

## AF interactions with smooth muscle

Focusing on the AF trajectory, there was a gradual increase of markers such as *COL13A1* and *SEMA3E*[35] in mature cl-16 (Extended Data Fig. 4a). Spatial analyses showed that AFs surround the ASMs, with cl-16 located most proximal to ASM (Fig. 2e,f) and the more immature AF state (cl-5) in more peripheral positions (Fig. 2e). To explore potential communication routes between AF and ASM, we focused on signalling pathways emanating from the one and targeting the other (Extended Data

Fig. 6a,b). IGF, WNT and BMP pathways were among the most prominent ones (Extended Data Fig. 6c–e). The *IGF1* was mainly expressed in immature ASM2 (mes cl-12), as early as 5 PCW and increased over time (Extended Data Fig. 6f,g). The expression of the corresponding receptor, *IGF1R* was also evident at that stage, in immature AFs (mes cl-5) showing relatively stable expression until 14 PCW. The predicted IGF1-target gene, *LUM*, was expressed by AFs (Fig. 2g and Extended Data Fig. 6c) and may facilitate the alignment and formation of collagen bundles around proximal airways, as previously reported[36]. *WNT5A* was produced by ASM cells and targeted AFs through the *FZD1* receptor, in a communication pattern that intensifies overtime, as indicated by the gradually elevated expression of both proteins (Extended Data Fig. 6d,g,h). Our computational predictions suggested *BMP4* as a *WNT5A* target (Extended Data Fig. 6d), in agreement with previous in vitro experiments[37]. *BMP4* is in turn predicted to upregulate *ACTA2* expression in ASM[38], suggesting a positive feedback loop, between adjacent AFs and ASM (Extended Data Fig. 6e). Our results identify AFs as an undescribed cell type in contact with ASM and suggest their mutual signalling interactions.

## SCPs produce lung parasympathetic neurons

The trachea and lungs are innervated by the vagus nerve, containing sympathetic, parasympathetic and sensory neurons. These fibres comprise a pre-ganglionic and a post-ganglionic compartment[39,40]. Only parasympathetic ganglia are localized inside the lung, close to the airways, containing the somata of post-ganglionic neurons that innervate the ASM[41] and regulate bronchoconstriction[40]. The source for parasympathetic neurons in mice[42,43] is the neural crest-derived Schwann cell precursors (SCPs), which migrate towards trunk and cephalic ganglionic positions to differentiate into neurons, in an ASCL1-dependent process[42].

Subclustering of neuronal cells revealed eight cell states, which can be ordered into one main differentiation trajectory, resembling the transition of SCPs to neurons (Fig. 3a,b). The dataset also contains proliferating SCPs (cl-1, cl-5 and cl-7) (Extended Data Fig. 7a and Supplementary Table 1 (6)). The neuronal cl-0 and cl-3 gradually lose SCP-marker expression while increasing *ASCL1*, suggesting transient states from SCPs to neurons. cl-2 and cl-6 expressed the neuronal markers *PRPH*, *NRG1* and *PHOX2B* (Extended Data Fig. 7a), together with the acetylcholine receptors M2 and M3 (*CHRM2* and *CHRM3*) and the nicotinic acetylcholine receptor subunits α3 and α7 (*CHRNA3* and *CHRNA7*). This suggested that they can respond to acetylcholine. Similarly, they expressed acetylcholinesterase (*ACHE*) and *SLC5A7*, encoding the high-affinity choline transporter for intraneuronal acetylcholine synthesis[44] (Extended Data Fig. 7b). However, the lack *NOS1* and *VIP* (Extended Data Fig. 7a) suggests that they are still immature parasympathetic neurons.

Stereoscope analysis detected the collective signature of both SCPs and neuronal cells in the trachea at 6 PCW (Fig. 3c). Intra-lobar signal was first detected close to the trachea at 7 PCW (Fig. 3d, asterisk). At later timepoints the signal was detected more centrally, within

---

**Fig. 2 | Analysis of mesenchymal cells. a**, PAGA plot of the analysed 138,000 mesenchymal cells, from all 17 analysed donors, superimposed on their UMAP plot. Line thickness indicates the probability of the cluster connections. Colours indicate the 21 suggested clusters. ASM, airway smooth muscle; prol, proliferating; imm, immature; adv, adventitial; AF, airway fibroblast; fibro, fibroblast. **b**, Stereoscope analysis, based on ST data, showing the spatial distribution of the developing (1) mesothelial cells (cl-19), (2) pericytes (cl-14), (3) chondroblasts (cl-18), (4) ASM (cl-13), (5) AdvFs (cl-10) and (6) AFs (cl-16), in 6, 8.5 and 11.5 PCW lung sections. Red numbers: the highest percentage value of the indicated cell type. Dark red, high; grey, 0%. Tissue structure is shown by H&E staining. Scale bar, 400 μm. arw, airway; tr, trachea; prox, proximal; pcw, post conception week; br-v bundle, bronchovascular bundle. **c**, Pseudotime analysis of the ASM cells, with Slingshot showing the proliferation (cl-20) and

maturation (cl-12 and cl-13) trajectories. Same colours as in **a**. **d**, As in **b** for the ASM trajectory, in a 6 PCW lung section. **e**, Spatial localization of the ASM and AF clusters, in a 6 PCW lung section, using probabilistic cell typing (pciSeq) with HybISS data. The pie charts show the percentage of the indicated cell identities. **f**, Representative image of one out of six distal epithelial bud tips for a 6 PCW whole lung section, showing the *MYH11* (red), *IGF1* (green) and *COL13A1* (blue) detected mRNAs (HybISS) around the same airway, as in **e**. Data can be accessed at https://hdca-sweden.scilifelab.se/tissues-overview/lung/. **g**, Single-plane, confocal-microscopy image of immunofluorescence for COL13A1 (magenta), LUM (yellow) and ACTA2 (cyan), to show AFs and ASM, respectively, in an 8.5 PCW proximal airway (left). Square bracket indicates the area of the images on the right. Nuclear DAPI, grey. Scale bar, 20 μm.

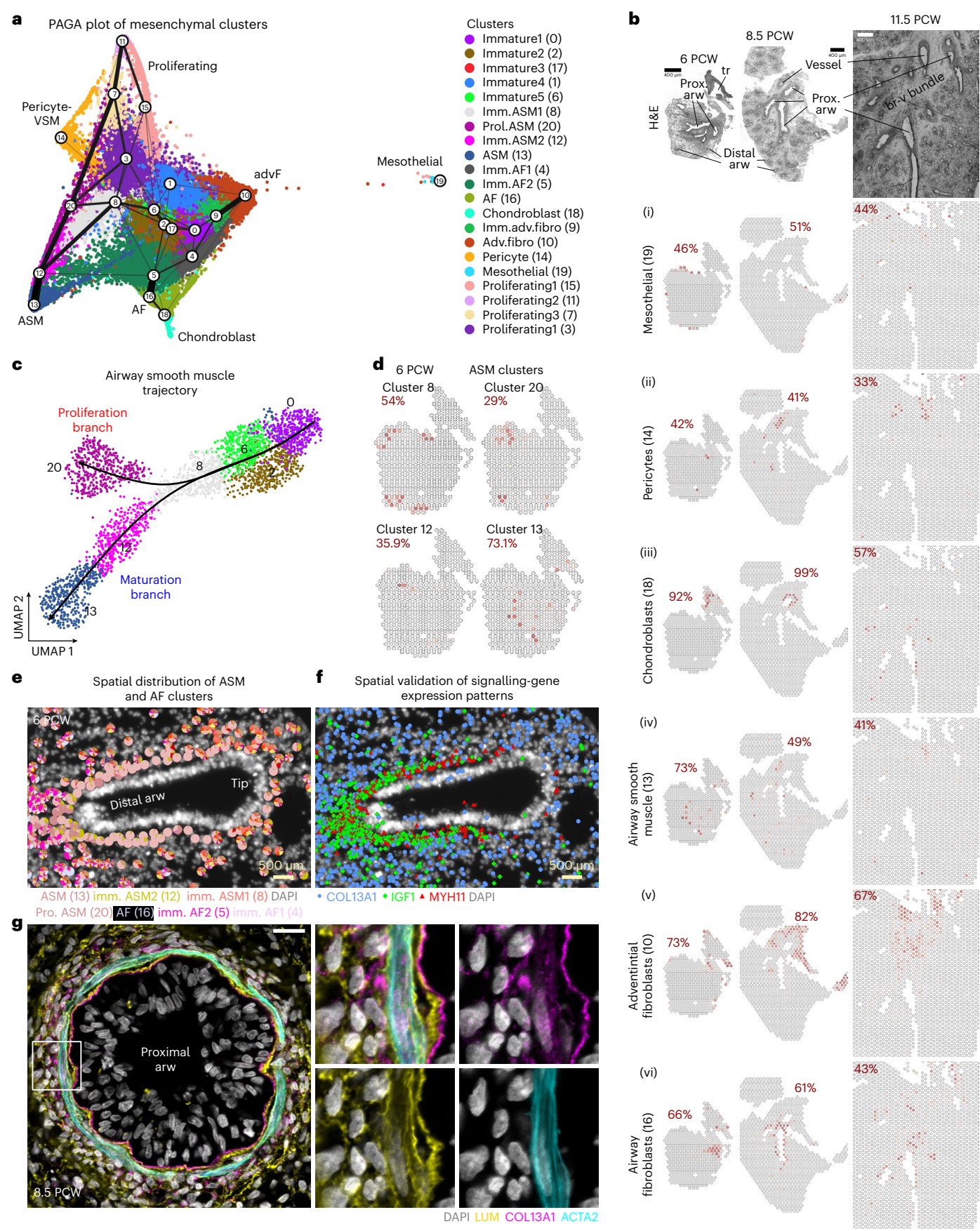

**a** PAGA plot of mesenchymal clusters

Proliferating

Pericyte-VSM

advF

Mesothelial

ASM

AF

Chondroblast

Clusters
- Immature1 (0)
- Immature2 (2)
- Immature3 (17)
- Immature4 (1)
- Immature5 (6)
- Imm.ASM1 (8)
- Prol.ASM (20)
- Imm.ASM2 (12)
- ASM (13)
- Imm.AF1 (4)
- Imm.AF2 (5)
- AF (16)
- Chondroblast (18)
- Imm.adv.fibro (9)
- Adv.fibro (10)
- Pericyte (14)
- Mesothelial (19)
- Proliferating1 (15)
- Proliferating2 (11)
- Proliferating3 (7)
- Proliferating1 (3)

**c** Airway smooth muscle trajectory

Proliferation branch

Maturation branch

UMAP 2
UMAP 1

**d** 6 PCW — ASM clusters

Cluster 8 — 54%
Cluster 20 — 29%
Cluster 12 — 35.9%
Cluster 13 — 73.1%

**e** Spatial distribution of ASM and AF clusters

6 PCW

Distal arw — Tip

500 μm

ASM (13) — imm. ASM2 (12) — imm. ASM1 (8) — DAPI
Pro. ASM (20) — AF (16) — imm. AF2 (5) — imm. AF1 (4)

**f** Spatial validation of signalling-gene expression patterns

500 μm

● COL13A1  ♦ IGF1  ▲ MYH11  DAPI

**g**

Proximal arw

8.5 PCW

DAPI  LUM  COL13A1  ACTA2

**b**

8.5 PCW

11.5 PCW

H&E — 6 PCW
tr
Prox. arw
Distal arw

Vessel
Prox. arw

br-v bundle

400 μm

(i) Mesothelial (19)
46% — 51% — 44%

(ii) Pericytes (14)
42% — 41% — 33%

(iii) Chondroblasts (18)
92% — 99% — 57%

(iv) Airway smooth muscle (13)
73% — 49% — 41%

(v) Adventitial fibroblasts (10)
73% — 82% — 67%

(vi) Airway fibroblasts (16)
66% — 61% — 43%

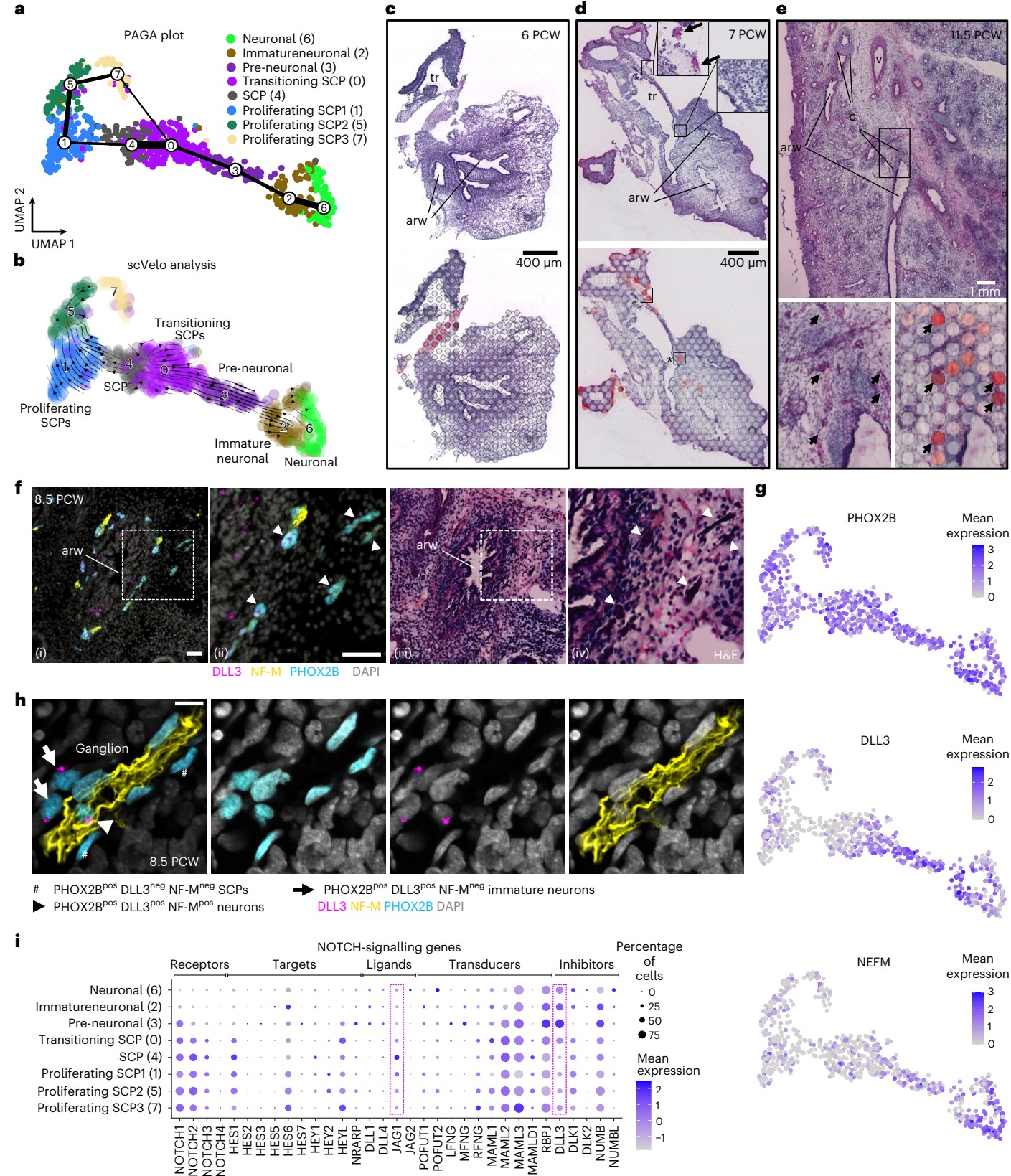

the bronchovascular bundle interstitium[19], coinciding with a distinct haematoxylin and eosin (H&E) staining pattern (Fig. 3e) that overlaps with the protein expression of the SCP and neuronal markers PHOX2B, DLL3 and NEFM (Fig. 3f). This suggests that the SCPs, presumably deriving from neural crest, enter the lung and mature to parasympathetic neurons in ganglia embedded in the bronchial interstitium.

To explore the cellular composition and differentiation states in the proposed embryonic ganglia we first stained for PHOX2B (SCPs and neurons), DLL3 (differentiating neurons[45]) and NF-M (mature neuron projections) (Fig. 3g,h). At 8.5 PCW, we found several clusters of PHOX2B[pos] cells in NF-M[pos] domains, that contained some DLL3[pos] cells, which would correspond to differentiating neurons. We further

**Fig. 3 | Parasympathetic neuron development in the embryonic lung. a**, PAGA plot of the analysed 752 neuronal cells, from 10 analysed donors (Methods), superimposed on their UMAP plot. Line thickness indicates the probability of the cluster connections. Colours indicate the eight suggested clusters. **b**, scVelo-analysis on the neuronal cells. Colours as in **a**, and direction of arrows shows the future state of the cells. **c–e**, Stereoscope neuronal score on 6 (**c**), 7 (**d**) and 11.5 (**e**) PCW lung sections. Top: high-resolution H&E images. Bottom: stereoscope score of neuronal cells (SCPs and neurons, together). Arrows: ST spots with high percentage of neuronal cells, possibly corresponding to ganglia. Asterisk: possible ganglion, within lung. Dark red, high; grey, 0%. 'arw', airway; 'tr', trachea; 'v', vessel; 'c', cartilage rings. Interactive inspection of the presented data can be accessed at https://hdca-sweden.scilifelab.se/tissues-overview/lung/. **f**, (i) Low-magnification image of immunofluorescence for the PHOX2B (cyan), DLL3 (magenta) and NF-M (yellow) on an 8.5 PCW lung section. Nuclei: DAPI (grey). Parasympathetic ganglia were detected around an airway. (ii) Magnified area designated by square bracket in (i). Arrowheads: positive ganglia for the

analysed markers. arw, airway. (iii) H&E staining of the same tissue section, after immunofluorescence and image acquisition. (iv) Magnified area corresponding to the square bracket in '(iii)'. The arrowheads indicate the same positions as in '(ii)', showing that the structures with intense H&E staining correspond to ganglia. Scale bar, 50 μm. **g**, UMAP plots of *PHOX2B* (SCPs and neurons), *DLL3* (developing neurons) and *NEFM* (NF-M, mature neurons). Expression levels: $\log_2$(normalized UMI counts + 1) (library size, normalized to 10.000). **h**, Immunofluorescence of PHOX2B (cyan), DLL3 (magenta) and NF-M (yellow). Nuclei: DAPI (grey). Scale bar, 20 μm. Hashes: PHOX2B$^{pos}$ DLL3$^{pos}$ NF-M$^{neg}$ SCPs. Arrows: PHOX2B$^{pos}$ DLL3$^{pos}$ NF-M$^{neg}$ immature neurons. Arrowhead: PHOX2B$^{pos}$ DLL3$^{pos}$ NF-M$^{pos}$ neuron. DLL3 staining pattern agrees with its previously reported localization in *cis*-Golgi, to sequester unprocessed NOTCH1-protein and render cells insensitive to NOTCH signalling[74]. **i**, Balloon plot of NOTCH-signalling gene expression in neuronal clusters, including receptors, targets, ligands, transducers and inhibitors[75]. Brackets highlight JAG1 and DLL3. Balloon size: percentage of positive cells. Colour intensity: scaled expression. Blue, high; grey, low.

explored this by analysing the characteristic TFs SOX10, ASCL1 and ISL1, which are sequentially activated along the trajectory (Extended Data Fig. 7c–e). We detected SOX10$^{pos}$ SCPs, SOX10$^{neg}$-ASCL1$^{pos}$ neuronal precursors and ISL1$^{pos}$ neurons, consistent with the differentiation steps proposed by the pseudotime analysis. The selective expression of ASCL1 and DLL3 in subclusters of the ganglionic cells prompted us to interrogate the expression of NOTCH-signalling pathway genes in the clusters (Fig. 3i). The selective expression of *JAG1* in SCPs suggested that it activates NOTCH signalling in parasympathetic ganglia, similarly to its role in mouse limb nerves, which also derive from neural crest[46].

### Early developmental trajectories of epithelial differentiation

We subclustered epithelial cells into 15 groups (Fig. 4a) and annotated them on the basis of known markers (Extended Data Fig. 8a and Supplementary Table 1 (7)), spatial distribution (Fig. 4b and Extended Data Fig. 8b) and their trajectory relationships illustrated by PAGA plot and scVelo analyses (Extended Data Fig. 8c,d). We detected four distal cell identities (cl-10, cl-2, cl-3 and cl-9) and seven proximal ones, corresponding to ciliated (cl-14), secretory (cl-0), neuroendocrine (NE) cells (cl-11 and cl-12) and their progenitors (cl-6, cl-7 and cl-4). We also found an intermediately located population (cl-1) and three proliferating cell states (cl-8, cl-13 and cl-5), which were preferentially localized in distal airways (Extended Data Fig. 8b). Surprisingly, we did not detect any cluster with characteristic basal cell features but only a few *TP63*$^{pos}$ cells within cl-7, being negative for typical embryonic[47] or adult[2] basal markers (Extended Data Fig. 8e,f). Similar to the scRNA-seq analysis, immunofluorescence of 8.5 and 14 PCW lung sections showed TP63$^{pos}$ cells in large airways with only a small fraction being KRT5$^{pos}$ at only 14 PCW (Extended Data Fig. 8g). This suggests that basal cells begin to differentiate at 14 PCW in the intra-lobar airways.

In distal airways, epithelial cl-2, cl-3, cl-9 and cl-10 were positive for *SOX9* and *ETV5* (refs. 6,48) (Extended Data Fig. 8a,b and Fig. 4b,c). Among them, cl-2 and cl-10 cells highly expressed *SOX9* and

were located in the most distal part of the bud tips. Trajectory analyses (Extended Data Fig. 8c,d) and their topology suggested that they function as the source of the remaining two distal clusters, which were predominantly composed of later-timepoint cells (>10 PCW) (Extended Data Fig. 9a). Accordingly, cl-9 included *SFTPC*$^{high}$ cells co-expressing A*CSL3*, which participates in lipid metabolism[49], a prerequisite for surfactant biosynthesis[50] (Extended Data Fig. 9b,c). By contrast, cl-3 cells were found scattered in the distal epithelium as early as 5 PCW (Extended Data Fig. 8b) and expressed elevated *CTGF* levels (Extended Data Fig. 9d), a growth factor implicated in mouse alveolar development[51] and in stimulation of fibroblasts during mouse lung fibrosis[52]. Immunofluorescence for KRT17, another cl-3 selective marker (Extended Data Fig. 8e) confirmed the existence of sparsely distributed Ecad$^{pos}$ KRT17$^{pos}$ cells in the 14 PCW distal airway epithelium (Fig. 4d). Overall, these cells share gene expression similarities with 'basaloid' cells (Extended Data Fig. 9f,g and Supplementary Table 1 (8)), a pathogenic cell state in interstitial pulmonary fibrosis[4,53]. However, the embryonic clusters are distinguished by marked differences, as they are TP63$^{neg}$ and are localized in the luminal rather than basal part of the epithelium (Fig. 4d).

### Cell communication patterns in the distal lung compartment

We utilized the definitions of cell neighbourhoods (Fig. 1c) to explore candidate cell communication pathways in the distal lung compartment (Viewer: CellChat). FGF signalling was among the most prominent predictions (Fig. 4e) with *FGF10* being mainly expressed in scattered mesenchymal cells (cl-0) around the epithelium (Fig. 4f and Extended Data Fig. 4b). This expression pattern differs in the mouse embryonic lungs, where *FGF10* is focally expressed at the bud tips to induce branching[54]. This difference might explain why FGF10 induces cyst formation instead of branching in human explants[55]. Additional FGF-ligand genes (Fig. 4f,g) were detected in the distal epithelium, defining both mesenchymal and epithelial cells as sources. For example, *FGF18* and *FGF20*

**Fig. 4 | Epithelial diversity in developing human lungs. a**, UMAP plot of 10,940 epithelial cells, from all 17 analysed donors. Colours indicate the 15 suggested clusters. Dotted outlines: main cell groups of proximal (magenta), proliferating (grey) and distal cells (black). **b**, Heat map showing the spatial correlation of the indicated clusters, based on stereoscope scores (ST data). Positive correlations, red; negative correlations, blue. Brackets: distal, intermediate and proximal main patterns. **c**, Region of interest (ROI) showing a 14 PCW distal airway, analysed with SCRINSHOT. *SOX2* (cyan), *SOX9* (red), *ETV5* (yellow), *SFTPC* (grey), NKX2-1 (grey, not shown in merge image) and DAPI (blue). Scale bar, 40 μm. **d**, Single-plane confocal-microscopy image of immunofluorescence for the characteristic basaloid marker KRT17 (magenta) in addition to Ecad (cyan), showing KRT17$^{pos}$ Ecad$^{pos}$ cells in a 14 PCW lung section. DAPI, blue. Scale bar, 10 μm. **e**, CellChat heat map showing the sender, receiver, mediator and influencer roles of the

different epithelial clusters described in **a** for the FGF-signalling pathway. Colour intensity shows the importance of the cluster contribution to each role. Dark red, high; white, low importance. All identified communication patterns can be accessed at https://cellchat.serve.scilifelab.se/. **f**, Balloon plot of FGF ligands, receptors and target expression levels, in distal lung clusters. Epithelial intermediate (cl-0) and ASM (cl-13): control cell states (not in the specific neighbourhood, with grey shadow). Balloon size: percentage of positive cells. Colour intensity: scaled expression. Blue, high; grey, low. **g**, HybISS in situ validation of FGF-pathway genes. DAPI, nuclei (top left). Top: general epithelial marker *EPCAM*, *FGF18* and *FGF20* ligands. Middle: *FGFR1-4* receptors. Bottom: *ETS1*, *ETV3*, *ETV5* and *SPRY2* targets. Scale bar, 500 μm. Data can be accessed at https://hdca-sweden.scilifelab.se/tissues-overview/lung/.

were detected in distal epithelium by both scRNA-seq (cl-2, cl-3, cl-9 and cl-10) and HybISS. The localized expression of *FGFR2*, *FGFR3* and *FGFR4* agreed with an independent study[55]. Potential FGFR downstream targets, such as *ETV5* (ref. 56) and *SPRY2* (ref. 57), were detected in distal epithelium, suggesting a potential epithelial-intrinsic function for FGF signalling (Fig. 4f,g). Another prominent predicted target of epithelial FGFR activation is *SOX9* (Extended Data Fig. 9h), consistent with its reported regulation by FGF/Kras[48,55].

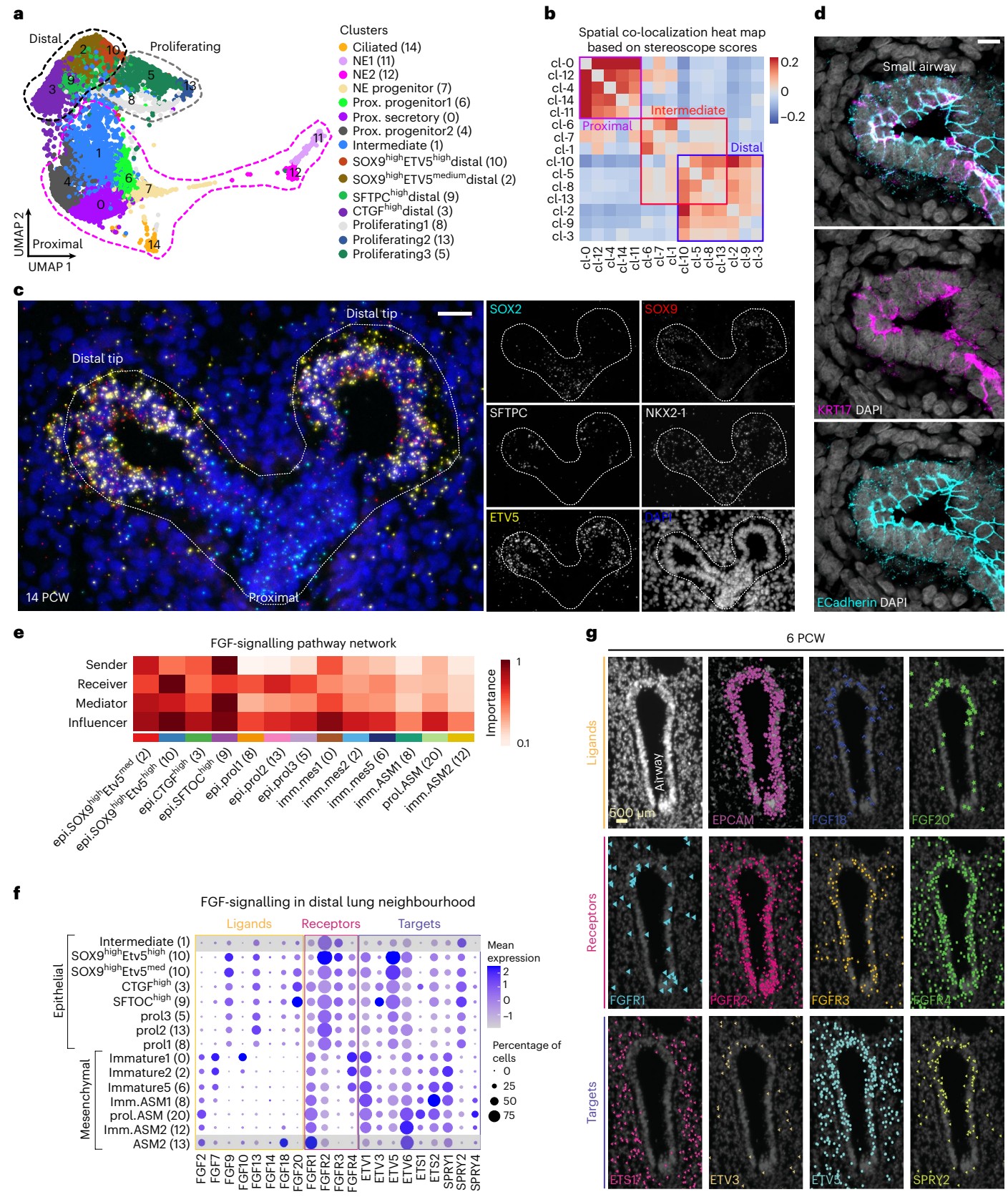

## Distinct steps in proximal airway cell differentiation

The secretory (cl-0 and cl-4), ciliated (cl-14) and NE (cl-11 and cl-12) clusters were located in the most proximal airway positions. However, their putative progenitors (cl-6 and cl-7) were found in slightly more distal positions (Fig. 4b, Viewer: HybISS). The *FOXJ1*[pos] cl-14 cells expressed only early ciliogenesis genes, suggesting an early differentiation state (Extended Data Fig. 9i and Supplementary Table 1 (24)). The major difference between secretory cl-0 and cl-4 was the high levels of *HOPX* and *KRT17* in cl-4 (Extended Data Fig. 8a), which also expressed activated epithelial markers (Extended Data Fig. 9g), similar to the distal epithelial cl-3. These cl-0 and cl-4 cells showed similar spatial distribution (Fig. 4b and Extended Data Fig. 8b), but cl-4 was enriched for migration-related genes (Extended Data Fig. 9j and Supplementary Table 1 (25)). Thus, cl-4 may correspond to a transient progenitor state giving rise to the 'default', static airway secretory cl-0. PAGA plot (Extended Data Fig. 8c) and pseudotime (Fig. 5a,b) analyses suggested that cl-6 cells can function as a source for either secretory cl-0 or NE-progenitor cl-7 cells, which further progresses towards the NE cl-12 and cl-11 states. Differential expression analysis along the two trajectories identified 569 genes that were grouped in nine modules (Supplementary Table 1 (18), top 10, and Fig. 5c). Among the earliest activated genes in the secretory trajectory, we detected *YAP1* and the WNT extracellular inhibitor *GPC5* (Fig. 5c, module 6) (refs. 58,59). These were followed by increased levels of the characteristic secretory marker *SCGB3A2* and the NOTCH-signalling targets *HES1* and *HES4* (Fig. 5c, module 9), further arguing for an evolutionary conserved role of NOTCH-signalling in airway secretory cell differentiation[60] and maintenance[61].

## Distinct topologies and possible functions of NE identities

In the NE trajectory, cl-7 probably represents a progenitor expressing low levels of *ASCL1*, a critical factor in NE cell differentiation[62] (Fig. 5c, module 4). The differentially expressed TFs along the secretory and NE trajectories included the direct ASCL1-target, *MYCL*[63], which was transiently expressed along the NE trajectory (Fig. 5d and Extended Data Fig. 9k). The NE progenitor cl-7 was connected by few cells with the NE2 (cl-12), creating a stalk that splits in two directions, one towards the remaining NE2-cells and the other towards NE1-cells (cl-11) (Fig. 5a). In this part, gene module 4 contained *ASCL1*, its direct target *IGFBP5* (ref. 64), together with *HES6* (ref. 65) (Fig. 5c). Finally, at the part towards NE1 cells, module 1 contained *NEUROD1* (Extended Data Fig. 9l), its target *HNF4G*[63] (Fig. 5c, module 1, and Extended Data Fig. 9m) and *SSTR2* (Fig. 5c, module 1). Gene expression comparison between cl-11 and cl-12 (Extended Data Fig. 9n and Supplementary Table 1 (9)) showed that cl-12 produces the characteristic pulmonary neuropeptides *GRP* and *CALCA* together with SST, whereas cl-11 expresses *GHRL* and *CRH*. Gene Ontology (GO) analysis for enriched biological processes suggested hormone secretion (GO:0030072) and neuronal axon guidance (GO:0007411), as characteristic terms for cl-11 compared with cl-12 (Extended Data Fig. 9o,p and Supplementary Table 1 (26, 27)). The NE1 cells (cl-11) resemble a recently identified NE cell type in human embryos[7].

To investigate the spatial arrangement of NE clusters, we used SCRINSHOT to detect a panel of 31 genes, encompassing NE, epithelial and mesenchymal markers (Extended Data Fig. 10a–d). We defined NE-specific patterns by segmenting the sections in hexagonal bins (7 μm width), approximating the size of epithelial cells. Among 20,351 bins expressing general epithelial and characteristic NE genes (Methods), we found three main NE-associated categories, corresponding to NE-progenitors, *GRP*[pos] and *GHRL*[pos] NE-cells in situ (Extended Data Fig. 10e,f). These expression patterns match the ones of scRNA-seq analysis. *GHRL*[pos] NE-cells were located exclusively in the most proximal airways, while NE progenitors and *GRP*[pos] NE-cells were less restricted in their location along the airway proximal–distal axis (Extended Data Fig. 10d,g). Immunofluorescence analysis confirmed that GRP[pos] and GHRL[pos] NE cells are differentially distributed along the airways (Extended Data Fig. 10h).

As different levels of graded NOTCH-signalling activation are required for NE and non-NE cell-fate specification in the airway epithelium[66], we interrogated the proximal clusters for the expression of NOTCH-signalling genes (Fig. 5e). Both NE clusters (cl-11 and cl-12) expressed *HES6* (a pathway target and inhibitor[65]). However, cl-12 expressed higher levels of *JAG1* and *DLL3* (a NOTCH cell-autonomous inhibitor[67]), in addition to low levels of *JAG2* and *DLL1*. This suggests that cl-12 cells are a source of NOTCH signalling and that they are less capable of receiving it. The downregulation of *DLL3* might be permissive for lower NOTCH-signalling activation, contributing to the cl-11 gene-expression programme defined by the *NEUROD1*, *RFX6*, *HNF4G* and *NKX2-2* TFs (Fig. 5d and Extended Data Fig. 9l,m). Upstream, in the trajectory, at the bifurcation of secretory (cl-6) and NE-progenitor (cl-7) states, the repressor *REST*[68] and the receptor *NOTCH2* showed similar expression levels, but *HES6* and *NOTCH1* were higher expressed in the NE-progenitor cluster, suggesting differences in strength or duration of NOTCH signalling[69,70]. NOTCH2 activation in proximal progenitors (cl-6) is expected to be more potent[69,70], promoting the secretory differentiation.

Overall, the pseudotime analysis suggests two sequential but distinct NOTCH-signalling events, utilizing different ligands and intracellular effectors: one promotes secretory differentiation, and the other controls the transition of cl-12 to cl-11 (Fig. 5f). Further interactome analysis revealed another unique communication pattern between the two NE clusters involving somatostatin (*SST*) expressed by cl-12 and its receptor *SSTR2* in cl-11 (Fig. 5g,h).

In summary, we mapped the distinct topologies and developmental trajectories of airway secretory and NE identities from naïve epithelial cells in the embryonic lung. Each trajectory contains distinct candidate regulators of NOTCH signalling for the respective cell-state transitions.

## Mesenchymal cell zonation patterns along two airway axes

Stromal cell populations in fully grown lungs show distinct distributions along the proximal–distal axis of the airways[2]. They also show specialized radial arrangements surrounding each major airway, with ASM adjacent to the epithelium (centre) and AdvFs and chondroblasts positioned more peripherally. To explore the spatial organization of different mesenchymal trajectories (AF, ASM and AdvF) relative to the

**Fig. 5 | Analysis of developmental trajectories in proximal epithelium. a**, UMAP plot of proximal clusters and pseudotime of secretory and NE trajectories, estimated by Slingshot, containing cells from all 17 analysed donors,. Colours as in Fig. 4a. Asterisk: bifurcation point of the two NE clusters. **b**, scVelo analysis on the proximal epithelial cells. Colours as in **a**, and direction of arrows shows the future state of the cells. **c**, Heat map of the top-ten markers of each stable gene module of the 569 differentially expressed genes (Supplementary Data 3) (bootstrap values module 1: 0.60, module 2: 0.69, module 3: 0.84, module 4: 0.57, module 5: 0.80, module 6: 0.73, module 7: 0.61, module 8: 0.55, module 9: 0.85) along the two trajectories, shown in **a**, according to tradeSeq. Colour intensity: scaled expression. Dark red, high; grey, low. **d**, Balloon plot of the top-ten selective TFs in the proximal epithelial secretory and NE clusters. The top-20 TF genes (based on average log$_2$ fold change) were sorted according to the percentage of positive cells, and the top-10 TFs were plotted. Gene order follows the cluster order. **e**, Balloon plot of NOTCH-signalling components[75], in addition to the neuronal gene inhibitor *REST*[68], the TF *YAP1*, the secretory marker *SCGB3A2* and the NE markers *MYCL*, *ASCL1*, *GRP*, *NEUROD1* and *GHRL*. In all balloon plots, balloon size: percent of positive cells; colour intensity: scaled expression. Blue, high; grey, zero. **f**, Schematic representation of the suggested NOTCH-signalling function on secretory and NE cell specification. **g**, CellChat hierarchical plot of SST-–SSTR2 communication pattern between the two NE cell states. **h**, Single-plane confocal-microscopy image of immunofluorescence for the SST (cyan), SSTR2 (magenta) and NE1 (cl-11) marker GHRL (yellow) to validate the communication pattern between the two NE-cell SSTR2[pos] GHRL[pos] cells with the adjacent SST[pos] NE2 (cl-12) cells. Cyan arrows: SST[pos] cells. Yellow arrows: GHRL[pos] SSTR2[pos] cells. Scale bar, 5 μm.

growing airways on the tissue level, we defined two axes. A proximal–distal one, which was defined by the graded expression of proximal (*SOX2* and *SCGB3A2*) and distal (*ETV5* and *TPPP3*) epithelial genes, validated by HybISS (Methods) and a radial one, extending from the airway centre towards peripheral positions in the mesenchyme. We positioned the ST spots and HybISS-annotated cells corresponding to immature and differentiated states of AdvFs (mes cl-10), ASM (mes cl-13) and AFs (mes cl-16) relative to these two airway-dependent axes (Fig. 6 and Methods). This analysis revealed that the immature cell states

occupy predominantly distal and peripheral positions relatively to the airway branches. By contrast, the more mature mesenchymal clusters are found proximally and centrally located. In particular, the most immature ASM clusters (cl-0, cl-2 and cl-6) were the most peripheral. More differentiated clusters (cl-8, cl-20 and cl-12) were found closer to the airways and in more proximal positions, whereas the most mature ASM (cl-13) was found proximal and tightly associated with the airways. At all three consecutive timepoints (6, 8.5 and 11.5 PCW), the immature fibroblast (mes cl-4) was consistently found more proximal compared

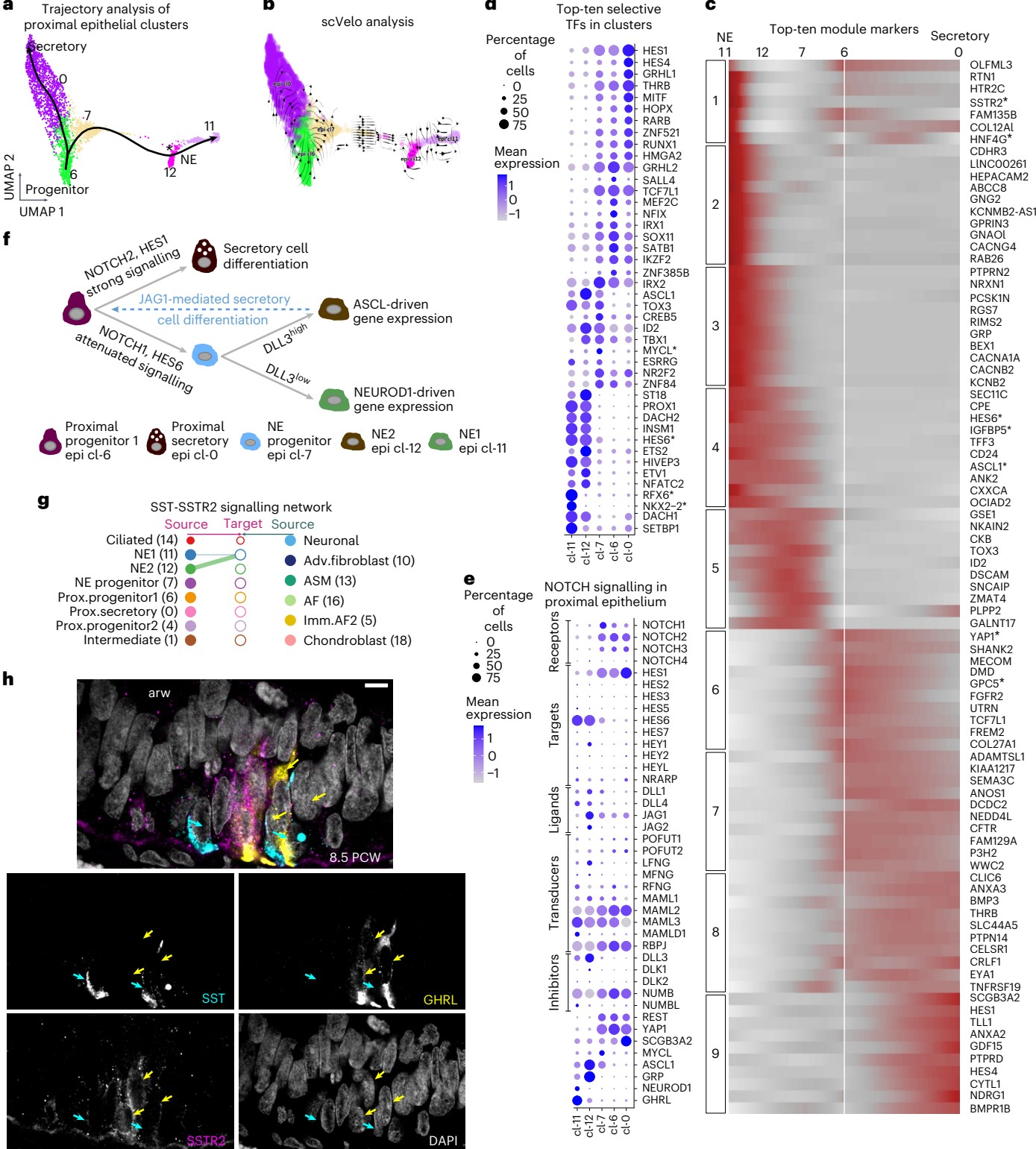

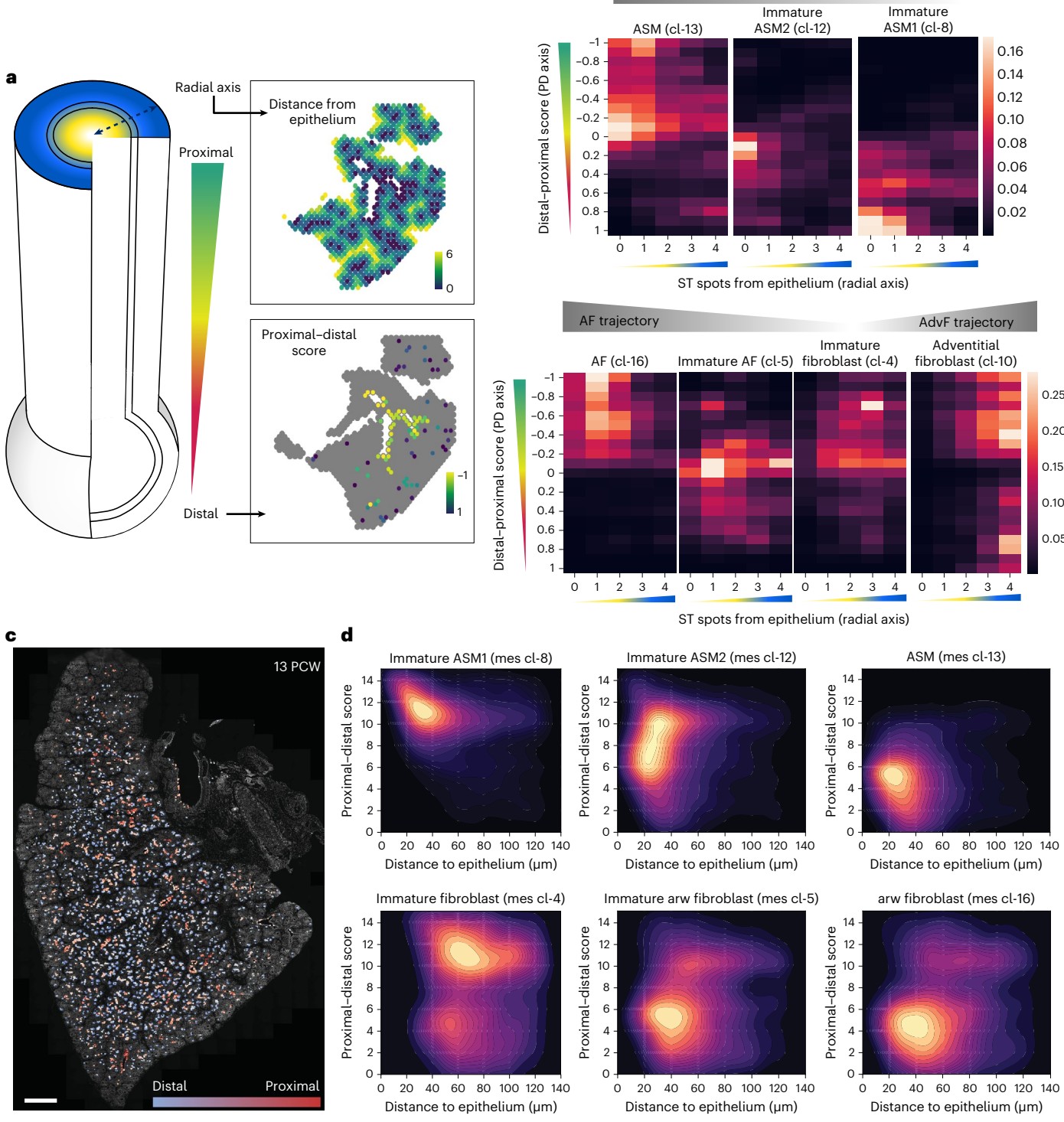

**Fig. 6 | Assessing the molecular complexity of embryonic human airways.**
**a**, Left: schematic representation of the radial and proximal–distal airway-dependent axes. Right: spatial maps of the radial (top) and proximal–distal (bottom), scores of an 8.5 PCW lung section, analysed by ST. Colour indicates distance from epithelium (number of ST spots). Yellow, high; dark green, zero. Proximal–distal score as scaled aggregated expression of SOX2, SCGB3A2 (proximal) and ETV5, TPPP3 (distal). Proximal, –1; distal, 1. **b**, Heat maps of ASM-, AF- and AdvF-related cluster-density scores along the two analysed axes. Colour indicates relative cell frequency in the indicated position. Yellow, high; black, zero. **c**, Proximal–distal axis score of the epithelium of a 13 PCW lung section, analysed by HybISS. DAPI, grey; proximal, red; distal, blue. Scale bar, 1,000 µm. **d**, Density maps of ASM and AF clusters, showing their distribution along proximal–distal axis (y axis) and their distance from the epithelium (x axis), as in **a** and **b**. Colour indicates relative cell frequency in the indicated position. Yellow, high; black, zero.

with the ASM progenitor clusters (viewer: ST). This argues for the presence of a peripheral central zone of mesenchymal progenitors giving rise to AdvFs, AFs and chondroblasts and reveals an early origin of radial patterning in the mesoderm. We suggest that undifferentiated cells from the distinct progenitor regions proliferate and continuously differentiate while migrating radially towards the centre and their functional positions, similarly to the model of the mesenchymal progenitor niche in the mouse lung[15].

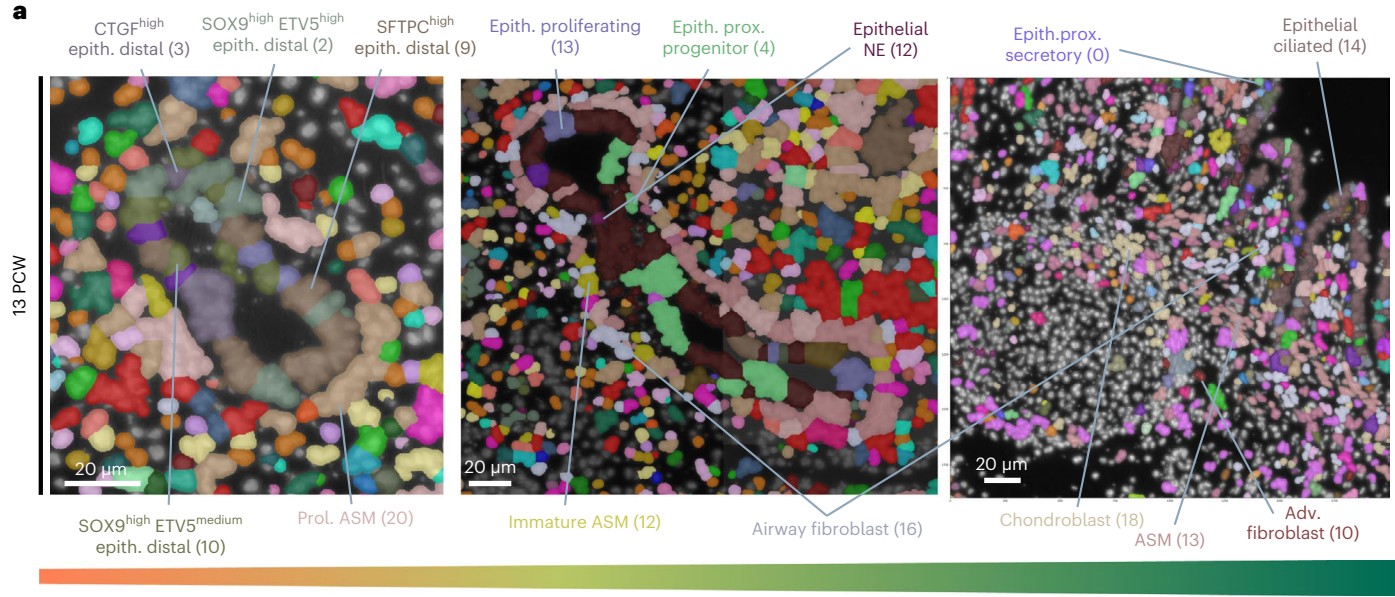

**Fig. 7 | Synopsis of the spatial organization and communication in the developing human lung. a**, Spatial cell-type maps of distal (left), intermediate (middle) and proximal (right) airways. Segmented nuclei are coloured according to the most probable, predicted cell type according to PciSeq, using HybISS data. Colours as in Fig. 1a. **b**, Scheme of the cellular and molecular complexity in developing lung. The included cell types were identified via scRNA-seq,

and their spatial context was defined by spatial methods. CellChat-predicted communication patterns: curved arrows. NicheNet-predicted ligands (black) and corresponding target genes or outcome: cyan text. Bottom: description of all involved cell types and sensory neurons (not found in scRNA-seq). Spatial and interactome analyses data can be accessed at https://hdca-sweden.scilifelab.se/tissues-overview/lung/.

## Cell heterogeneity and possible communication patterns

The spatial probabilistic methods (PciSeq[71] and Tangram) generated systematic spatial maps of several stages, showing the cellular composition of distinct organ compartments over time (Fig. 7a). On the tissue level, this allows the definition of spatial rules of tissue organization and estimation of developmental origins by interrogating the relative positions of pseudotime trajectories. A graphical representation of the developing lung shows a summary of mature and intermediate cell states, localized in distinct tissue positions, creating cell 'neighbourhoods' with specific communication patterns (Fig. 7b).

We integrated our scRNA-seq data with the HybISS, ST and SCRINSHOT spatial analyses, together with the CellChat results in the TissUUmaps viewing tool (https://hdca-sweden.scilifelab.se/tissues-overview/lung/). This portal provides an open interactive atlas of early lung development that directly facilitates exploration, sharing and hypothesis building.

## Discussion

We have generated a systematic topographic atlas of the developing human lung, combining gene expression profiling by scRNA-seq with spatially resolved transcriptomics on intact tissue sections. We identified 83 cell states and inferred developmental trajectories leading to a remarkable heterogeneity reflecting the structural and functional complexity of the lung. Although we present an extensive analysis of weekly intervals during the first trimester, our data have a few limitations. Our first datapoint is at 5 PCW and we analysed only about 180,000 cells. Earlier and broader sampling is likely to uncover additional diversity and infer more precise trajectories than the proposed ones. We aimed to collect and analyse freshly dissociated cells, omitting tracheas, without enrichment for specific populations. The lack of enrichment may have hampered detection of rare, fragile or difficult-to-dissociate cells. Indeed, we detected chondroblasts and mesothelial cells only in the samples deriving from earlier timepoints. We performed iterative clustering, where a conservative first clustering was followed by subclustering of the major populations. Although most of the subclusters showed distinct topologies and gene expression profiles, some of the cell states may result from overclustering, which is difficult to define because of the presence of immature but committed states of distinct cell types. Finally, we have described the spatial diversity of the developing lung mainly at the messenger RNA level, relating this diversity to the proteome and further to physiological functions remains a future task.

We suggest that the diversity of gene expression patterns in the developing human lung can be explained at distinct but hierarchically coupled levels. First, the major cell classes of epithelial, endothelial, immune, stromal and neuronal cells are characterized by distinct gene expression programmes of their ancestries from distinct germ layers: endoderm, mesoderm and ectoderm. We show several levels of subdivisions in each of these classes, during the first trimester. For example, within the endothelial group there are lymphatic, venous, arterial, bronchial and capillary clusters characterized by distinct regulatory and functional gene-expression profiles (Supplementary Note 1). Second, some cell clusters show region-specific gene expression profiles, presumably reflecting their developmental history. This is exemplified by the separation of proximal and distal compartments in the epithelium. The SOX2$^{pos}$-proximal and the SOX9$^{pos}$-distal domains are specified earlier and are maintained during the glandular stages. This suggests that transcriptional networks are conveyed into the later diversification of more specialized cell states specific to each region. Our spatial analysis illustrates this by the striking correlation of characteristically different radial arrangements of AFs and ASM states along different positions of the epithelial proximal–distal axis. This suggests that the different values of the proximal–distal axis intersect with distinct values of a radial axis visualized by the organization of surrounding smooth muscle and fibroblast states. The potential regulatory relationships between these axes are unknown. A third level of

diversification results from cell communication patterns within local environments reflecting inducible or transient regulation of gene modules. The integration of single-cell sequencing with ST data defined specific neighbourhoods for most of the cell states. Our curated interactome analyses predicted several known and new examples of this organization level. They include the activation of NOTCH signalling between the SCP and neuronal states[46], within parasympathetic ganglia.

Lung diseases are major causes of death worldwide[72]. An outstanding challenge for medical research is to define deviation points from normal cellular trajectories at the start and during the advancement of lung pathologies and to analyse cellular responses after treatments[73]. Our atlas of early human lung development revealed several distinct cell states and proposed their interactions with neighbours and progression along differentiation trajectories.

As single-cell analysis technologies are increasingly used in the description of detailed cell-state trajectories in disease, we believe that our integrated scRNA-seq data, with spatially resolved transcriptomics and local interactome analyses in an open, interactive portal will provide a useful resource towards understanding and reversal of pulmonary disease progression.

## Online content

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

[1]Science for Life Laboratory, Solna, Sweden. [2]Department of Molecular Biosciences, Wenner-Gren Institute, Stockholm University, Stockholm, Sweden. [3]Department of Biochemistry and Biophysics, Stockholm University, Stockholm, Sweden. [4]Division of Molecular Neurobiology, Department of Medical Biochemistry and Biophysics, Karolinska Institute, Stockholm, Sweden. [5]Department of Information Technology, Uppsala University, Uppsala, Sweden. [6]BioImage Informatics Facility, Science for Life Laboratory, SciLifeLab, Sweden. [7]Science for Life Laboratory, Department of Gene Technology, KTH Royal Institute of Technology, Stockholm, Sweden. [8]Department of Neurobiology, Care Sciences and Society, Karolinska Institutet, Stockholm, Sweden. [9]Science for Life Laboratory, School of Engineering Sciences in Chemistry, Biotechnology and Health, KTH - Royal Institute of Technology, Stockholm, Sweden. [10]Center for Infectious Medicine, Department of Medicine Huddinge, Karolinska Institutet, Stockholm, Sweden. [11]Molecular Pneumology, Cardiopulmonary Institute, Justus Liebig University, Giessen, Germany. [12]These authors contributed equally: Alexandros Sountoulidis, Sergio Marco Salas. ✉e-mail: Mats.nilsson@scilifelab.se; Christos.Samakovlis@scilifelab.se

## Methods

### Human lungs

The tissue donors were recruited among pregnant women after their decision to terminate their pregnancy. The referral to hospitals was done by a central office for all abortion clinics in the Stockholm region, and according to our information it was random. The recruitments were done by midwifes who were not involved in the conducted research. Thus, there was no bias regarding which women were recruited. Inclusion criteria: 18 years of age or older and fluent in Swedish. Exclusion criteria: abortions performed for any medical reasons, by socially compromised women and/or by women showing any signs that the consent may not be informed. All women provided written consent for tissue usage for research purposes and for their ability to withdraw their consent at any time. There was no compensation to the tissue donors.

The use of human foetal material from the elective routine abortions was approved by the Swedish National Board of Health and Welfare and the analysis using this material was approved by the Swedish Ethical Review Authority (2018/769-31). After the clinical staff acquired the informed written consent by the donor, the retrieved tissue was transferred to the research prenatal material. The lung samples were retrieved from foetuses between 5 and 14 PCW.

### Tissue treatment for spatial analyses

One of the two lungs (preferentially the left), from each donor, was snap frozen in cryomatrix and further used for histological analyses. We cut 10–12-µm-thick tissue sections with a cryostat (Leica CM3050S or analogue) and collected them onto poly-lysine-coated slides (VWR cat. no. 631-0107) for SCRINSHOT and immunofluorescence or Superfrost Plus (VWR cat. no. 48311-703) for in situ sequencing (ISS). Sections were left to dry in a container with silica gel or at 37 °C for 15 min and then stored at −80 °C until usage.

### Tissue dissociation of human embryonic lungs

For tissue dissociation, tracheas were removed and lungs were finely minced. For later timepoints, lobes were first dissected into smaller pieces. Then, they were digested in 4 U ml⁻¹ Elastase (Worthington, cat. no. LS002292), 1 mg ml⁻¹ of DNase (Worthington, cat. no. LK003170) in Hanks' balanced salt solution (HBSS) (Gibco, cat. no. 14170) at 37 °C ranging between 30 min and 3 h depending on age (older timepoints require longer digestion times). HBSS supplemented with 2% fetal calf serum (FCS) (Gibco, cat. no. 10500064) was used for the whole procedure. The tissues were triturated with glass Pasteur pipettes every 15–20 min to enhance dissociation. After digestion, the cell suspension was filtered in a 15 ml Falcon tube using a 30 µm cell strainer (CellTrics, Sysmex), to remove clumps and debris. The cell suspension was kept ice cold and was diluted (roughly 1:2) with ice-cold HBSS. The filtered cells were pelleted at 200g for 5 min at 4 °C and the pellet resuspended in a small volume of calcium- and magnesium-free HBSS (Gibco, cat. no. 14170) and transferred to 1.5 ml Eppendorf tubes pre-coated with 30% BSA (A9576, Sigma-Aldrich). A Bürker chamber was used for cell counting.

### scRNA-seq of human embryonic lung cells

scRNA-seq was carried out with the Chromium Single Cell 3′ Reagent Kit v2 and v3. Cell suspensions were counted and diluted to concentrations of 800–1,200 cells µl⁻¹ for a target recovery of 5,000 cells on the Chromium platform. Downstream procedures including cDNA synthesis, library preparation and sequencing were performed according to the manufacturer's instructions (10X Genomics). Libraries were sequenced on an Illumina NovaSeq 6000 (Illumina). We aimed to obtain 75,000 and 200,000 sequencing reads per cell for the v2 and v3 libraries, respectively, to match the different performances of the Chromium Single Cell 3′ Reagent v2 and v3 Kits and to achieve sufficient sequencing saturation. Across all 39 libraries we obtained an average of 187,242 reads per cell. Reads were aligned to the human reference genome GRCh38-3.0.0 and libraries were demultiplexed and aligned with the 10X Genomics pipeline CellRanger (version 3.0.2). Loom files were generated for each sample by running Velocyto (0.17.17) (ref. [76]) to map molecules to unspliced and spliced transcripts.

### Bioinformatic analysis for scRNA-seq

All *.loom files were imported to R as 'Seurat objects', using the 'connect' function of the loomR package and the 'as.Seurat' function of SeuratDisk for *.loom files >3.0.0 (refs. [77],[78]). The counts were obtained using the 'ReadVelocity' function of SeuratWrappers package and we created objects with 'merged', 'spliced', 'unspliced' and 'ambiguous' counts.

The scRNA-seq datasets from the same donor that were sequenced in the same sequencing run were merged to create donor-specific objects. The only exception was the cells of donor 17 that were analysed as two individual datasets because 10 × 256 was sequenced after 10 × 253, but we identified no 'batch effect' separating its cells from the others of the same donor ('10 × 253' and '10 × 256' in Viewer).

The individual donor datasets were analysed separately using Seurat package in R, to inspect their quality. Firstly, we removed the cells with low and high number of detected genes, based on their histogram distribution (likely cell fragments and multiplets, respectively). Next, we ran the DoubletFinder package[79] to identify and remove possibly cell multiplets, considering that 4% of the analysed cells are multiplets.

To integrate the resulting datasets of 163,000 cells, we used the SCTranform function in Seurat, with 5,000 variable genes. We used 5,000 integration features for the dataset integration, setting as reference dataset the donor 17 that corresponds to the oldest timepoint of our analysis (14 PCW). We observed no profound clustering of the cells according to the examined technical covariates, like the utilized 10X Genomics chemistry or the donor identity, especially for those of the same age (Viewer).

The principal component analysis (PCA) was based on the first 100 top principal components (PCs). For definition of the neighbourhood graph and the clusters, we used the default settings of 'FindNeighbors' and 'FindClusters' functions of Seurat[77],[78], with 100 PCs. For identification of cluster selective markers, we used the 'FindAllMarkers' function[77],[78], with MAST[80] statistical test and maximum cell number/cluster set to 126, which corresponds to the smallest suggested cluster. To accept a gene as a cluster marker, it had to be expressed in at least 25% of the cells in the cluster, have 0.1 logarithmic fold increase and be expressed in at least 10% more cells in the cluster than the remaining dataset. We also selected the statistically significant markers (adjusted $P$ value <0.001, after Bonferroni correction) for all downstream analyses.

For the analysis of (1) epithelial, (2) endothelial and (3) immune cells, we selected the corresponding clusters of the 163,000 cell dataset and harmonized the cells according to the donor parameter, using the 'PrepSCTIntegration' function in Seurat with default settings and 5,000 features (genes) and regressing out stress-related genes ('AddModuleScore' function in Seurat)[81],[82], that have been previously shown to get induced by enzymatic tissue dissociation at 37 °C (ref. [83]). Because of the large size of mesenchymal cell subset (>138,000 cells), we used donor 17 as a reference dataset for the harmonization of the different donor datasets. Especially for the analysis of the neuronal cells, we selected the donor datasets with more than 29 cells, that facilitated their decent integration (5 PCW: 49 cells, 5.5 PCW: 187 cells, 6 PCW: 169 cells, 7 PCW: 227 cells, 8 PCW: 38 cells, 8.5 PCW: 52 cells and 14 PCW: 30 cells). The selected 752 cells were further processed as all other categories.

For dimension reduction and clustering of the above main cell-type categories, we applied the same approach as with whole dataset but with the first 50 PCs.

To further filter the cells for possible multiplets, we firstly normalized the counts to 10,000 and then we removed possible red-blood contaminants, setting expression of HBA1 <4, when necessary. For each of the epithelial, endothelial and immune datasets, we detected a cluster that expressed mesenchymal cell markers. Taking into account that

(1) mesenchymal cell number is 12 times larger than epithelial, 21 times larger than endothelial and 33 times larger than immune cell number and (2) it is unlikely for immune cells to express mesenchymal cells markers, we considered these clusters doublets and removed them.

For trajectory inference analysis of complex multicellular developmental tissue architecture, we guided our analysis towards understanding key lineage branching points inspired by the graph abstraction concept. We used the cell–cell unweighted shared nearest neighbour graph ($G \in \{0,1\} cDaN \times N$) and their assigned one-hot clusters ($O \in \{0,1\}$ $N \times k$) to compute for each cluster $k$ the number of edges shared with all clusters ($E \in \Re k \times k$), including itself.

$$E = (GO) TO$$

The number of cluster shared edges was then element-wise normalized by its total number of edges (Hadamard division), resulting in transition probabilities ($P \in [0,1] k \times k$) that range between 0 and 1 for each cluster, representing the proportion of connections shared between each cluster, where $J \in \{1\} k \times k$ is a square all-ones matrix.

$$P = E \oslash (E \cdot J)$$

Spurious weak connections with transition probabilities below $10^{-4}$ were filtered out by setting its value to zero. Edges were then projected onto the cluster centroids on the UMAP embedding for visualization. Cluster transition probabilities on existing edges ($p\,ij > 0$) were converted to graph weights ($w\,ij$) defined by the inverse of transition probabilities:

$$w\,ij = 1/(p\,ij)$$

and optimal paths from immature (that is, root) to mature cell states were calculated using Dijkstra's shortest path algorithm implemented in the igraph package[84]. The indicated clusters, for distinct trajectories, were selected and re-analysed to create a new UMAP plot with 'RunUMAP' function in Seurat[77,78]. The Slingshot package was used for pseudotime analysis. Firstly, we set the root and the end-point clusters with 'getLineages' function, and then we calculated the principal curves ('getCurves' function), the pseudotime estimates ('slingPseudotime' function) and the lineage assignment weights ('slingCurveWeights' function). To identify differentially expressed genes along the trajectories, we used the 'fitGAM' function of tradeSeq. 'patternTest' was used for the analyses of two trajectories and the 'associationTest' function for the differential expression analysis along one trajectory. The differentially expressed genes were ordered on the basis of the hierarchical clustering ward.D2 method, using 'hclust' function in fastcluster package[85] and plotted using a custom script. The 'clusterboot' function of fpc package[86] was used to calculate stability values of gene modules. For the RNA-velocity analyses, we transformed the Seurat objects to *.h5ad with SeuratWrappers and used scVelo pipeline, filtering for 50 'shared counts' and 5,000 'top genes'. As described in the pipeline, the analyses used the packages scvelo, cellrank[87] loompy, matplotlib[88], numpy[89], pandas[90] and scanpy[91].

For the analyses of aberrant basaloid[4] gene expression programmes in the scRNA-seq dataset, we used the 'AddModuleScore' function in Seurat[77,78] to calculate the aggregated gene-expression scores of their characteristic markers, as they have been defined in the corresponding studies.

For the identification of TFs and co-factors, between the differentially expressed genes, we used the AnimalTFDB 3.0 database[92]. The Human Protein Atlas was used for screening of secreted and surface (CD) proteins[93], and Neuropedia database was used to find differentially expressed neuropeptides[94]. Statistically significant (adjusted $P$ value <0.001, average logarithmic fold change >0.25) genes were used in Toppgene suite[95], for GO analyses, with default settings. Their

$P$ values were calculated according to the hypergeometric probability mass function, and the top-ten biological processes were plotted with GraphPad Prism 9 (GraphPad Software, LLC).

## ST

The capture areas of Visium arrays contain 55-µm-diameter spots, with barcoded oligo-dT anchors (unique for each spot) that allow hybridization of the mRNA molecules in a tissue section that are released through its digestion. The anchors are used as primers to facilitate cDNA synthesis and the produced libraries are sequenced. The unique barcodes for each spot allow the spatial resolution of the detected mRNA-species back the tissue, using the spot coordinates.

### ST library preparation
Spatial gene expression libraries ($n = 9$) (6–13 PCW) were generated with the Visium Spatial Gene Expression Slide & Reagent kit (PN-1000184;10X Genomics), according to manufacturer's protocol. Before the analyses, RNA integrity numbers (RIN) were obtained for all samples to assess the quality of the RNA.

Depending on the size of each section, one or more sections of the same sample were placed in each capture area (6.5 × 6.5 mm) of the Visium arrays. The sections were first fixed for 10 min in acetone, stained with Mayer's H&E Y and imaged with a Zeiss Imager.Z2 Microscope (Carl Zeiss Microscopy GmbH), using the Metafer5 software MetaSystems Hard & Software GmbH). Depending on the age of the lung, the tissue sections were permeabilized for 8–20 min to capture the mRNA molecules. The optimal fixative and permeabilization time for developing lung samples was determined before the Visium experiments using a Visium Spatial Tissue Optimization Slide & Reagent Kit (PN1000193;10X Genomics). The cDNA synthesis and library preparation were done according to manufacturer's protocol (PN-1000184 and PN-1000215;10X Genomics). Sufficient amount of 2–4 nM concentration libraries was used for sequencing for Illumina platform, following the manufacturer's instructions.

### ST data analysis
Sequenced ST libraries were processed using Space Ranger 1.0.0 Pipeline (10X Genomics). Reads were aligned to the human reference genome to obtain an expression matrix. The count matrix was filtered for all mitochondrial, ribosomal and non-coding genes. Spots with fewer than 300 unique molecular identifier (UMIs), fewer than 100 genes and genes detected in fewer than five spots were excluded from the analysis. After filtering, a total of 18,125 features were retained for final analysis across 66,626 spots (6 PCW: 1,439, 7 PCW: 2,692, 8 PCW: 1,840, 8.5 PCW: 1,882, 9 PCW: 3,284, 10 PCW: 11,720, 11 PCW: 15,534, 12 PCW: 13,287 and 13 PCW: 14,948).

Normalization and dimension reduction were performed using the Seurat and STUtility packages (version 0.1.0, https://ludvigla. github.io/STUtility_web_site/Installation.html). Technical variability across samples was reduced with RunSCT and RunHarmony (version 1.0, https://github.com/immunogenomics/harmony) functions. PCA was used to select the most important components and a total of 30 principal components were used in downstream analyses, in all cases.

### Integration of scRNA-Seq and ST data
For the integration between scRNA-seq and Visium data, we used the Python package stereoscope (v.03). This method uses scRNA-seq data to characterize the expression profile of each cluster and then find the combination of the clusters that best explains the detected gene mRNAs in every ST spot, using a probabilistic model. Thus, it produces a matrix with ST spots as rows and percentages of each cluster as columns.

Raw counts from the scRNA-seq and Visium data were used as input, along with the scRNA-seq cluster labels. For the scRNA-seq data from each donor, we used the top 5,000 most variable genes as

input, obtained by the 'VariableFeatures' function in Seurat[77,78]. Stereoscope was run with 25,000 epochs with default parameters (more details in the 'README' file in package github page). For the integrated scRNA-seq, that is, all age groups, the entire set of scRNA-seq was used as input to each Visium sample individually and stereoscope was run with 20,000 epochs. For visualization, the output matrix was imported into R and the stereoscope proportion values for each ST spot were plotted as features with the STUtility R package (v.1.0) (ref. 96).

### Interactome analyses of spatially related cell identities
For the definition of cell neighbourhoods, that include cell identities being consistently found with high percentage in the same ST spots, we used the stereoscope data and performed Pearson correlation analysis comparing the frequencies of the different cell types in the analysed ST spots, across all samples and timepoints. We further proceeded with the pairwise connections, that had Pearson's $r$ higher than 0.04. The interactome analyses were based on (1) CellChat because of its ability to identify cell communications based on the interactions between ligands, receptors and co-factors and (2) Nichenet, which predicts cell communications by estimating ligand–target links, based on their expression levels in the interrogated cells, to identify signalling pathways that facilitate cell communications. We initially kept the genes with average gene expression >0.3 $\log_2$(normalized UMI counts + 1) in any of the analysed clusters and then used default settings for the downstream analyses. To analyse the predicted target genes of specific ligands, we used the ligand–target score matrix of NicheNet and selected the same genes as for CellChat, applying an extra filter by keeping the expressed genes in at least 25% of any of the clusters and have 10% increase in the number of positive cells and in the logarithmic fold change. Then, we used Seurat to plot the top-predicted genes, using 'Dotplot' function. The ligand and the identified by CellChat receptors were also included at the beginning of the plot.

### HybISS
ISS is a targeted method for detecting RNA species on tissue sections[97,98]. It utilizes padlock probes that upon specific hybridization to the targeted RNA molecule and enzymatically ligated to become circular. Rolling cycle amplification (RCA) is used to produce large DNA molecules of hundreds of complementary repeats of the padlock probe, that provides high signal-to-noise ratios. Multiplexing is achieved with a four-digit barcode approach that decodes distinct combinations of fluorescence of a given RCA product to the initial targeted RNA species, allowing for spatial expression analysis of several tenths of different genes.

### Gene panel selection
The HybISS gene panel was selected on the basis of two independent criteria: gene potential to be markers of the different identified populations and their role in different key signalling pathways. To select the minimum amount of marker genes needed to uncover the cell type of every cell in the analysed samples, an initial list of candidate marker genes was generated by selecting the top four markers of the main clusters found when analysing individually four samples from different timepoints (5 PCW, 8.5 PCW, 13 PCW and 14 PCW), based on their δpct (difference in the percentage of positives in the cluster against all other cells). This list was curated by assessing the importance of every gene in accurately predicting the different cell types (https://github.com/Moldia/Tools/tree/master/Gene_selection). For this, ISS datasets were simulated by randomly distributing cells in a bidimensional space, assigning a cell type to each cell and simulating the expression of each gene by sampling in a negative binomial distribution with $r$ being the mean expression of a certain gene in a certain cell type. Then, probabilistic cell typing by ISS (pciSeq) was used to assess the cell type of each simulated cell, obtaining the contribution of each gene to predict correctly each cell type. Top-five genes contributing to correctly

predict each cell type were kept, and further simulations were run, obtaining a final list of 72 genes that were able to predict correctly all the cell types on simulated datasets. For the pathway gene selection, we interrogated the above four scRNA-seq datasets for the expression of WNT, SHH, NOTCH and RTK pathway components, such as ligands, receptors, transducers, inhibitors and targets. We further proceeded with those that showed non-ubiquitous expression patterns. The final gene panel of 147 markers was sent to CARTANA with accompanying customized ID sequences for in-house HybISS chemistry detection.

### HybISS mRNA detection
The HybISS experiments were performed by the ISS facility at Science for Life Laboratories (SciLifeLab) following the manufacturer's instructions of CARTANA's High-Sensitivity library preparation kit, using customized backbones, as described in ref. 97 (probe sequences are provided in Supplementary Table 1 (28–30)). After fixation, the tissue sections were overnight incubated with the probe mix, in a hybridization buffer, followed by stringent washing. Then, they were incubated with ligation mix. After washes, RCA was performed overnight. Finally, labelling for detection was performed as described in <protocols.io> (https://doi.org/10.17504/protocols.io.xy4fpyw). Twelve detection cycles were performed on each sample to avoid optical crowding. Therefore, detected genes were divided in three groups, and their four cycle-based barcode was detected in either detection cycles 1–4, 5–8 or 9–12.

### Imaging of HybISS detection cycles
Imaging was performed using a Zeiss Axio Imager.Z2 epifluorescence microscope (Carl Zeiss Microscopy, GmbH), with a Zeiss Plan-Apochromat 20×/0.8 objective (Carl Zeiss Microscopy, GmbH, 420650-9901) and an automatic multi-slide stage (PILine, M-686K011) to allow re-call of coordinates for the regions of interest, facilitating repetitive cycle imaging. The system was equipped with a Lumencor SPECTRA X light engine LED source (Lumencor), having the 395/25, 438/29, 470/24, 555/28, 635/22 and 730/40 filter paddles. The filters, for wavelength separation, included the quad band Chroma 89402 (DAPI, Cy3, Cy5), the quad band Chroma 89403 (AlexaFluor750) and the single band Zeiss 38HE (AlexaFluor488). Images were obtained with an ORCA-Flash4.0 LT Plus sCMOS camera (2,048 × 2,048, 16-bit, Hamamatsu Photonics K. K.).

### HybISS image processing
Imaging data were processed with an in-house pipeline based on MATLAB (https://github.com/Moldia/iss_starfish). Maximum intensity projection was performed on each field of view to obtain a two-dimensional representation of each tile. Then, stitching of tiles was performed using a MATLAB implementation of MIST algorithm, obtaining, after exporting, different *.tiff images corresponding to each channel and round. Then, data were retiled and formatted to fit the Starfish required input. As genes can be either detected in 1–4, 5–8 or 9–12 detection cycles, each group was then decoded independently. Using Starfish tools, individual tiles were registered across cycles and a top hat filter was applied on each channel to get rid of the background noise. Channel intensities were also normalized, and spots were detected. Finally, decoding was performed on each tile using MetricDistance, obtaining the identity of all the detected RCA products.

### HybISS data analysis
Two different yet complementary strategies were followed to characterize the cellular heterogeneity within the ISS datasets. Probabilistic cell typing for in situ sequencing (PciSeq) was performed to identify the identity of every cell in the tissue. For this, cells were segmented on the basis of DAPI using a watershed segmentation, and reads were assigned to cells as described in ref. 71. In addition, Tangram was used to couple the scRNA-seq with the HybISS datasets, functioning similarly to stereoscope. Gene expression imputation was performed as

described in ref. 99. In 5 PCW sections, where nuclear segmentation was not possible, hexagonal binning was used to segment the tissue. In this case, the expression of each hexagonal bin was used as input for probabilistic cell typing and Tangram.

## SCRINSHOT

SCRINSHOT is also a targeted method of RNA-species in situ detection that utilizes padlock probes for signal amplification, similarly to ISS. Its major difference is the usage of SplintR-ligase for padlock probe circularization and the simplest detection approach that assigns a fluorophore to a distinct gene, in each detection cycle. The different chemistry and the omission of decoding results in better sensitivity than ISS. However, it has reduced multiplexity (three to five genes per detection cycle), being more laborious than ISS.

### Gene selection, padlock probe design and mRNA detection

For spatial analysis of the two identified NE-cell identities, we used the highly expressed *GRP* and *GHRL*, for easy identification of epi cl-12 and epi cl-11, respectively. Then, we selected markers that are expressed in intermediate and low levels, focusing mainly on TFs, such as *ASCL1*, *RFX6*, *NKX2-2*, *ARX* and *PROX1*. Markers such as *SCGB3A2*, *FOXJ1* and *TP63* were used to identify the non-NE cells. The *SCGB1A1*, *SFTPC*, *ETV5*, *FOXJ1*, *AGER*, *SOX2* and *SOX9* padlock probes were designed as in SCRINSHOT original publication. For the rest, a unique barcode was inserted in the backbone of all probes that recognize the same mRNA, that allowed their detection by only one detection oligo, reducing substantially the cost (all sequences are found in Supplementary Table 1 (31)). All the reactions were done according to the original SCRINSHOT protocol, except for an increase of the detection-oligo hybridization temperature to 30 °C.

### Imaging of SCRINSHOT signals on tissue sections

For signal acquisition we did 13 detection cycles, using a Zeiss Axio Observer Z.2 fluorescent microscope (Carl Zeiss Microscopy, GmbH) with a Colibri 7 LED light source (Carl Zeiss Microscopy, GmbH, 423052-9770-000), equipped with a Zeiss 20×/0.75 Plan-Apochromat, a Zeiss AxioCam 506 Mono digital camera and an automated stage, that allowed imaging of the same regions in every cycle. For signal detection, we used the following Chroma filters: DAPI (49000), FITC (49003), Cy3 (49304), Cy5 (49307), Texas Red (49310) and Atto740 (49007).

### SCRINSHOT image analysis

The nuclear staining was used to align the images of the same areas between the hybridizations, using Zen2.5 (Carl Zeiss Microscopy GmbH). The images were analysed as 16-bit *.tiff files, without compression or scaling. Images were tiled using a custom script in Fiji[100,101]. The signal dots were counted using Cell-Profiler 4.13 (ref. [102]), Fiji[100,101] and R-RStudio[103–107] custom scripts. The identified signal-dot coordinates were used to project the signals on DAPI images, using TisUUmaps[108].

For the analysis of the 11.5 PCW SCRINSHOT dataset, nuclei images were segmented into hexagonal bins of 7 μm radius. Only bins with a clear proximal epithelial component (*SOX2* dots >3, *EPCAM* dots >3) were further processed. To maintain NE-related bins, we used the analysed genes that were specifically expressed in NE cells according to scRNA-seq (*ARX*, *NKX2-2*, *GHRL*, *ACSL1*, *CALCA*, *GRP*, *RFX6*, *CFC1*, *PCSK1* and *ASCL1*). Bins with a presence of at least 12 signals of the above genes were further processed. We also kept bins containing more than ten ASCL1 dots, which was found to be expressed by NE progenitors. We created AnnData objects with the counts for each gene in every bin, in addition to the bin coordinates. We used Scanpy to perform Leiden clustering with 0.1 resolution and represented those clusters using UMAP plots. We further assessed the correlation in expression between the different NE genes and represented the Pearson's correlation results as heat map. Finally, the suggested clusters were annotated on the basis of the combination of different NE markers, according to the scRNA-seq data.

### Exploration of the zonation patterns in the developing lung using ISS

To calculate the relative position of distinct cell types in the proximal–distal and radial axis, analysed tissues with HybISS were segmented into bins (radius 20 μm). Only bins with more than three detected EPCAM mRNAs were considered to be airway related. We calculated the distance of each bin in the tissue to the closest identified airway-related bin, defining the first axis explored (radial axis considering the airway as the centre). Cells with a radial distance higher than 140 μm were excluded from the analysis. To define the second axis, we explored the diversity within airway-related bins and, by UMAP-dimension reduction, we identified that the first dimension recapitulated the proximal–distal typical patterning, based on the expression of known markers. We used that value as pseudotime to assign a proximal–distal value to each of the detected bins. These values served as the second axis of the analysis, considering the proximal–distal value of the closest epithelial bin as the proximal–distal value of the analysed mesenchymal cells. The distribution of the cells analysed was represented using kernel density estimation (KDE)-based heat maps.

### Exploration of the zonation patterns in the developing lung using ST

To explore the zonation of mesenchymal populations present in the developing lung with ST datasets, we analysed sections from 8.5 PCW. We identified ST spots containing airways by looking at the expression top ten differentially expressed epithelial markers (Extended Data Fig. 2g). Cells containing more than eight UMIs were considered as airway-related ST spots. To define the radial axis, each ST spot was given a value depending on its distance from its closer airway-related ST spot. The proximal–distal axis was calculated on the basis of the compared relative expression levels of known proximal (SOX2 and SCGB3A2) and distal (ETV5 and TPPP3) epithelial markers. On the basis of the relative expression of proximal and distal markers, every epithelial ST spot was given a value between −1 (proximal) and 1 (distal). ST spots that were not airway related were given the proximal–distal score of their closest airway-related ST spot. After rounding the proximal–distal scores of every ST spot, the frequency of every cluster detected using stereoscope was then computed by averaging ST spots with the same proximal–distal and radial coordinates.

### Immunofluorescence

Tissue sections were prepared, using the same protocol as SCRIN-SHOT. Fresh frozen material was fixed with 4% PFA for 10 min at room temperature, and slides were washed three times for 5 min with phosphate-buffered saline (PBS) 1× (pH 7.4). We incubated the sections with 5% donkey serum (Jackson ImmunoResearch, 017-000-121) in PBS 1× (pH 7.4) with 0.1% Triton X100 (blocking buffer) for 1 h at room temperature, and then they were incubated with primary antibodies in blocking buffer overnight at 4 °C. Slides were washed with PBS 1× (pH 7.4) three times for 5 min and incubated with secondary antibodies in 2% donkey serum in PBS 1× (pH 7.4) with 0.1% Triton X100 for 1 h at room temperature. After three washes with PBS 1× (pH 7.4) for 10 min each, nuclei were counterstained with 0.5 μg ml$^{-1}$ DAPI (Biolegend, 422801) in PBS 1× (pH 7.4) in 0.1% Triton X100 and slides were mounted with ProLong Diamond Antifade Mountant (Thermo, P36961).

Sections treated with anti-PHOX2B goat, anti-DLL3 rabbit, anti-COL13A1 rabbit and Cy3 anti-Actin, α-Smooth Muscle (ACTA2) mouse monoclonal antibodies were incubated in TE buffer (10 mM Tris and 1 mM EDTA pH 9.0) for 30 min, at 80 °C in a waterbath and cooled on ice for 30 min to facilitate antigen retrieval and washed three times for 5 min with PBS 1× (pH 7.4), before incubation with the blocking solution. Sections treated with anti-Krt5 chicken and anti-p63a rabbit antibodies were incubated in sodium citrate (10 mM pH 6.0) and processed as above.

## Image acquisition for immunofluorescence

Image acquisition was initially done as in SCRINSHOT, with a 10× lens, allowing the identification of informative regions of interest. For high-resolution images, we used a Zeiss LSM800 confocal microscope, equipped with a Plan-Apochromat 40×/1.30 oil lens or a Zeiss LSM780 confocal microscope, equipped with a Plan-Apochromat 63×/1.40 oil DIC M27 objective. Optimal resolution settings were used and images were acquired as optical stacks. For imaging of the ACSL1-CGRP-CDH1 stainings, we used a Leica DMI8 microscope (Leica Microsystems, 11090148013000), with a SOLA light engine light source (Lumencor,16740), equipped with a 40×/0.80 HC Fluotar, a Hamamatsu camera (2,048 × 2,048, 16-bit, C13440-20C-CL-301201) and an automated stage (ITK Hydra XY). For the signal detection, we used the following Chroma filters: QUAD-S filter set: DFTC (DC: 425; 505; 575; 660). Imaging was done via the LASX software (Leica Microsystems), and images were analysed with Fiji[100,101].

## Browser-based interactive visualization of the scRNA-seq, spatial and interactome analyses

For the browser-based representation of our data, we used the TissUUmaps tool[109]. In the presented version, we have modified TissUUmaps for accelerated GPU-based rendering, enabling real-time interactive multiscale viewing of millions of data points directly via a web browser. Furthermore, we have added functionality so that ST data and single-cell pciSeq data from ISS can be presented as pie charts for efficient viewing of spatial heterogeneity. TissUUmaps supports FAIR sharing of data by allowing users to select regions of interest and directly download raw data in a flexible *.csv format, enabling further exploration and analysis, of all datasets. We based the interactome browser in the Cell Chat shiny app, described in ref. 10.

## Statistics and reproducibility

No statistical method was used to pre-determine sample size. No data were excluded from the analyses. The experiments were not randomized, and the investigators were not blinded to allocation during experiments and outcome assessment. For differential expression analyses of scRNA-seq datasets, MAST package was used in Seurat, and when it is mentioned in figure legends, the results were filtered according to the adjusted *P* value that was based on Bonferroni correction using all features in the datasets.

For scRNA-seq experiments, we analysed one 5 PCW lung, one 5.5 PCW lung, two 6 PCW lungs, two 7 PCW lungs (twins), one 8 PCW lung, two 8.5 PCW lung, one 10 PCW lung, two 11.5 PCW lungs, two 12 PCW lungs, two 13 PCW lung and one 14 PCW lung. All attempts at replication with the provided scripts were successful.

For ST experiments, we analysed four sections of 6 PCW lungs, (Figs. 1b, 2b,d and 3c and Extended Data Fig. 4c), eight sections of 7 PCW lungs (Fig. 3d), four sections of 8–8.5 PCW lungs (Figs. 2b and 6a and Extended Data Fig. 4c) and four sections of 11.5 lungs (Figs. 2b and 3e and Extended Data Fig. 4c). Sections of each stage were processed in at least two independent experiments with similar results.

For HybISS experiments, we analysed three sections of 5.5 PCW lungs, (Extended Data Figs. 4d, 6g and 8b), two sections of 6 PCW lungs (Figs. 1e, 2e,f and 4g and Extended Data Fig. 2h) and two sections of 13 PCW lungs (Figs. 6c and 7a and Extended Data Figs. 6g and 8b). Sections of each stage were processed in two independent experiments with similar results.

For SCRINSHOT experiments, we analysed one section of a 6 PCW lung, one section of an 8.5 PCW lung, one section of an 11 PCW lung (Extended Data Fig. 10g) and one section of a 14 PCW lung (Fig. 4c and Extended Data Fig. 10d). The sections were processed in two independent experiments, showing similar distal tip (>500 cases) and NE cell patterns (>100 cases).

For LUM COL13A1 ACTA2 immunofluorescence, we analysed four 8.5 PCW lung sections and one 12 PCW lung section in two experiments.

More than ten patterns similar to those shown in Fig. 2g were found in each section. For ACTA2 Ecad MKI67 immunofluorescence, we analysed three 8.5 PCW, two 12 PCW and one 14 PCW lung sections, in two independent experiments with similar results. Extended Data Fig. 4e contains representative images of large airways (8.5 PCW: >20, 12 PCW: >40 and 14 PCW: >50), of airway stalks with tips (8.5 PCW: >20, 12 PCW: >50 and 14 PCW: >50) and of distal tips (8.5 PCW: >20, 12 PCW: >50 and 14 PCW: >50). For the DLL3 NF-M PHOX2B stainings in Fig. 3f–h, we stained three 8.5 PCW and one 12 PCW lung sections in two independent experiments. One 8.5 PCW and one 12 PCW lung sections were independently processed for H&E staining. In both stainings, the different tissues gave similar results. For the SOX10 ASCL1 ISL1 immunofluorescence (Extended Data Fig. 7e), we analysed two 8.5 PCW, two 12 PCW and one 14 PCW lung sections, in two independent experiments, with similar results. For the KRT17 Ecad immunofluorescence (Fig. 4d), we stained two 12 PCW and one 14 PCW in two independent experiments with similar results. For TP63 KRT5 Ecad immunofluorescence, we stained two 8.5 PCW and two 14 PCW lung sections in two independent experiments with similar results (Extended Data Fig. 8g). For the SST SSTR2 GHRL staining, we analysed four 8.5 PCW and one 12 PCW lung sections, in three independent experiments with similar results. For GRP GHRL immunofluorescence four 8.5 PCW and one 12 PCW lung sections were analysed, in three independent experiments with similar results.

For all spatial methods, we acquired images of whole lung sections. Representative areas of interest were identified, imaged and used in the figures.

## Reporting summary

Further information on research design is available in the Nature Portfolio Reporting Summary linked to this article.

## Data availability

The datasets generated during and/or analysed during the current study are available at GEO (GSE215898), comprising single-cell data (GSE215895) and ST data (GSE215897). The scRNA-seq data can be additionally accessed in https://hdca-sweden.scilifelab.se/tissues-overview/lung/ and https://cells.ucsc.edu/?ds=lung-dev. scRNA-seq datasets of individual donors can be accessed at https://doi.org/10.5281/zenodo.6386452. The used scRNA-seq datasets, containing subsets of the whole dataset and of the mesenchymal cell dataset are available at https://doi.org/10.5281/zenodo.7143999. The raw data of the fluorescence images can be accessed at https://doi.org/10.1101/2022.01.11.475631 and https://doi.org/10.5281/zenodo.6673650. ST raw data can be accessed at https://doi.org/10.5281/zenodo.6661019. scVelo datasets and analysis files can be accessed at https://doi.org/10.5281/zenodo.6673667. Raw-image datasets of HybISS (180 GB) and SCRINSHOT (683 GB) are available from the corresponding authors on reasonable request because of data size limitations.

## Code availability

The scripts for all analyses can be accessed at https://doi.org/10.5281/zenodo.7143091.

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

## Acknowledgements

We thank National Genomics Infrastructure for sequencing services, the Karolinska Institutet Developmental Tissue Bank for providing human prenatal tissue and the ISS facility, at SciLifeLab for ISS service. This work was supported by grants from the Knut and Alice Wallenberg Foundation (KAW 2018.0172), the Erling Persson Foundation, the Chan Zuckerberg Initiative (SVCF 2017-173964), Cancerfonden (MN: CAN 2018/604) and the Swedish Research Council (MN: 2019-01238). A.S., A.F., J.T., A.L. and C.S. were supported by grants from Cancerfonden, the Swedish Research Council and the German Research Foundation (DFG), grant KFO309 (project number 284237345) to C.S.

## Author contributions

E.L., E.S., S.L., J.L., M.N. and C.S. designed the study. A.S., E.B., J.T. and A.L. and X.L. isolated and processed the tissues. L.H. and E.B. performed the scRNA-seq experiments, while A.S., E.B. and J.M. analysed the scRNA-seq datasets generated. A.S. and S.M.S. evaluated and implemented the interactome-related analyses. X.A., Z.A., R.M. and M.A. performed the ST experiments. P.C., M.V., J.B. and S.M.S. analysed ST experiments. A.S., J.T. and A.F. selected and validated the SCRINSHOT probes. J.T. and A.S. performed the SCRINSHOT experiments and analysed the data. B.A., A.M.C. and S.S. optimized antibodies for immunofluorescences. S.S. performed the immunofluorescences. S.M.S., A.S. and A.L. selected the gene panel for ISS experiments. The ISS facility and S.M.S. performed ISS experiments. S.M.S. analysed ISS experiments. C.A. and C.W. implemented the TissUUmaps viewer and data portal. A.S., S.M.S., C.S. and M.N. wrote the manuscript. All authors read the manuscript and suggested improvements on its content and forms.

## Funding

## Competing interests

J.L. and M.N. are advisors to 10X Genomics. All other authors declare no competing interests.

## Additional information

**Extended data** is available for this paper at https://doi.org/10.1038/s41556-022-01064-x.

**Correspondence and requests for materials** should be addressed to Mats Nilsson or Christos Samakovlis.

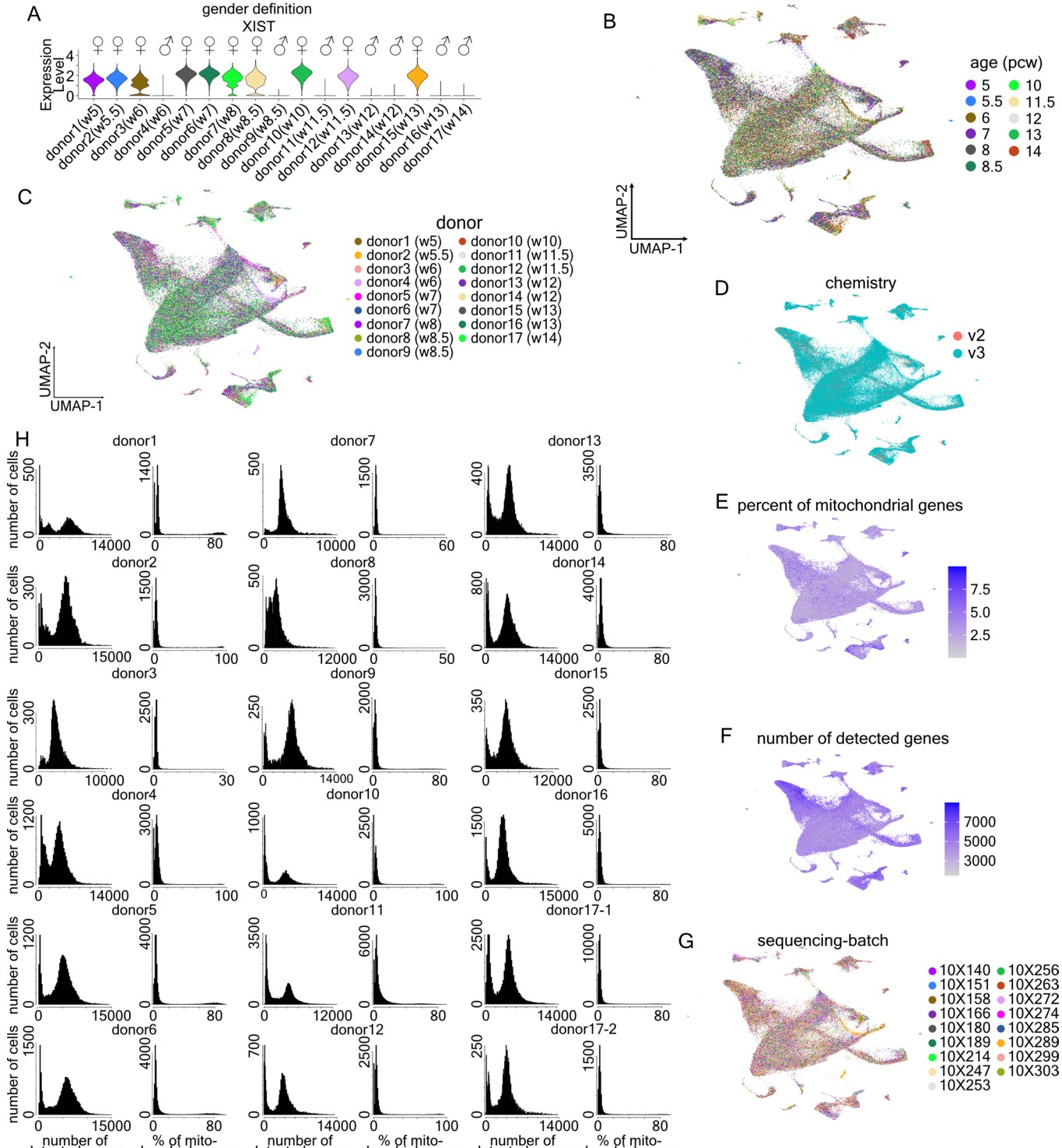

**Extended Data Fig. 1 | Quality controls (QC) of the scRNA-Seq datasets from all analyzed donors. (a)** Violin plot of XIST expression levels for sex determination of the donors. ♀-female: XIST$^{pos}$ and ♂-male: XIST$^{neg}$. Expression levels: $\log_2$(normalized UMI-counts+1) (library size was normalized to 10.000). **(b-g)** UMAP-plots of all cells, labeled according to the (B) age, (C) donor-identity, (D) 10X Chromium version (E) percentage of mitochondrial genes, (F) number of detected genes and (G) sequencing-batch. **(h)** Histograms of detected gene numbers and percent of mitochondrial genes in the analyzed datasets, before application of QC-criteria. Additional QC-information and gene expression levels, in the whole dataset can be accessed at https://hdca-sweden.scilifelab.se/tissues-overview/lung/.

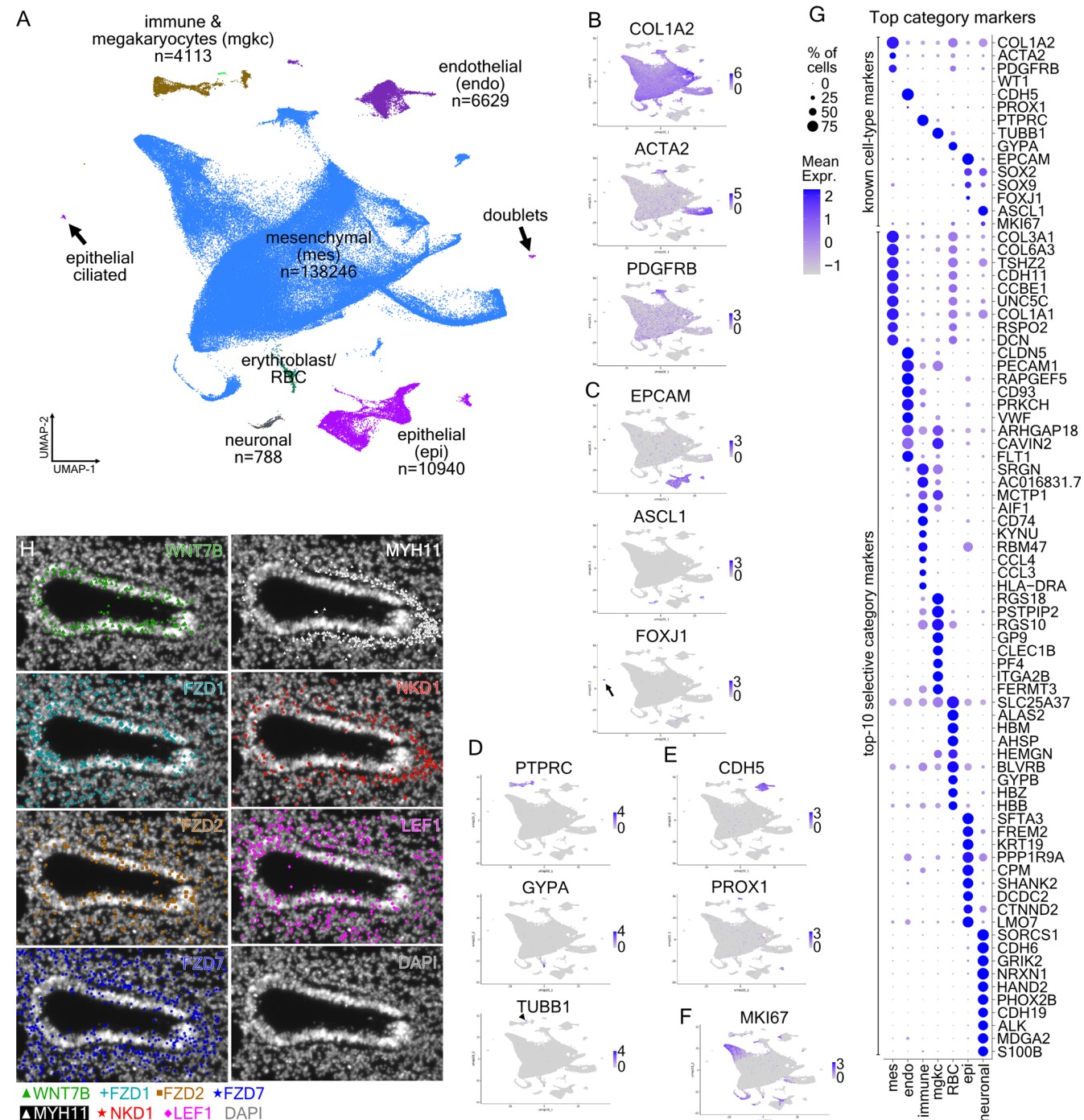

**Extended Data Fig. 2 | Initial scRNA-Seq analysis suggests six main cell categories, with distinct gene-expression profiles. (a)** Whole-dataset UMAP-plot of the 6 main cell categories, from the 17 donors. 'n': number of cells/category. The arrows indicate two clusters of doublets (top) and epithelial ciliated cells (bottom), which have been moved from their original position, in the UMAP-plot and placed in inserts. **(b-f)** UMAP-plots showing the expression of known markers: mesenchymal (*COL1A2*[2], *ACTA2*[2], *PDGFRB*[110]) (B), epithelial (EPCAM, *ASCL1*[111], *FOXJ1*[112]) (C), immune and erythroblasts/erythrocytes (PTPRC[113], GYPA, TUBB1[81]) (D), endothelial (CDH5[82], PROX1[114,115]) (E) and proliferation (MKI67[116]) (F). Expression levels: log$_2$(normalized UMI-counts+1) (library size was normalized to 10.000). Blue: high, Gray: zero. **(g)**

Balloon-plot showing the expression of known cell-type markers together with the top-10 most selective category markers (adjusted p-value < 0.001, MAST, Bonferroni corrected using all features)). The top-20 genes (log2 fold-change) were sorted according to positive cells number in the cluster and the top-10 were plotted. Balloon-size: percent of positive cells in cluster. Color intensity: scaled expression. Blue: high, Gray: low. Gene order follows the cell-category order. **(h)** Single-gene images of the projection in Fig. 1e, showing the mRNAs of *WNT7B*, *FZD1, FZD2, FZD7, LEF1, NKD1 MYH11*, detected by HybISS, Interactive inspection of the data is available through the https://hdca-sweden.scilifelab.se/tissues-overview/lung/.

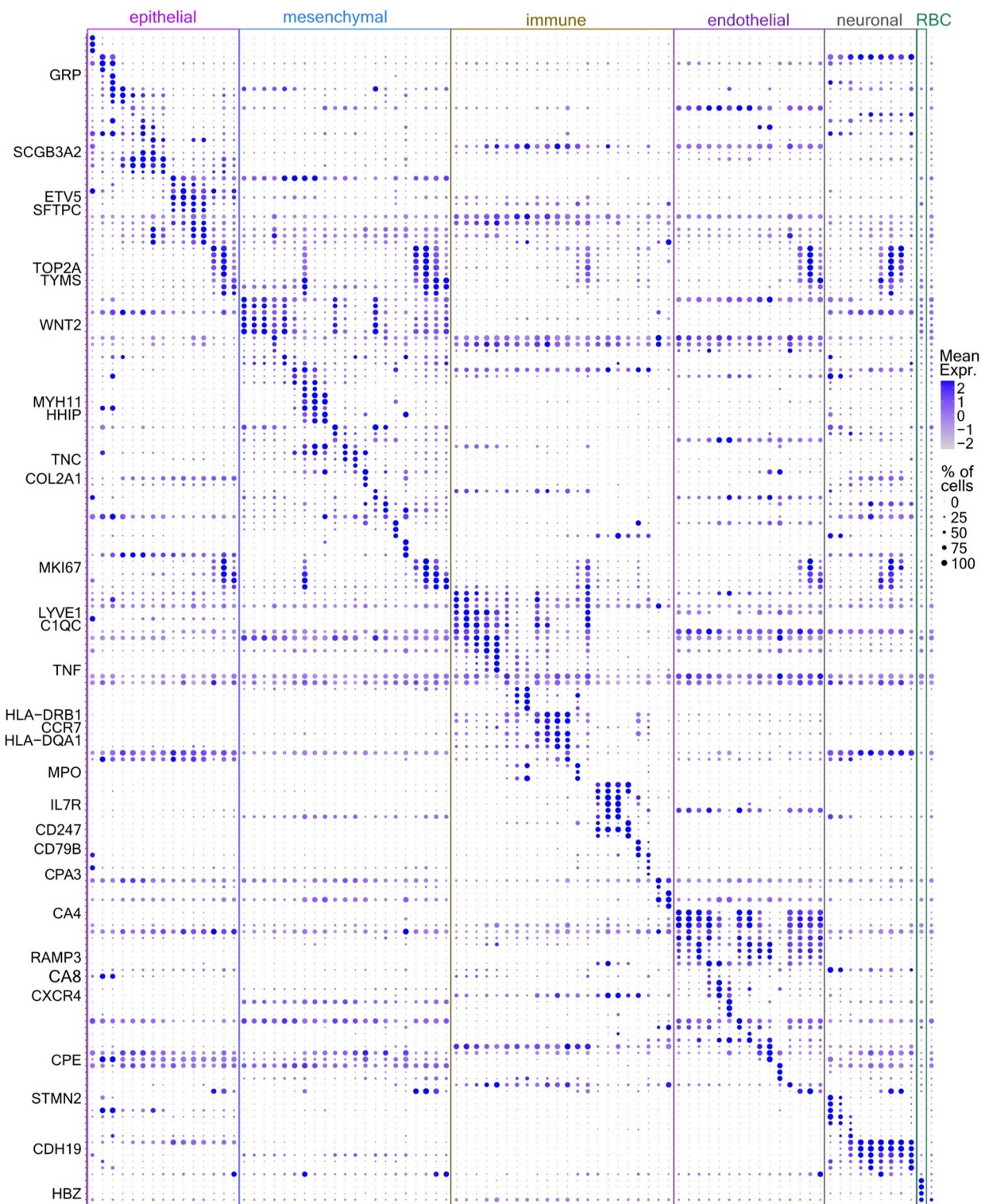

**Extended Data Fig. 3 | Top selective markers of the 83 identified cell states.**
Balloon-plot of the top-3, most selective genes for each of the 83 suggested clusters of the whole dataset that contains all analysed donors. Clusters of same main cell categories were placed together. Colored boxes indicate the main cell categories. Characteristic genes are shown on the left (adjusted p-value < 0.001, MAST Bonferroni corrected using all features), The top-6 genes (log2 fold-change) were sorted according to positive cell numbers in the cluster and the top-3 markers were plotted. Balloon size: percent of positive cells. Color intensity: scaled expression. Blue: high, Gray: low. Gene order follows the cluster order. All genes and clusters of the plot are included in the Supplementary Table 1–14.

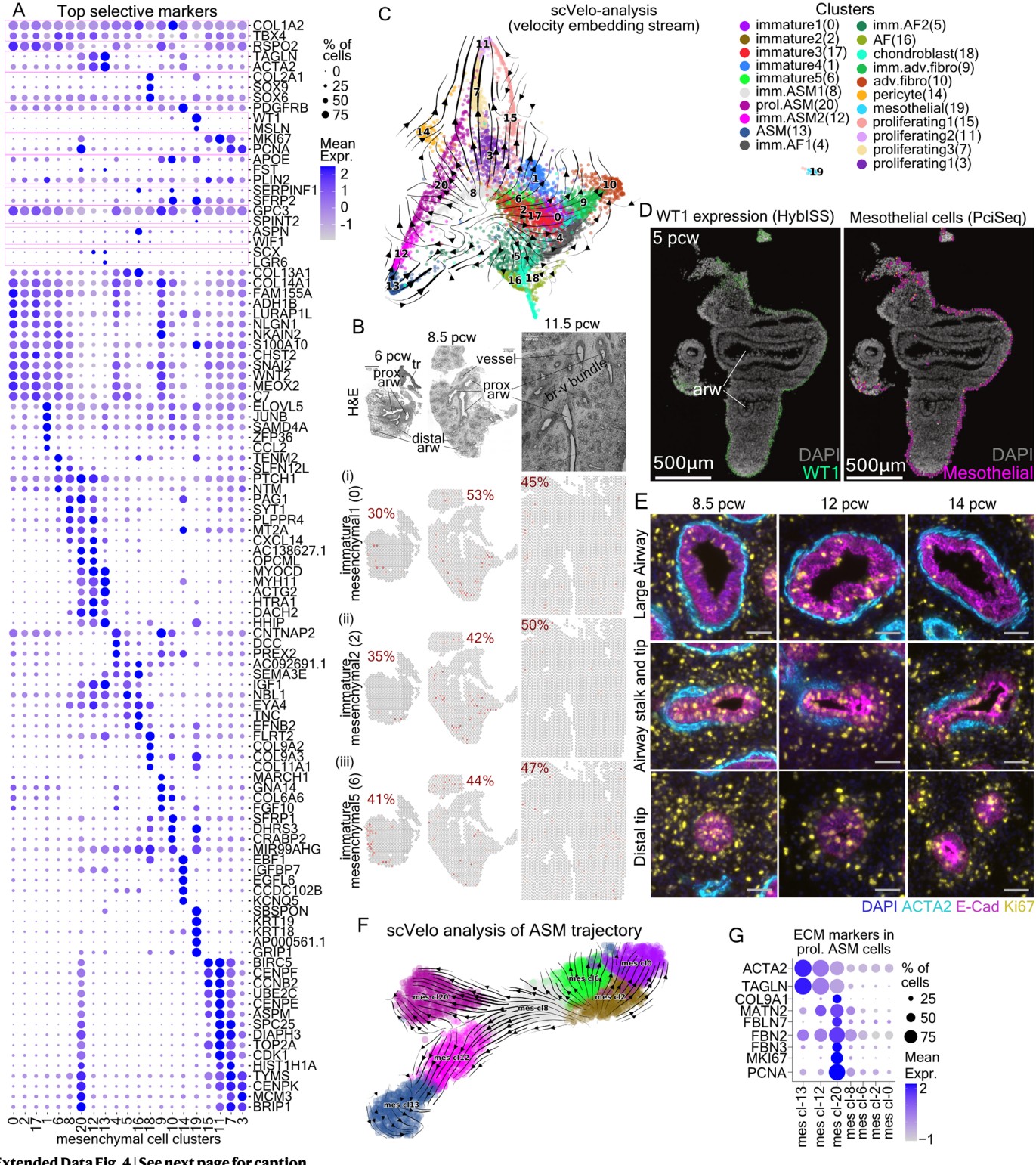

**Extended Data Fig. 4 | See next page for caption.**

**Extended Data Fig. 4 | Analysis of mesenchymal cell heterogeneity. (a)** Balloon-plot of known mesenchymal markers (*COL1A2-COL14A1*), together with the top-5 cluster markers of the mesenchymal dataset (17 donors). General: *COL1A2*[2], *TBX4*[15], immature: *RSPO2*[117], Smooth Muscle (SM): *TAGLN*, *ACTA2*[2], Chondroblast: *COL2A1*, *SOX9*, *SOX6*[118,119], Pericyte: *PDGFRB*[105], Mesothelial: *WT1*[120], *MSLN*[121], Proliferating: *MKI67*[114], *PCNA*[122], Lipofibroblast: *APOE*, *FST*, *PLIN2*[2], Adventitial-fibroblast: *SERPINF1*, *SFRP2*[2], Alveolar-fibroblast: *GPC3*, *SPINT2*[2], Myofibroblast: *ASPN*, *WIF1*[2], Fibromyocyte: *SCX*, *LGR6*[2], COL13A1pos-fibroblast: *COL13A1*[31] and COL14A1pos-fibroblast: *COL14A1*[31]. From the differentially expressed genes (adjusted p-value < 0.001, MAST, Bonferroni corrected), the top-10 (log2 fold-change) were sorted according to proportion of positive cells in the cluster and the top-5 of these were plotted. **(b)** Stereoscope assigned distribution of (i) mesechymal1 (cl-0), (ii) mesenchymal2 (cl-2) and (iii) mesenchymal5 (cl-6) cells in three timepoints. Red numbers: the highest percent of the indicated cell-state. Dark red: high, gray: zero. H&E staining: tissue structure. Scale-bar: 400 µm. **(c)** scVelo-analysis, using a dataset subset (441 cells/cluster) from all donors. Arrow direction: future state, arrow size: transition possibility. **(d)** HybISS analysis of a 5 pcw lung section showing the mesothelial marker *WT1* mRNA expression in tissue periphery[120,121] (top) and the prediction of mesothelial-cell spatial distribution, according to PciSeq (bottom). Representative data in: https://hdca-sweden.scilifelab.se/tissues-overview/lung/ **(e)** Immunofluorescence for α-SMA (cyan, SM), Ecad (magenta, epithelium) and MKI67 (yellow, proliferating cells) on 8.5 (left), 12 (middle) and 14 (right) pcw lungs, in proximal-large (top), stalk (middle) and distal (bottom) airways. Nuclei (blue, DAPI). Scale-bars: 50 µm. **(f)** scVelo-analysis of the proliferation (cl-20) and maturation (cl-12 and −13) airway SM-trajectories. Colors as in 'B'. **(g)** Balloon-plot of *ACTA2* and *TAGLN* (SM), *COL9A1*, *MATN2*, *FBLN7*, *FBN2* and *FBN3* (extracellular matrix) and *MKI67* and *PCNA* (proliferation). In Balloon-plots, size: percent of positives. Color intensity: scaled expression. Blue: high, Gray: low. 'arw': airway, 'prox.': proximal, 'tr': trachea, br-v bundle: bronchovascular bundle.

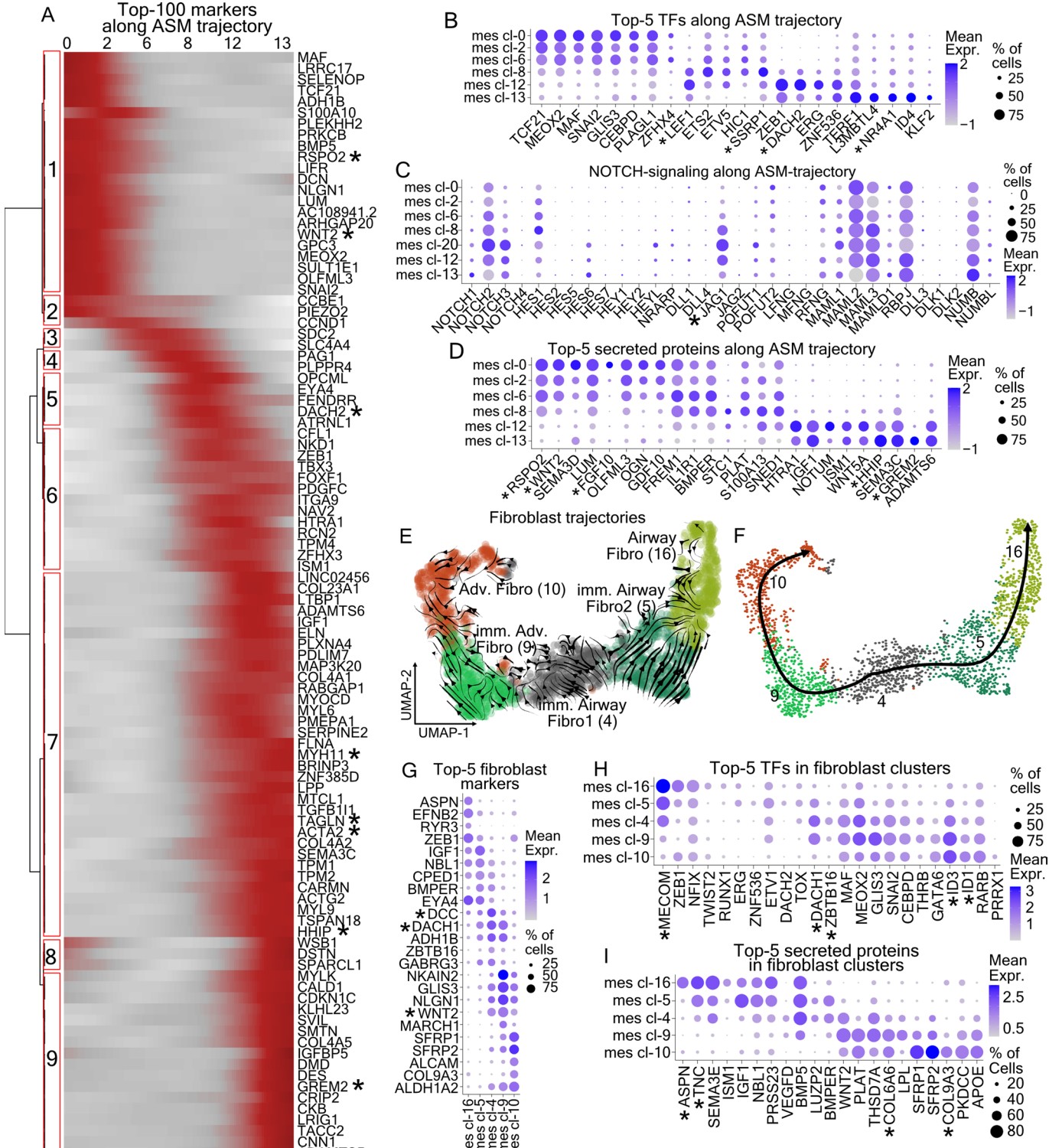

**Extended Data Fig. 5 | Analysis of mesenchymal trajectories. (a)** Heatmap of the top-100 differentially expressed genes along the airway smooth muscle (ASM) maturation trajectory, based on tradeSeq[21]. Numbers: stable gene-modules (Bootstrap values module-1: 0.88, module-2: 0.84, module-3: 0.81, module-4: 0.73, module-5: 0.75, module-6: 0.76, module-7: 0.83, module-8′ 0.62, module-9: 0.87). Color intensity: scaled expression. Dark red: high, Gray: low. **(b–d)** Balloon-plots of the top-5 transcription factors (TFs) (B), NOTCH-signaling components (C) and secreted (D) proteins, identified by differential expression analysis of the indicated clusters, along the ASM maturation-trajectory. **(e)** scVelo-analysis on the mesenchymal fibroblast clusters. Colors as in Fig. 2a. The direction of arrows shows the progression towards more differentiated states. **(f)** UMAP-plot of the mesenchymal fibroblast clusters and pseudotime trajectories,

estimated by Slingshot. Colors as in Fig. 2a. A randomly selected subset of 441 cells/cluster from all donors was used in 'E' and 'F'. **(g–i)** Balloon-plots of the top-5 markers (G), transcription factors (TFs) (H) and secreted proteins (H), identified by differential expression analysis of the indicated clusters. Gene order follows the cluster order. In all Balloon-plots, balloon size: percent of positive cells. Color intensity: scaled expression (B-D) or log₂(normalized UMI-counts+1) (library size was normalized to 10.000) (G-I). Blue: high. Gray: zero. In all Top-5 plots, from the statistically significant genes (adjusted p-value < 0.001, MAST with Bonferroni correction using all features), the top-10 genes (log2 fold-change) were sorted according to the percent of positive cells and the top-5 markers were plotted. Gene order follows the cluster order. The '*' indicate commended genes.

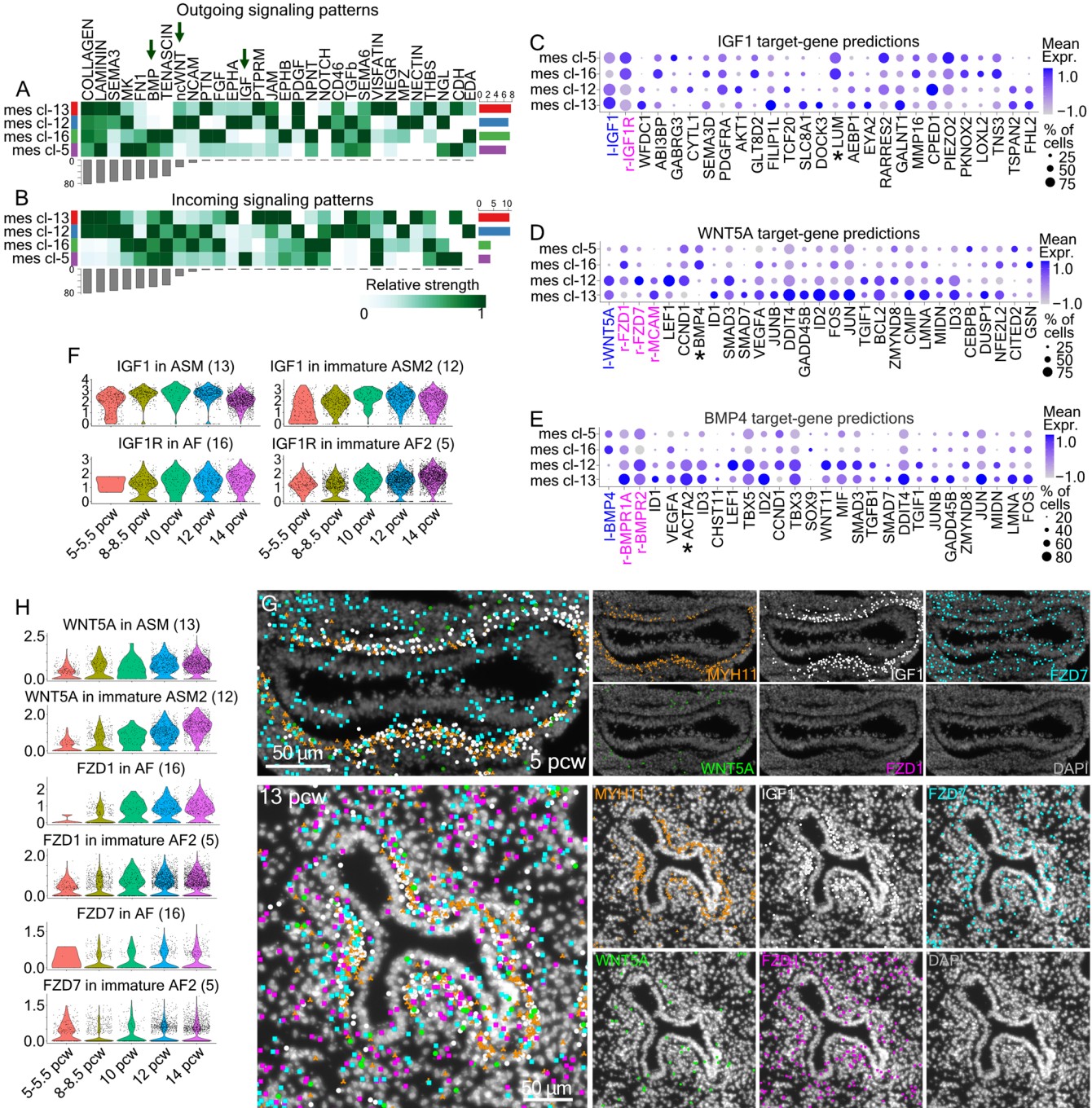

**Extended Data Fig. 6 | Exploration of interactions between mesenchymal cell-types. (a, b)** Heatmaps of CellChat predictions of outgoing (A) and incoming (B) signaling patterns between the analyzed ASM and AFs. Bars represent the outgoing/incoming overall potential of each cluster (top) and pathway (right). Color intensity shows the relative strength of cluster contribution to the communication pattern. Dark green: high, White: low importance. **(c, e)** Balloon-plots of the top-20 NicheNet-predicted *IGF1* (C), *WNT5A* (D) and *BMP4* (E) -target genes, expressed in the ASM and AF clusters. Ligands (l-): blue.

Receptors (r-): magenta. Balloon size: percent of positive cells. Color intensity: scaled expression. Blue: high, Gray: low. **(f)** Violin-plots of the IGF1-ligands and its receptor (IGF1R) in the indicated clusters, at 5–5.5, 8–8.5, 10, 12 and 14 pcw cells. Expression levels: log₂(normalized UMI-counts+1) (library size was normalized to 10.000). **(g)** HybISS spatial validation of IGF1 (white), *WNT5A* (green) and its predicted receptors *FZD1* (magenta) and *FZD7* (cyan) on 5 and 13 pcw lung sections. *MYH11* (orange): airway smooth muscle. DAPI (gray): nuclei. Scale-bars: 50 μm. **(h)** As in 'F' for *WNT5A*, *FZD1* and *FZD7*. The '*' indicate commended genes.

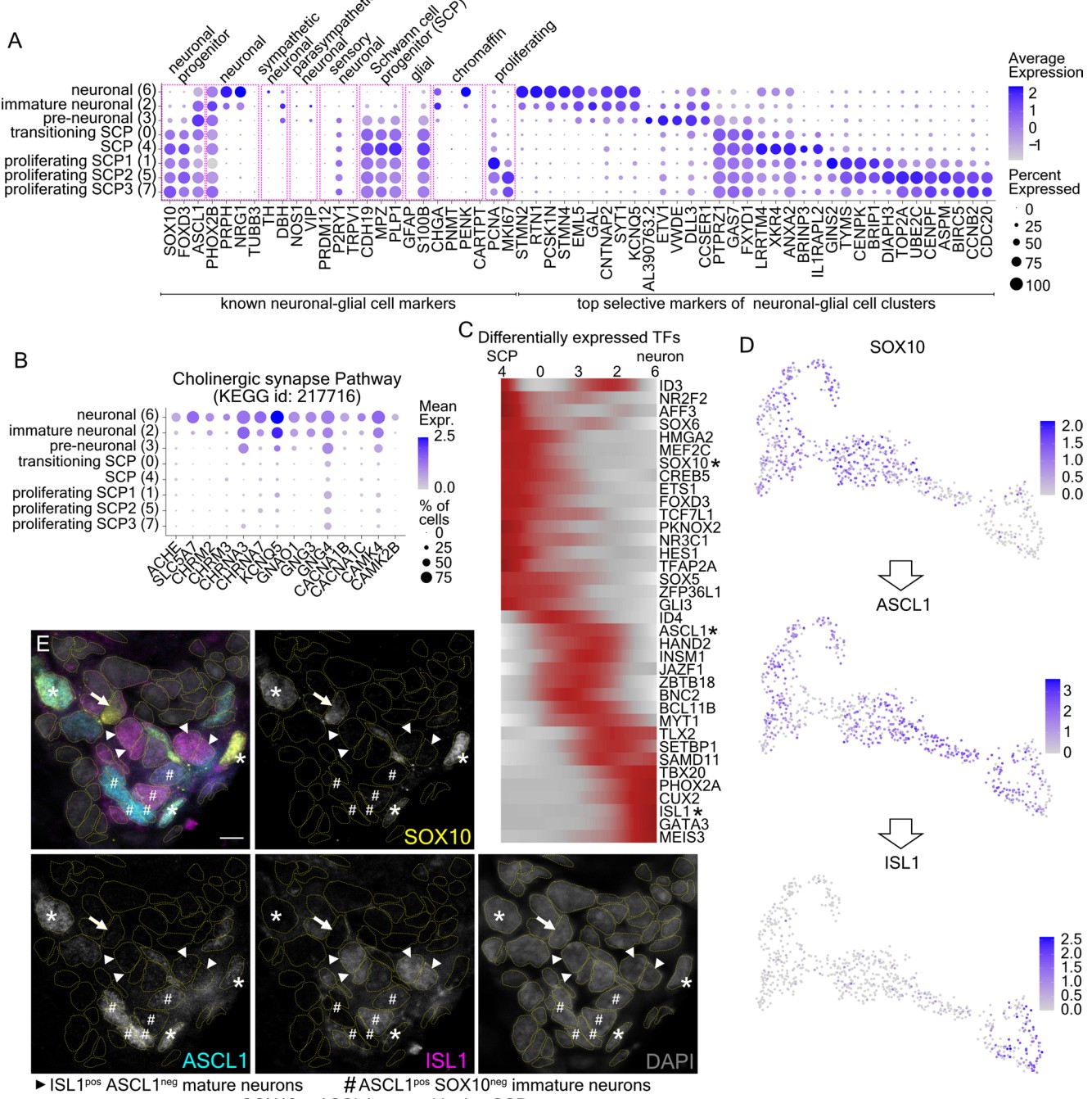

**Extended Data Fig. 7 | Signaling pathways involved in neuronal cell communications. (a)** Balloon-plot of known neuronal and glial cell markers (*SOX10-MKI67*). Progenitor: *SOX10*[123], *FOXD3*[124], *ASCL1*[42], Neuronal: *PHOX2B*[125], *PRPH*[126], *NRG1*[127], *TUBB3*[128], Sympathetic neurons: *DBH*, *TH*[129], Parasympathetic neurons: *NOS1*, *VIP*[130], Sensory neurons: *PRDM12*, *P2RY1*, *TRPV1*[131,132], Schwann Cell Progenitors (SCPs): *CDH19*, *MPZ*, *PLP1*[133], Glial cells: *GFAP*, *S100B*[134,135], Chromaffin cells: *PNMT*, *PENK*, *CARTPT*[136] and Proliferating cells: *MKI67*[114], *PCNA*[120]. The remaining genes correspond to the top-5, most selective genes for each cluster. From the statistically significant genes (adjusted p-value < 0.001, MAST with Bonferroni correction using all features), the top-10 (log2 fold-change) were sorted according to the percent of positive cells and the top-5 were plotted. Gene order follows the cluster order. Balloon size: percent of positive cells. Color intensity: scaled expression. Blue: high, Gray: low. **(b)** Balloon-plot of the detected cholinergic-synapse pathway genes (KEGG id: 217716). Balloon size: percent of positive cells. Color intensity: log₂(normalized UMI-counts+1) (library size was normalized to 10.000) expression. Blue: high, Gray: low. **(c)** Heatmap of differentially expressed transcription factors (TFs) along the SCP-neuronal trajectory, according to tradeSeq[21]. Stars: analyzed genes in 'D-E'. Color intensity: scaled expression. Dark red: high, Gray: low. **(d)** UMAP-plots of *SOX10*, *ASCL1* and *ISL1* TFs. Expression levels: log₂(normalized UMI-counts+1) (library size was normalized to 10.000). Blue: high. Gray: zero. **(e)** Confocal-microscopy image of an 8.5 pcw ganglion, showing SOX10, ASCL1 and ISL1 expression, detected with immunofluorescence. Dashed outlines: manually segmented nuclei. SOX10^pos SCPs (arrows), SOX10^pos ASCL1^pos transitioning SCPs (asterisks), ASCL1^pos SOX10^neg immature neurons (hashes), ISL1^pos ASCL1^neg mature neurons (arrowheads). Scale-bar: 5 μm.

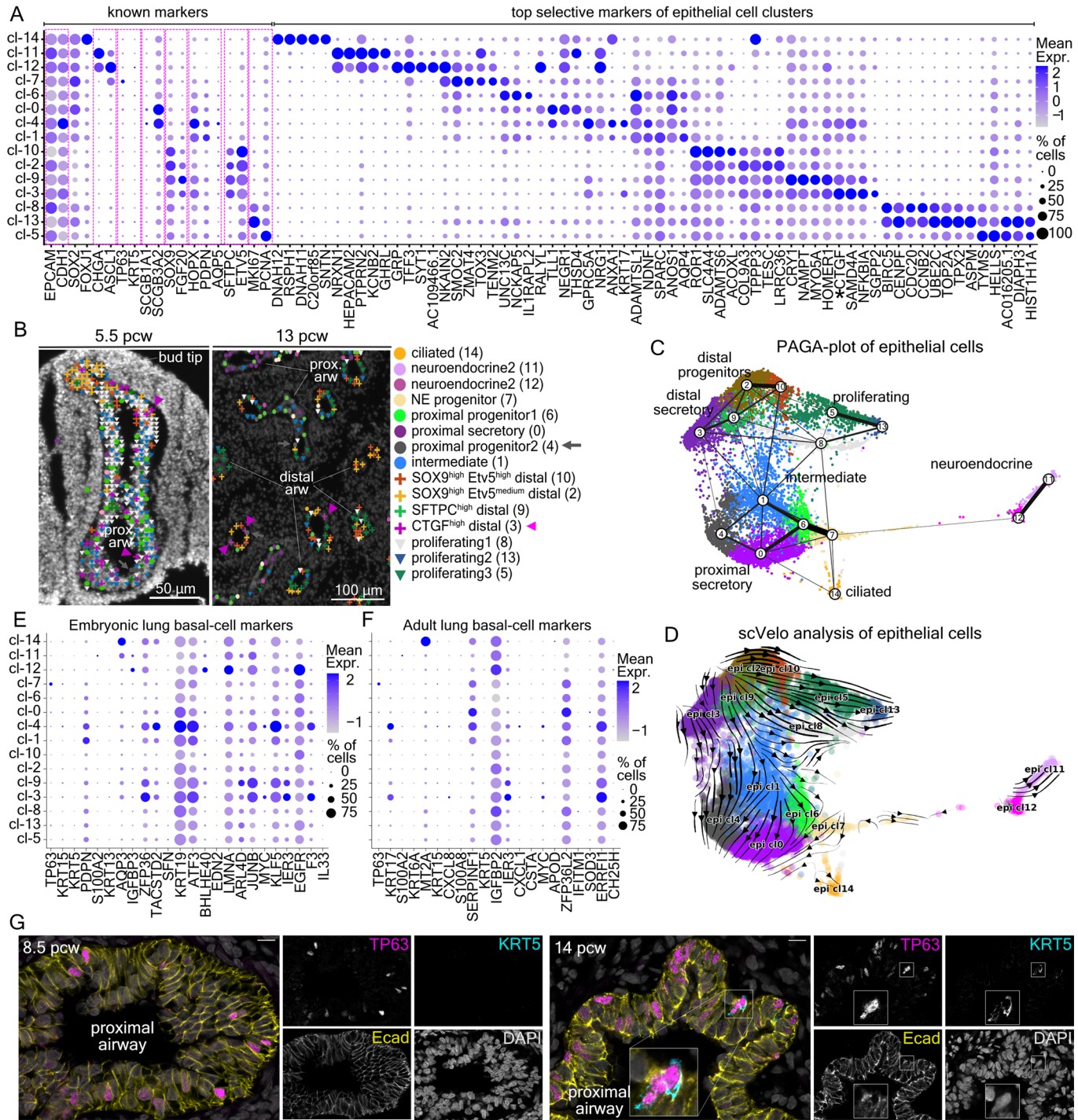

**Extended Data Fig. 8 | Analysis of epithelial cell heterogeneity. (a)** Balloon-plot of known epithelial markers in the clusters of Fig. 4a, using data from all analyzed donors. General: *EPCAM*, *CDH1*, Proximal: *SOX2*[6], Ciliated: *FOXJ1*[107], Neuroendocrine: *CHGA*, *ASCL1*[106], Basal: *TP63*, *KRT5*[137], Club cells: *SCGB1A1*, *SCGB3A2*[138], Distal: *SOX9*[6], *FGF20*, Alveolar Type 1 (AT1): *HOPX*, *PDPN*, *AQP5*[6], AT2: *SFTPC*, *ETV5*[139] and Proliferating: *MKI67*[114], *PCNA*[120] together with the top-5 identified selective markers (adjusted p-value <0.001, MAST, Bonferroni corrected). The top-10 (log2 fold-change) were selected according to the percentage of positive cells in the cluster. The top-5 were plotted. Gene order follows the cluster order. **(b)** Annotation of segmented airway areas with PciSeq, using HybISS data in 5.5 pcw (left) and 13 pcw (right) airways. Distal clusters: cross, proliferating: inverted triangle and proximal: circle. Gray arrows: prox. progenitor2 (cl-4), magenta arrowheads: *CTGF*[high] distal (cl-3). 'prox.': proximal,

'arw': airway. **(c)** PAGA-plot of the analyzed epithelial cells, superimposed on the Fig. 4a UMAP-plot. Line thickness: cluster-connection probability. **(d)** Epithelial-cell scVelo-analysis. Arrow direction: future cell-state, arrow size: transition possibility. **(e)** Balloon-plot of known embryonic basal-cell markers[47]. **(f)** Balloon-plot of the top-20 adult basal-cell markers[2], together with *TP63* expression in our dataset (blue) shows minimal expression of typical adult basal-cell markers in epithelial cells. **(g)** Single-plane confocal-microscopy immunofluorescence images for TP63 (magenta), KRT5 (cyan) and E-cadherin (yellow) on 8.5 (top) and 14 (bottom) pcw lung sections. TP63[pos] cells were mainly localized in proximal airways, with a very small portion being KRT5[pos]. Nuclear DAPI: gray. Scale-bar: 10 μm. In Balloon-plots, balloon size: percent of positive cells. Color intensity: scaled expression. Blue: high, Gray: low.

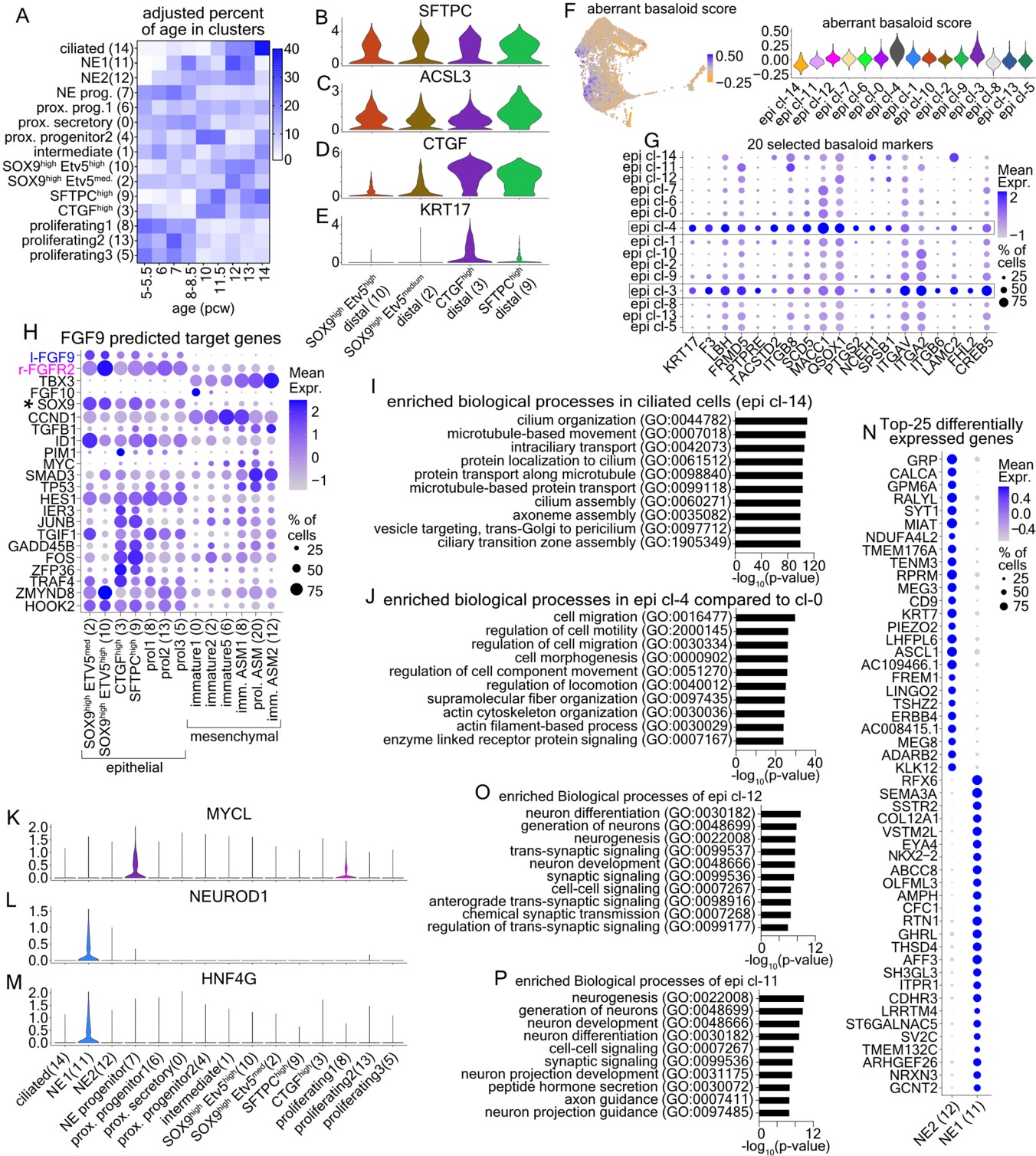

**Extended Data Fig. 9 | Exploring the diversity within airway neighborhoods.**
**(a)** Heatmap of proportions of donor ages in epithelial clusters. To avoid bias, we normalized according to cell numbers in each stage. Dark blue: high, White: zero. **(b–e)** Violin plots of *SFTPC* (B), *ACSL3* (C), *CTGF* (D) and *KRT17* (E) expression levels in the distal epithelial clusters. **(f)** All epithelial-cell UMAP-plot (left) and Violin-plot (right) of the activated-epithelial score, according to the aggregate expression of 96 basaloid[4] selective markers (see Supplementary Table 1–8). Blue: high, orange: low. **(g)** Balloon-plot of epithelial cell-clusters, showing 20 selected basaloid-cell markers. **(h)** Balloon-plot of the top-20 predicted *FGF9*-target genes (by NicheNet). **(i)** p-value bar-plot of the top-10 biological processes in ciliated cells (epi cl-14). **(j)** As in 'I' for the proximal progenitor cells (epi cl-4) compared to the proximal secretory (epi cl-0). **(k–m)** Violin-plots of the *MYCL*

(K), *NEUROD1* (L) and *HNF4G* (M) in all epithelial clusters. **(n)** Balloon-plot of NE-cluster markers. The top-50 markers (log2 fold-change, adjusted p-value <0.001, MAST, Bonferroni corrected) were sorted according to the number of positive cells in each cluster and the top-25 were plotted **(o)** p-value bar-plot of the top-10 biological process in epi cl-11 compared to epi cl-12, using its upregulated genes (adjusted p-value <0.001, calculated by MAST). **(p)** as in 'O' for epi cl-12, compared to epi cl-11. The p-values of enriched biological processes were calculated according to the Hypergeometric Probability Mass Function of https://toppgene.cchmc.org/, using default settings. In Balloon-plots, balloon size: percent of positive cells. Color intensity: scaled expression. Blue: high, Gray: low. In 'B-D' and 'K-M', expression levels: log₂(normalized UMI-counts+1) (library size was normalized to 10.000). All donors were included in the analyses.

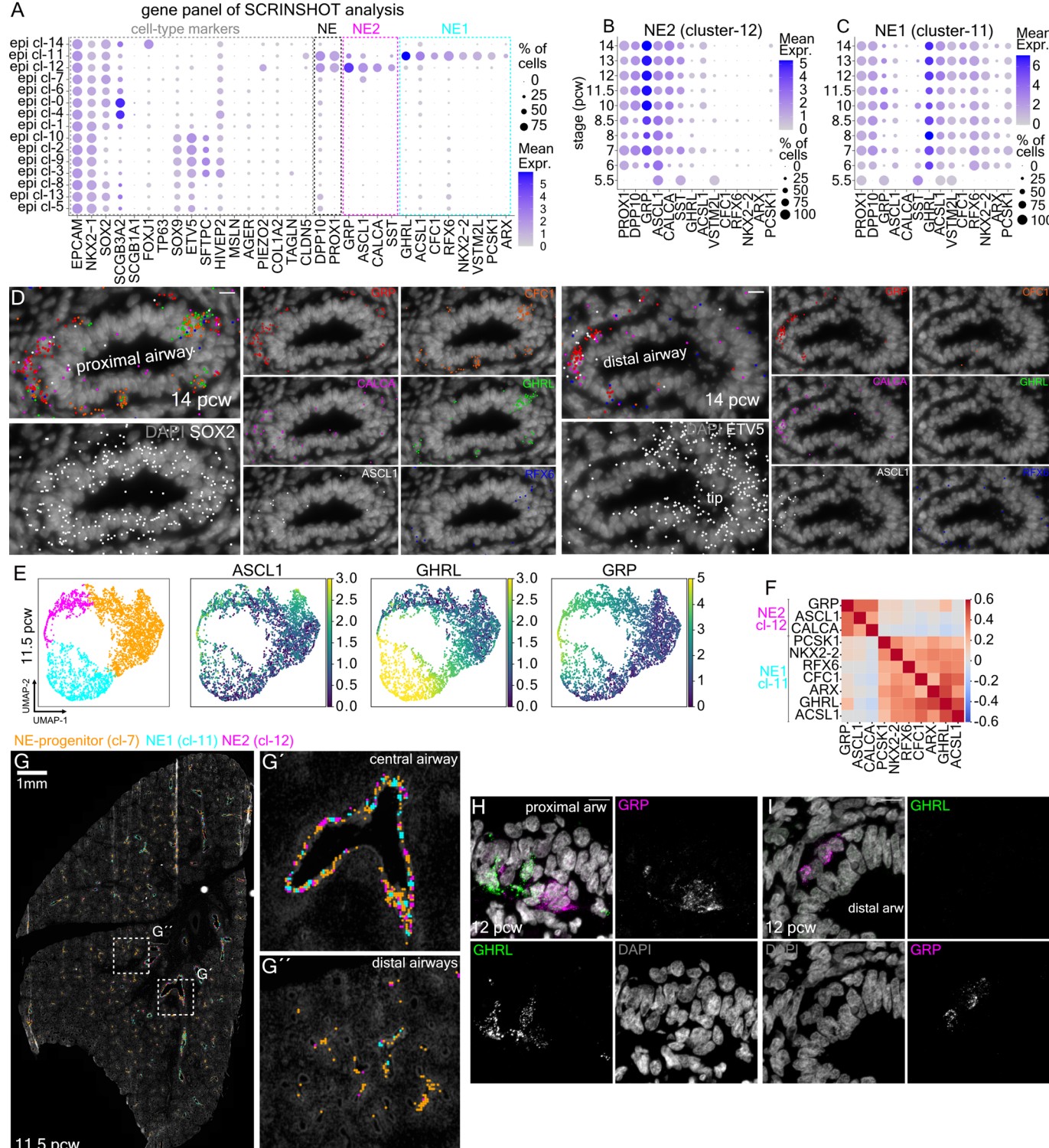

**Extended Data Fig. 10 | See next page for caption.**

**Extended Data Fig. 10 | Spatial distribution of neuroendocrine cell identities.**
**(a)** Balloon-plot of the expression of the selected 31 genes for SCRINSHOT analysis. i) general NE-markers *(PROX1, DPP10)*, ii) cl-12 markers (*ASCL1, GRP, SST* and *CALCA*), iii) cl-11 markers (*GHRL, ACSL1, RFX6, ARX, CFC1, VSTM2L, PCSK1* and *NKX2-2*), together with epithelial and mesenchymal markers (*EPCAM, NKX2-1, SOX2, SCGB3A2, SCGB1A1, FOXJ1, TP63, SOX9, ETV5, SFTPC, HIVEP2, MSLN, AGER, PIEZO2, COL1A2, TAGLN* and *CLDN5*). **(b)** Balloon-plot showing NE-marker expression changes over time in cl-12 cells and **(c)** in cl-11 cells. In 'A-C', the whole epithelial scRNA-Seq dataset (17 donors) was used. Balloon size: percent of positives. Color intensity: $\log_2$(normalized UMI-counts+1) (library size was normalized to 10.000). Blue: high, Gray: zero. **(d)** Images of a 14 pcw lung proximal (top) and a distal (bottom) airway, analyzed by SCRINSHOT. *CFC1* (orange), *GHRL* (green), *RFX6* (blue), *GRP* (red), *CALCA* (magenta) and *ASCL1*

(gray). Scale-bar: 10 μm. Data are available in: https://hdca-sweden.scilifelab.se/tissues-overview/lung/ **(e)** UMAP-plots of neuroendocrine-assigned bins (see Methods) showing the suggested clusters and the *ASCL1, GHRL* and GRP detected mRNAs. Color-scale: $\log_2$(detected mRNAs of the indicated gene + 1). Yellow: high, Dark-blue: zero. NE-progenitor (cl-7), NE1 (cl-12) and NE2 (cl-11) resemble epithelial clusters −7, −12 and −11, respectively. **(f)** Correlation heatmap of the detected mRNAs for the indicated NE-markers. Red: positive, Blue: negative correlation. 'E' and 'F' are based on the 11.5 pcw analyzed lung section of 'G'. **(g)** A spatial map for the indicated NE-populations. DAPI: gray, NE-progenitor: orange, NE1: cyan, NE2: magenta. Magnified **(G´)** proximal and **(G´´)** distal airways of the squares in 'G'. **(h–i)** Confocal-microscopy images of immunofluorescence for GRP (epi cl-12 marker: magenta) and GHRL (epi cl-11 marker: green), on 12 pcw proximal (H) and distal (I) lung airways. Nuclear DAPI: gray. Scale-bar: 10 μm.

| | |
|---|---|

# Reporting Summary

## Statistics

For all statistical analyses, confirm that the following items are present in the figure legend, table legend, main text, or Methods section.

| n/a | Confirmed | |
|---|---|---|
| ☐ | ☒ | The exact sample size (*n*) for each experimental group/condition, given as a discrete number and unit of measurement |
| ☒ | ☐ | A statement on whether measurements were taken from distinct samples or whether the same sample was measured repeatedly |
| ☐ | ☒ | The statistical test(s) used AND whether they are one- or two-sided<br>*Only common tests should be described solely by name; describe more complex techniques in the Methods section.* |
| ☐ | ☒ | A description of all covariates tested |
| ☐ | ☒ | A description of any assumptions or corrections, such as tests of normality and adjustment for multiple comparisons |
| ☐ | ☒ | A full description of the statistical parameters including central tendency (e.g. means) or other basic estimates (e.g. regression coefficient) AND variation (e.g. standard deviation) or associated estimates of uncertainty (e.g. confidence intervals) |
| ☐ | ☒ | For null hypothesis testing, the test statistic (e.g. *F*, *t*, *r*) with confidence intervals, effect sizes, degrees of freedom and *P* value noted<br>*Give P values as exact values whenever suitable.* |
| ☒ | ☐ | For Bayesian analysis, information on the choice of priors and Markov chain Monte Carlo settings |
| ☒ | ☐ | For hierarchical and complex designs, identification of the appropriate level for tests and full reporting of outcomes |
| ☐ | ☒ | Estimates of effect sizes (e.g. Cohen's *d*, Pearson's *r*), indicating how they were calculated |

*Our web collection on statistics for biologists contains articles on many of the points above.*

## Software and code

Policy information about availability of computer code

| Data collection | The cDNA Libraries were prepared with the Chromium Single Cell 3′ Reagent Kit v2 and v3 and sequenced on an Illumina NovaSeq 6000 (Illumina) sequencer. Reads were aligned to the human reference genome GRCh38-3.0.0 and libraries were demultiplexed and aligned with the 10X Genomics pipeline CellRanger (version 3.0.2). Loom files were generated for each sample by running Velocyto (0.17.17). Tissues sections, analyzed by Spatial Transcriptomics, were stained with Hematoxylin and Eosin Y and imaged with a Zeiss Imager.Z2 Microscope (Carl Zeiss Microscopy GmbH), using the Metafer5 software (version 3.14.3) (MetaSystems Hard & Software GmbH).<br><br>HybISS data were collected with a Zeiss Axio Imager.Z2 epifluorescence microscope (Carl Zeiss Microscopy, GmbH), equipped with a Zeiss Plan-Apochromat 20x/0.8 objective (Carl Zeiss Microscopy, GmbH, 420650-9901), an automatic multi-slide stage (PILine, M-686K011), a Lumencor® SPECTRA X light engine LED source (Lumencor, Inc.) and the quad band Chroma 89402 (DAPI, Cy3, Cy5), the quad band Chroma 89403 (AlexaFluor750), and the single band Zeiss 38HE (AlexaFluor488). Images were obtained with an ORCA-Flash4.0 LT Plus sCMOS camera (2048 × 2048, 16-bit, Hamamatsu Photonics K. K.), using the Zen Blue 2.5 software (Carl Zeiss Microscopy, GmbH).<br><br>SCRINSHOT data were collected with a Zeiss Axio Observer Z.2 fluorescent microscope (Carl Zeiss Microscopy, GmbH) with a Colibri 7 led light source (Carl Zeiss Microscopy, GmbH, 423052-9770-000), equipped with a Zeiss 20X/0.75 Plan-Apochromat, a Zeiss AxioCam 506 Mono digital camera, an automated stage and the Chroma filters: 49000 (DAPI), 49003 (FITC), 49304 (Cy3), 49307 (Cy5), 49310 (Texas Red) and 49007 (Atto740), using the Zen Blue 2.5 software (Carl Zeiss Microscopy GmbH).<br><br>Immunofluorescence image datasets were generated with a Zeiss LSM800 confocal microscope, equipped with a Plan-Apochromat 40X/1.30 oil objective, using the Zen Blue 2.5 software (Carl Zeiss Microscopy GmbH) and a Zeiss LSM780 confocal microscope equipped with a Plan-Apochromat 63X/1.40 oil DIC M27 objective. |
|---|---|
| Data analysis | For scRNA-Seq analyses, we used the CellRanger 3.0.2 Pipeline (10X Genomics), the Velocyto 0.17.17 and the following R-packages: SeuratDisk 0.0.0.9019, loomR 0.2.1.9000, SeuratWrappers 0.3.0, Seurat 4.0.5, DoubletFinder 2.0.3, sctransform 0.3.2, MAST 1.18.0, slingshot 2.0.0, |

tradeSeq 1.6.0, fastcluster 1.2.3, CellChat 1.1.0, nichenetr 1.0.0, igraph 1.2.7 and fpc 2.2-9.
For scVelo analyses, we used the following Python 3.9.7 packages: cellrank 1.5.1, loompy 3.0.6, matplotlib 3.5.1, numpy 1.20.3, pandas 1.4.1, scanpy 1.8.2, scvelo 0.2.4.
For ST analyses, we used the Space Ranger 1.0.0 Pipeline (10X Genomics), the R-packages: harmony_0.1.0 and STutility 0.1.0 and the Python-package: stereoscope v.03.
For HybISS analyses, we used an in-house pipeline based on MATLAB (https://github.com/Moldia/iss_starfish). Cell type assignment was done with the Probabilistic cell-typing for in situ sequencing (PciSeq) function implemented in Matisse (1.1.0) (https://github.com/Moldia/Matisse).
For Tangram cell typing, we used the package tangram-sc (1.0.2) in Python 3.8.5.
For SCRINSHOT analyses, Zen Blue 2.5 (Carl Zeiss Microscopy GmbH), ImageJ Fiji, R 4.1.0, R-Studio 1.4.1106 and CellProfiler 4.13.
For neuroendocrine cell spatial analyses, we used the following Python 3.7.12 packages: anndata 0.7.8, scanpy 1.8.2, leidenalg 0.8.8, matplotlib 3.5.1, numpy 1.21.5, pandas 1.3.5.
The browser-based representation of the data is available with the TissUUmaps tool (https://tissuumaps.github.io/).

The scripts for all analyses can be accessed in 10.5281/zenodo.7143091.

For manuscripts utilizing custom algorithms or software that are central to the research but not yet described in published literature, software must be made available to editors and reviewers. We strongly encourage code deposition in a community repository (e.g. GitHub). See the Nature Portfolio guidelines for submitting code & software for further information.

## Data

Policy information about availability of data

All manuscripts must include a data availability statement. This statement should provide the following information, where applicable:
- Accession codes, unique identifiers, or web links for publicly available datasets
- A description of any restrictions on data availability
- For clinical datasets or third party data, please ensure that the statement adheres to our policy

The GRCh38-3.0.0 reference human genome can be accessed from 10X Genomics, in the section: References - 3.0.0 (November 19, 2018)
All sequencing data have been deposited in GEO (GSE215898), comprising of single cell data (GSE215895) and spatial transcriptomics data (GSE215897).
The datasets generated during and/or analyzed during the current study are available at https://hdca-sweden.scilifelab.se/tissues-overview/lung/.
The scRNA-Seq data can be additionally accessed in https://cells.ucsc.edu/?ds=lung-dev. scRNA-Seq datasets of individual-donors can be accessed at DOI: 10.5281/zenodo.6386452.
The used scRNA-Seq datasets, containing subsets of the whole-dataset and of the mesenchymal cell dataset are available at 10.5281/zenodo.7143999.
The raw-data of the fluorescence images can be accessed at DOI: 10.1101/2022.01.11.475631 and DOI: 10.5281/zenodo.6673650.
Spatial Transcriptomics (ST) raw data can be accessed at DOI: 10.5281/zenodo.6661019.
scVelo datasets and analysis files can be accessed at DOI: 10.5281/zenodo.6673667.
Raw-image datasets of HybISS (180 GB) and SCRINSHOT (683 GB) are available from the corresponding authors on reasonable request because of data size limitations.

# Field-specific reporting

Please select the one below that is the best fit for your research. If you are not sure, read the appropriate sections before making your selection.

☒ Life sciences          ☐ Behavioural & social sciences          ☐ Ecological, evolutionary & environmental sciences

For a reference copy of the document with all sections, see nature.com/documents/nr-reporting-summary-flat.pdf

# Life sciences study design

All studies must disclose on these points even when the disclosure is negative.

| | |
|---|---|
| Sample size | No statistical method was used to predetermine sample size. |
| Data exclusions | Low quality scRNA-Seq cDNA libraries based on the number of detected genes and the percent of mitochondrial genes in addition to potential multiplets were excluded from further analyses. Regarding all other methods no data were excluded from the analyses. |
| Replication | All attempts at replication were successful. |
| Randomization | The experiments were not randomized. |
| Blinding | The Investigators were not blinded to allocation during experiments and outcome assessment. |

# Behavioural & social sciences study design

All studies must disclose on these points even when the disclosure is negative.

| | |
|---|---|
| Study description | *Briefly describe the study type including whether data are quantitative, qualitative, or mixed-methods (e.g. qualitative cross-sectional, quantitative experimental, mixed-methods case study).* |

| Research sample | *State the research sample (e.g. Harvard university undergraduates, villagers in rural India) and provide relevant demographic information (e.g. age, sex) and indicate whether the sample is representative. Provide a rationale for the study sample chosen. For studies involving existing datasets, please describe the dataset and source.* |
|---|---|
| Sampling strategy | *Describe the sampling procedure (e.g. random, snowball, stratified, convenience). Describe the statistical methods that were used to predetermine sample size OR if no sample-size calculation was performed, describe how sample sizes were chosen and provide a rationale for why these sample sizes are sufficient. For qualitative data, please indicate whether data saturation was considered, and what criteria were used to decide that no further sampling was needed.* |
| Data collection | *Provide details about the data collection procedure, including the instruments or devices used to record the data (e.g. pen and paper, computer, eye tracker, video or audio equipment) whether anyone was present besides the participant(s) and the researcher, and whether the researcher was blind to experimental condition and/or the study hypothesis during data collection.* |
| Timing | *Indicate the start and stop dates of data collection. If there is a gap between collection periods, state the dates for each sample cohort.* |
| Data exclusions | *If no data were excluded from the analyses, state so OR if data were excluded, provide the exact number of exclusions and the rationale behind them, indicating whether exclusion criteria were pre-established.* |
| Non-participation | *State how many participants dropped out/declined participation and the reason(s) given OR provide response rate OR state that no participants dropped out/declined participation.* |
| Randomization | *If participants were not allocated into experimental groups, state so OR describe how participants were allocated to groups, and if allocation was not random, describe how covariates were controlled.* |

# Ecological, evolutionary & environmental sciences study design

All studies must disclose on these points even when the disclosure is negative.

| Study description | *Briefly describe the study. For quantitative data include treatment factors and interactions, design structure (e.g. factorial, nested, hierarchical), nature and number of experimental units and replicates.* |
|---|---|
| Research sample | *Describe the research sample (e.g. a group of tagged Passer domesticus, all Stenocereus thurberi within Organ Pipe Cactus National Monument), and provide a rationale for the sample choice. When relevant, describe the organism taxa, source, sex, age range and any manipulations. State what population the sample is meant to represent when applicable. For studies involving existing datasets, describe the data and its source.* |
| Sampling strategy | *Note the sampling procedure. Describe the statistical methods that were used to predetermine sample size OR if no sample-size calculation was performed, describe how sample sizes were chosen and provide a rationale for why these sample sizes are sufficient.* |
| Data collection | *Describe the data collection procedure, including who recorded the data and how.* |
| Timing and spatial scale | *Indicate the start and stop dates of data collection, noting the frequency and periodicity of sampling and providing a rationale for these choices. If there is a gap between collection periods, state the dates for each sample cohort. Specify the spatial scale from which the data are taken* |
| Data exclusions | *If no data were excluded from the analyses, state so OR if data were excluded, describe the exclusions and the rationale behind them, indicating whether exclusion criteria were pre-established.* |
| Reproducibility | *Describe the measures taken to verify the reproducibility of experimental findings. For each experiment, note whether any attempts to repeat the experiment failed OR state that all attempts to repeat the experiment were successful.* |
| Randomization | *Describe how samples/organisms/participants were allocated into groups. If allocation was not random, describe how covariates were controlled. If this is not relevant to your study, explain why.* |
| Blinding | *Describe the extent of blinding used during data acquisition and analysis. If blinding was not possible, describe why OR explain why blinding was not relevant to your study.* |

Did the study involve field work? ☐ Yes ☒ No

# Reporting for specific materials, systems and methods

We require information from authors about some types of materials, experimental systems and methods used in many studies. Here, indicate whether each material, system or method listed is relevant to your study. If you are not sure if a list item applies to your research, read the appropriate section before selecting a response.

## Materials & experimental systems

| n/a | Involved in the study |
|-----|----------------------|
| ☐ | ☒ Antibodies |
| ☒ | ☐ Eukaryotic cell lines |
| ☒ | ☐ Palaeontology and archaeology |
| ☒ | ☐ Animals and other organisms |
| ☐ | ☒ Human research participants |
| ☒ | ☐ Clinical data |
| ☒ | ☐ Dual use research of concern |

## Methods

| n/a | Involved in the study |
|-----|----------------------|
| ☒ | ☐ ChIP-seq |
| ☒ | ☐ Flow cytometry |
| ☒ | ☐ MRI-based neuroimaging |

# Antibodies

| | |
|---|---|
| Antibodies used | Primary antibodies:<br>anti-PHOX2B goat polyclonal antibody (R&D Systems, AF4940-SP, working dilution (wd:2µg/mL, Lot No: CBDC0321061)<br>anti-DLL3 rabbit monoclonal antibody (Cell Signalling Technology, 71804, clone: E3J5R, wd:1:50, Lot No: 3)<br>anti-NF-M mouse monoclonal antibody (DSHB, 2H3, clone: 2H3, wd:2µg/ml, Lot No: 7/16/20)<br>anti-COL13A1 rabbit polyclonal antibody (NovusBio, NBP2-13854, wd:1:100, Lot No: 21613)<br>Cy3 anti-αSmooth Muscle Actin (ACTA2) mouse monoclonal antibody (Sigma Aldrich, C6198, clone:1A4, wd:1:2000, Lot No: 124M4815V)<br>anti-GHRL rat monoclonal antibody (R&D Systems, MAB8200-SP, clone: 883622, wd:1.25µg/ml, Lot No: CILU0220021)<br>anti-GRP rabbit polyclonal antibody (Bioss, bs-0011R, wd:1:200, Lot No: AI08112480)<br>anti-SOX10 goat polyclonal antibody (R&D Systems, AF2864-SP, wd:5µg/ml, Lot No: VRY1220031)<br>anti-MASH1 rabbit monoclonal antibody (Abcam, ab240385, clone: EPR19840, wd:1:100, Lot No: GR3258827-2)<br>anti-ISL1 mouse monoclonal antibody (DSHB, 40.2D6, clone: 40.2D6, wd:1.3µg/ml, Lot No: 2/18/21)<br>anti-E-cadherin mouse Alexa Flour 555 (BD Biosciences, 560064, clone: 36/E-Cadherin, wd:1:100, Lot No: 4337645),<br>anti-E-cadherin mouse Alexa Flour 647 (BD Biosciences, 560062, clone: 36/E-Cadherin, wd:1:100, Lot No: 7040557)<br>anti-p63α rabbit monoclonal antibody (Cell Signaling Technologies, 13109, clone: D2K8X, wd:1:100, Lot No: 3)<br>anti-Krt5 chicken polyclonal antibody (Biolegend, 905901, wd:1:400, Lot No: B29722)<br>anti-E-cadherin rat monoclonal antibody (ThermoFisher Scientific, 13-1900, clone: ECCD-2, wd:10µg/ml, Lot SL2474201)<br>anti-KI67 rabbit polyclonal antibody (Invitrogen, PA5-19462, wd:1:500, Lot:GR32000471)<br>anti-KRT17 rabbit polyclonal antibody (Sigma Aldrich, HPA000452, wd:1:200, Lot:A08686)<br>anti-SST mouse monoclonal antibody (NovusBio, NBP2-37447, clone: 7G5, wd:1:200, Lot:160825)<br>and anti-SSTR2 rabbit polyclonal antibody (Proteintech, 20404-1-AP, wd:1:100, Lot:00013177)<br>Secondary antibodies:<br>Alexa Fluor 488 donkey anti-goat (Jackson ImmunoResearch, 705-545-147, wd: 1:400)<br>Cy3 donkey anti-rabbit (Jackson ImmunoResearch, 711-165-152, wd: 1:400)<br>Cy5 donkey anti-mouse (Jackson ImmunoResearch, 715-175-151, wd: 1:400)<br>Cy5 donkey anti-rabbit (Jackson ImmunoResearch, 711-175-152, wd: 1:400)<br>Alexa Fluor 488 donkey anti-rat (Jackson ImmunoResearch, 712-545-153, wd:1:400)<br>Alexa Fluor 488 donkey anti-rabbit (ThermoFischer Scientific, A-21206, wd:1:400)<br>Alexa Fluor Cy3 donkey anti-chicken (Jackson ImmunoResearch, 703-165-155, wd:1:400)<br>Alexa Fluor 647 goat anti-mouse (Invitrogen, A32728, wd:1:250)<br>Alexa Fluor 488 goat anti-rabbit (Invitrogen, A11034, wd:1:250)<br>Alexa Fluor 790 donkey anti-rabbit (Jackson ImmunoResearch, 711-656-152, wd:1:400)<br>Alexa Fluor Cy3 donkey anti-mouse (Jackson ImmunoResearch, 715-165-151, wd:1:400)<br>Alexa Fluor 647 donkey anti-rabbit (Jackson ImmunoResearch, 711-606-152, wd:1:400) |
| Validation | anti-PHOX2B goat polyclonal antibody (R&D Systems, AF4940-SP): Manufacturer provides information regarding the species-reactivity and citation(s) for usage in tissue sections for immunohistochemistry/immunofluorescence.<br>anti-DLL3 rabbit monoclonal antibody (Cell Signalling Technology, 71804): Manufacturer provides information and citation(s) regarding the species-reactivity and usage on tissue sections for immunohistochemistry/immunofluorescence.<br>anti-NF-M mouse monoclonal antibody (DSHB, 2H3): Manufacturer provides information regarding the species-reactivity and usage on tissue sections for immunohistochemistry/immunofluorescence.<br>anti-COL13A1 rabbit polyclonal antibody (NovusBio, NBP2-13854): Manufacturer provides information regarding the species-reactivity and usage on tissue sections for immunohistochemistry/immunofluorescence.<br>Cy3 anti-αSmooth Muscle Actin (ACTA2) mouse monoclonal antibody (Sigma Aldrich, C6198): Manufacturer provides information and citation(s) regarding the species-reactivity and usage on tissue sections for immunohistochemistry/immunofluorescence.<br>anti-GHRL rat monoclonal antibody (R&D Systems, MAB8200-SP): Manufacturer provides information and citation(s) regarding the species-reactivity and usage on tissue sections for immunohistochemistry/immunofluorescence.<br>anti-GRP rabbit polyclonal antibody (Bioss, bs-0011R): Manufacturer provides information regarding the species-reactivity and usage on tissue sections for immunohistochemistry.<br>anti-SOX10 goat polyclonal antibody (R&D Systems, AF2864-SP): Manufacturer provides information and citation(s) regarding the species-reactivity and usage on tissue sections for immunohistochemistry/immunofluorescence.<br>anti-MASH1 rabbit monoclonal antibody (Abcam, ab240385): Manufacturer provides information regarding the species-reactivity and usage on tissue sections for immunohistochemistry/immunofluorescence.<br>anti-ISL1 mouse monoclonal antibody (DSHB, 40.2D6): Manufacturer provides information regarding the species-reactivity and usage on tissue sections for immunohistochemistry/immunofluorescence.<br>anti-E-cadherin mouse Alexa Flour 555 (BD Biosciences, 560064): Manufacturer provides information and citation(s) regarding the species-reactivity and usage on tissue sections for immunohistochemistry/immunofluorescence.<br>anti-E-cadherin mouse Alexa Flour 647 (BD Biosciences, 560062): Manufacturer provides information and citation(s) regarding the |

March 2021

species-reactivity and usage on tissue sections for immunohistochemistry/immunofluorescence.

anti-p63α rabbit monoclonal (Cell Signalling Technologies, 13109): Manufacturer provides information and citation(s) regarding the species-reactivity and usage for immunofluorescence.

anti-Krt5 chicken polyclonal (Biolegend, 905901). Manufacturer provides information and citation(s) regarding the species-reactivity and usage on tissue sections for immunohistochemistry/immunofluorescence.

anti-E-cadherin rat monoclonal antibody (ThermoFisher Scientific, 13-1900). Manufacturer provides information and citation(s) regarding the species-reactivity and usage on tissue sections for immunohistochemistry/immunofluorescence

anti-KI67 rabbit polyclonal antibody (Invitrogen, PA5-19462). Manufacturer provides information and citation(s) regarding the species-reactivity and usage on tissue sections for immunohistochemistry/immunofluorescence

anti-KRT17 rabbit polyclonal antibody (Sigma Aldrich, HPA000452).  Manufacturer provides information and citation(s) regarding the species-reactivity and usage on tissue sections for immunohistochemistry/immunofluorescence

anti-SST mouse monoclonal antibody (NovusBio, NBP2-37447). Manufacturer provides information and citation(s) regarding the species-reactivity and usage on tissue sections for immunohistochemistry/immunofluorescence.

and anti-SSTR2 rabbit polyclonal antibody (Proteintech, 20404-1-AP).Manufacturer provides information and citation(s) regarding the species-reactivity and usage on tissue sections for immunohistochemistry/immunofluorescence.

## Eukaryotic cell lines

Policy information about cell lines

| Cell line source(s) | State the source of each cell line used. |
| --- | --- |
| Authentication | Describe the authentication procedures for each cell line used OR declare that none of the cell lines used were authenticated. |
| Mycoplasma contamination | Confirm that all cell lines tested negative for mycoplasma contamination OR describe the results of the testing for mycoplasma contamination OR declare that the cell lines were not tested for mycoplasma contamination. |
| Commonly misidentified lines (See ICLAC register) | Name any commonly misidentified cell lines used in the study and provide a rationale for their use. |

## Palaeontology and Archaeology

| Specimen provenance | Provide provenance information for specimens and describe permits that were obtained for the work (including the name of the issuing authority, the date of issue, and any identifying information). Permits should encompass collection and, where applicable, export. |
| --- | --- |
| Specimen deposition | Indicate where the specimens have been deposited to permit free access by other researchers. |
| Dating methods | If new dates are provided, describe how they were obtained (e.g. collection, storage, sample pretreatment and measurement), where they were obtained (i.e. lab name), the calibration program and the protocol for quality assurance OR state that no new dates are provided. |

☐ Tick this box to confirm that the raw and calibrated dates are available in the paper or in Supplementary Information.

| Ethics oversight | Identify the organization(s) that approved or provided guidance on the study protocol, OR state that no ethical approval or guidance was required and explain why not. |
| --- | --- |

Note that full information on the approval of the study protocol must also be provided in the manuscript.

## Animals and other organisms

Policy information about studies involving animals; ARRIVE guidelines recommended for reporting animal research

| Laboratory animals | For laboratory animals, report species, strain, sex and age OR state that the study did not involve laboratory animals. |
| --- | --- |
| Wild animals | Provide details on animals observed in or captured in the field; report species, sex and age where possible. Describe how animals were caught and transported and what happened to captive animals after the study (if killed, explain why and describe method; if released, say where and when) OR state that the study did not involve wild animals. |
| Field-collected samples | For laboratory work with field-collected samples, describe all relevant parameters such as housing, maintenance, temperature, photoperiod and end-of-experiment protocol OR state that the study did not involve samples collected from the field. |
| Ethics oversight | Identify the organization(s) that approved or provided guidance on the study protocol, OR state that no ethical approval or guidance was required and explain why not. |

Note that full information on the approval of the study protocol must also be provided in the manuscript.

## Human research participants

Policy information about studies involving human research participants

| Population characteristics | Tissue donors included pregnant women 18 years of age or older, fluent in Swedish that wanted to terminate their pregnancy. Exclusion criteria related to the abortions performed for any medical reasons, by socially compromised women |
| --- | --- |

and/or by women showing any signs that the consent may not be informed. The 17 lung samples were retrieved from fetuses between 5- and 14-weeks post conception (5 pcw: 1 sample, 5.5 pcw: 1 sample, 6 pcw: 2 samples, 7 pcw: 2 samples, 8 pcw: 1 sample, 8.5 pcw: 2 samples, 10 pcw: 1 sample, 11.5 pcw: 2 samples, 12 pcw: 2 samples, 13 pcw: 2 sample, 14 pcw: 1 sample). The gender was determined bioinformatically, after the tissue analysis showing that the tissues were collected from 10 female and 7 male fetuses. Thus there was no bias.

| | |
|---|---|
| Recruitment | The tissue donors were recruited among pregnant women after their decision to terminate their pregnancy. The referral to hospitals was done by a central office for all abortion clinics in the Stockholm region and according to our information it was random. The recruitments were done by midwifes that were not involved in the conducted research. Thus, there was no bias regarding which women were recruited. Inclusion criteria: 18 years of age or older, fluent in Swedish. Exclusion criterias: Abortions performed for any medical reasons, by socially compromised women and/or by women showing any signs that the consent may not be informed. There was not any type of compensation to the tissue donors. |
| Ethics oversight | The use of human fetal material from the elective routine abortions was approved by the Swedish National Board of Health and Welfare and the analysis using this material was approved by the Swedish Ethical Review Authority (2018/769-31). The clinical staff acquired the informed written consent by the donors, that the retrieved material can be used for research purposes and about their ability to withdraw their consent anytime. Then, the fetal material was transferred to the research prenatal material. |

Note that full information on the approval of the study protocol must also be provided in the manuscript.

# ChIP-seq

## Data deposition

☐ Confirm that both raw and final processed data have been deposited in a public database such as GEO.

☐ Confirm that you have deposited or provided access to graph files (e.g. BED files) for the called peaks.

| | |
|---|---|
| Data access links<br>*May remain private before publication.* | *For "Initial submission" or "Revised version" documents, provide reviewer access links. For your "Final submission" document, provide a link to the deposited data.* |
| Files in database submission | *Provide a list of all files available in the database submission.* |
| Genome browser session<br>(e.g. UCSC) | *Provide a link to an anonymized genome browser session for "Initial submission" and "Revised version" documents only, to enable peer review. Write "no longer applicable" for "Final submission" documents.* |

## Methodology

| | |
|---|---|
| Replicates | *Describe the experimental replicates, specifying number, type and replicate agreement.* |
| Sequencing depth | *Describe the sequencing depth for each experiment, providing the total number of reads, uniquely mapped reads, length of reads and whether they were paired- or single-end.* |
| Antibodies | *Describe the antibodies used for the ChIP-seq experiments; as applicable, provide supplier name, catalog number, clone name, and lot number.* |
| Peak calling parameters | *Specify the command line program and parameters used for read mapping and peak calling, including the ChIP, control and index files used.* |
| Data quality | *Describe the methods used to ensure data quality in full detail, including how many peaks are at FDR 5% and above 5-fold enrichment.* |
| Software | *Describe the software used to collect and analyze the ChIP-seq data. For custom code that has been deposited into a community repository, provide accession details.* |

# Flow Cytometry

## Plots

Confirm that:

☐ The axis labels state the marker and fluorochrome used (e.g. CD4-FITC).

☐ The axis scales are clearly visible. Include numbers along axes only for bottom left plot of group (a 'group' is an analysis of identical markers).

☐ All plots are contour plots with outliers or pseudocolor plots.

☐ A numerical value for number of cells or percentage (with statistics) is provided.

## Methodology

| | |
|---|---|
| Sample preparation | *Describe the sample preparation, detailing the biological source of the cells and any tissue processing steps used.* |

| Instrument | *Identify the instrument used for data collection, specifying make and model number.* |
| Software | *Describe the software used to collect and analyze the flow cytometry data. For custom code that has been deposited into a community repository, provide accession details.* |
| Cell population abundance | *Describe the abundance of the relevant cell populations within post-sort fractions, providing details on the purity of the samples and how it was determined.* |
| Gating strategy | *Describe the gating strategy used for all relevant experiments, specifying the preliminary FSC/SSC gates of the starting cell population, indicating where boundaries between "positive" and "negative" staining cell populations are defined.* |

☐ Tick this box to confirm that a figure exemplifying the gating strategy is provided in the Supplementary Information.

# Magnetic resonance imaging

## Experimental design

| Design type | *Indicate task or resting state; event-related or block design.* |
| Design specifications | *Specify the number of blocks, trials or experimental units per session and/or subject, and specify the length of each trial or block (if trials are blocked) and interval between trials.* |
| Behavioral performance measures | *State number and/or type of variables recorded (e.g. correct button press, response time) and what statistics were used to establish that the subjects were performing the task as expected (e.g. mean, range, and/or standard deviation across subjects).* |

## Acquisition

| Imaging type(s) | *Specify: functional, structural, diffusion, perfusion.* |
| Field strength | *Specify in Tesla* |
| Sequence & imaging parameters | *Specify the pulse sequence type (gradient echo, spin echo, etc.), imaging type (EPI, spiral, etc.), field of view, matrix size, slice thickness, orientation and TE/TR/flip angle.* |
| Area of acquisition | *State whether a whole brain scan was used OR define the area of acquisition, describing how the region was determined.* |

Diffusion MRI   ☐ Used   ☐ Not used

## Preprocessing

| Preprocessing software | *Provide detail on software version and revision number and on specific parameters (model/functions, brain extraction, segmentation, smoothing kernel size, etc.).* |
| Normalization | *If data were normalized/standardized, describe the approach(es): specify linear or non-linear and define image types used for transformation OR indicate that data were not normalized and explain rationale for lack of normalization.* |
| Normalization template | *Describe the template used for normalization/transformation, specifying subject space or group standardized space (e.g. original Talairach, MNI305, ICBM152) OR indicate that the data were not normalized.* |
| Noise and artifact removal | *Describe your procedure(s) for artifact and structured noise removal, specifying motion parameters, tissue signals and physiological signals (heart rate, respiration).* |
| Volume censoring | *Define your software and/or method and criteria for volume censoring, and state the extent of such censoring.* |

## Statistical modeling & inference

| Model type and settings | *Specify type (mass univariate, multivariate, RSA, predictive, etc.) and describe essential details of the model at the first and second levels (e.g. fixed, random or mixed effects; drift or auto-correlation).* |
| Effect(s) tested | *Define precise effect in terms of the task or stimulus conditions instead of psychological concepts and indicate whether ANOVA or factorial designs were used.* |

Specify type of analysis:   ☐ Whole brain   ☐ ROI-based   ☐ Both

| Statistic type for inference (See Eklund et al. 2016) | *Specify voxel-wise or cluster-wise and report all relevant parameters for cluster-wise methods.* |
| Correction | *Describe the type of correction and how it is obtained for multiple comparisons (e.g. FWE, FDR, permutation or Monte Carlo).* |

## Models & analysis

nature portfolio | reporting summary

March 2021

| n/a | Involved in the study |
|---|---|
| ☐ | ☐ Functional and/or effective connectivity |
| ☐ | ☐ Graph analysis |
| ☐ | ☐ Multivariate modeling or predictive analysis |

**Functional and/or effective connectivity**

*Report the measures of dependence used and the model details (e.g. Pearson correlation, partial correlation, mutual information).*

**Graph analysis**

*Report the dependent variable and connectivity measure, specifying weighted graph or binarized graph, subject- or group-level, and the global and/or node summaries used (e.g. clustering coefficient, efficiency, etc.).*

**Multivariate modeling and predictive analysis**

*Specify independent variables, features extraction and dimension reduction, model, training and evaluation metrics.*