## [Peer Review File · Nature Cell Biology]

Peer Review Information

Journal: Nature Cell Biology

Manuscript Title: A topographic atlas defines developmental origins of cell heterogeneity in the human embryonic lung

Corresponding author name(s): Mats Nilsson, Christos Samakovlis

Editorial Notes:

Reviewer Comments & Decisions:

Decision Letter, initial version:
--

Date: 10th May 22 04:31:29

Last Sent: 10th May 22 04:31:29

Triggered By: Stylianos Lefkopoulos

From: stylianos.lefkopoulos@springernature.com

To: christos.samakovlis@su.se

CC: ncb@springernature.com

Subject: Decision on Nature Cell Biology submission NCB-RS48074

Message: *Please delete the link to your author homepage if you wish to forward this email to co-authors.

Dear Professor Samakovlis,

Please accept our sincere apologies for the length of time your manuscript, "Developmental origins of cell heterogeneity in the human lung", has been under review at our journal. This is due to the unforeseen inability of Reviewer #3 to provide the referee report in a timely manner. As the expertise of Reviewer #1 and Reviewer #2 can cover the major aspects of the current study, we have reached our decision based on their comments to avoid further delay. If/when we receive the report from Reviewer #3 within a reasonable timeframe, we will send the comments to you, and in that case we would also expect revisions to address Reviewer #3's comments.

Therefore, your manuscript has now been seen by 2 referees, who are experts in single cell analysis, atlas, human lung (referee 1); transcriptomic atlas, lung (referee 2). As you will

see from their comments (attached below) they find this work of potential interest, but have raised substantial concerns, which in our view would need to be addressed with considerable revisions before we can consider publication in Nature Cell Biology.

Nature Cell Biology editors discuss the referee reports in detail within the editorial team, including the chief editor, to identify key referee points that should be addressed with priority. To guide the scope of the revisions, I have listed these points below. We are committed to providing a fair and constructive peer-review process, so please feel free to contact me if you would like to discuss any of the referee comments further.

In particular, it would be essential to:

(A) Strengthen the claims stemming from analyses by providing experimental validations, as specifically indicated by referees. Please note that, in order for the manuscript to be further considered as a Nature Cell Biology study, mere textual and data representation rearrangements or “toning down” claims will not be sufficient. We therefore request that you address the concerns raised by providing experimental validations wherever noted by the reviewers.

Referee 1 notes:

“In the most general respect, there is frankly too much data for one manuscript included here. The types of data presented in the manuscript are descriptive; however, many functional claims are made. The manuscript should be primarily focused on describing the presented spatial transcriptomic data, and functional data (i.e. description of mesenchymal cell maturity lineages or PNEC lineages) should either be limited to a single section at the end with the aim of highlighting how this data could be used as a hypothesis-generating tool, or it should be more briefly mentioned in the discussion section. Examples of line numbers in the manuscript where functional claims were made from descriptive data:
 o 139 – 41; 157 – 61; 185 – 7; 229 – 31; 232 – 45; 274 – 8; 296 – 305; 350 – 7; 385; 387 – 96; 484 – 6; 487 – 505
 o This is also true for the data presented in the “Supplementary Note.”

“Several attempts to correlate adult cell states to developmental precursors are not well validated. For example, authors attempt to benchmark their data against an adult basal cell signature, but a fetal reference seems more relevant. The basaloid phenotype is not described during development and appears unsubstantiated here. Finally, claims about similarities between fetal states and diseased states are not well supported and appear overreaching for the data presented here. The authors should remove these sections or provide additional experimental evidence to support their claims”.

Referee 2 notes:

“Although extensive bioinformatics analysis are performed in the current manuscript, more experimental validation should be performed and the quality of some data should be improved. The bellows are my concerns”.

“It will be better to perform several experimental validation of some cell marker genes, such as the cell proliferation marker Ki67 to evaluate whether the cells are proliferating cells. In

addition, it is difficult to see the signal of SCPs and neuronal cells using H&E staining. Specific markers for staining SCPs and neuronal cells should be performed in Figure 3C-E”.

(B) All other referee concerns pertaining to strengthening existing data, providing controls, methodological details, clarifications and textual changes, should also be addressed.

(C) Finally please pay close attention to our guidelines on statistical and methodological reporting (listed below) as failure to do so may delay the reconsideration of the revised manuscript. In particular please provide:

We would be happy to consider a revised manuscript that would satisfactorily address these points, unless a similar paper is published elsewhere, or is accepted for publication in Nature Cell Biology in the meantime.

- ensure that it conforms to our format instructions and publication policies (see below and www.nature.com/nature/authors/).

- provide a point-by-point rebuttal to the full referee reports verbatim, as provided at the end of this letter.

- provide the completed Editorial Policy Checklist (found here <https://www.nature.com/authors/policies/Policy.pdf>), and Reporting Summary (found here <https://www.nature.com/authors/policies/ReportingSummary.pdf>). This is essential for reconsideration of the manuscript and these documents will be available to editors and referees in the event of peer review. For more information see <http://www.nature.com/authors/policies/availability.html> or contact me.

Nature Cell Biology is committed to improving transparency in authorship. As part of our efforts in this direction, we are now requesting that all authors identified as ‘corresponding author’ on published papers create and link their Open Researcher and Contributor Identifier (ORCID) with their account on the Manuscript Tracking System (MTS), prior to acceptance. ORCID helps the scientific community achieve unambiguous attribution of all scholarly contributions. You can create and link your ORCID from the home page of the MTS by clicking on ‘Modify my Springer Nature account’. For more information please visit www.springernature.com/orcid.

[REDACTED]

We would like to receive a revised submission within six months. We would be happy to consider a revision even after this timeframe, however if the resubmission deadline is missed and the paper is eventually published, the submission date will be the date when the revised manuscript was received.

We hope that you will find our referees' comments, and editorial guidance helpful. Please do not hesitate to contact me if there is anything you would like to discuss.

Best wishes,

Stelios

Stylios Lefkopoulos, PhD
He/him/his
Associate Editor
Nature Cell Biology
Springer Nature
Heidelberger Platz 3, 14197 Berlin, Germany

E-mail: stylios.lefkopoulos@springernature.com
Twitter: @s_lefkopoulos

Reviewers' Comments:

Reviewer #1:

Remarks to the Author:

This manuscript combines scRNA-seq with spatial transcriptomic techniques to create a unique topographic atlas of the developing human lung spanning 5-14 PCW (predominantly the pseudoglandular phase). The authors claim to identify distinct lineages in both secretory, neuroendocrine, and mesenchymal cell states that are predicted from sequencing and spatial data. Several developmentally active pathways described are correlated to pathologic states in human disease (fibrosis, cancer, etc.). The authors demonstrate the utility and implementation of their online tool that integrates these data sets as a means for further exploration of signaling among cell types, spatially and temporally. Overall, the data in the given manuscript will be an extremely valuable resource for the field. The combination of several types of analyses (sequencing, spatial, and signaling) is an achievement and can be instrumental for further studies. There are some major hesitations regarding the claims made in this manuscript:

Major

- In the most general respect, there is frankly too much data for one manuscript included here. The types of data presented in the manuscript are descriptive; however, many

functional claims are made. The manuscript should be primarily focused on describing the presented spatial transcriptomic data, and functional data (i.e. description of mesenchymal cell maturity lineages or PNEC lineages) should either be limited to a single section at the end with the aim of highlighting how this data could be used as a hypothesis-generating tool, or it should be more briefly mentioned in the discussion section. Examples of line numbers in the manuscript where functional claims were made from descriptive data:
 o 139 – 41; 157 – 61; 185 – 7; 229 – 31; 232 – 45; 274 – 8; 296 – 305; 350 – 7; 385; 387 – 96; 484 – 6; 487 – 505
 o This is also true for the data presented in the “Supplementary Note.”

- Were trachea samples or proximal airway included in the analysis? It is unclear what regions of lung were sampled. This is mirrored by the claim that there are no basal cells at the timepoints analyzed in the study. This is inconsistent with previous reports in human (Miller et al. Dev Cell 2020) and mice (Yang et al. Dev Cell 2018), and see preprint (<https://www.biorxiv.org/content/10.1101/2022.01.11.474933v1.full>) How do the authors reconcile these differences?
- Several attempts to correlate adult cell states to developmental precursors are not well validated. For example, authors attempt to benchmark their data against an adult basal cell signature, but a fetal reference seems more relevant. The basaloid phenotype is not described during development and appears unsubstantiated here. Finally, claims about similarities between fetal states and diseased states are not well supported and appear overreaching for the data presented here. The authors should remove these sections or provide additional experimental evidence to support their claims.
- It is unclear how the authors are defining immature mesenchymal cell states. Is this based on PAGA analysis? Moreover, it is unclear what the “intermediate” epithelial cell cluster/state is. The authors should elaborate on how immature cell states were defined and describe what an intermediate cell state means. Similarly, it is unclear why in some cases RNA velocity was used and in other cases lineage trajectory analysis was used; also labelling is inconsistent throughout figures (RNA Velocity vs Velocityto).
- In the interactome analysis between distal epithelial and mesenchymal cell identities, the authors delve into FGF signaling; however, there is no mention of FGFs (Danopoulos et al J Pathol 2019, Nicolic et al. eLife 2017), which have been functionally shown to be important FGF ligands during this stage of development. Additionally, the authors claim mouse-human differences in FGF signaling; however, predictive single cell data in the current manuscript should not be compared to experimental data from mice. Is there comparable negative data in mice to corroborate these claims (lines 283-7)?

Minor

- In general, how the number of cell clusters were defined is unclear. The authors should explore over-clustering and if all cell clusters/cell states presented are biologically meaningful.
- Given the complexity of many of the analytic tools used throughout the manuscript, some brief explanations of each tools function or purpose would be helpful to the naïve reader, along with reasoning for choosing this specific tool (i.e. SCRINSHOT, HYBISS etc).
- Under the header “Distinct populations for known and novel mesenchymal cell states,” the citation of “Extended Figure 1A” should be “Extended Figure 1G.”

- In Figure 1B, the authors should consider simplifying the labels to just the main cell states (mesenchyme, epithelium, etc.) since in following figures, each of these compartments are explored further.
- Why was a lineage trajectory analyzed for ASM but not for other mesenchymal end-states, such as chondrocytes, advF, and pericytes (Figure 2C)? Similarly, why are only a limited selection of clusters included in communication analyses (Extended Figure 3H – I and Extended Figure 4I)?
- Figure title in Figure 3 legend is repeated.
- The orange and purple color pallet on feature plots is confusing because orange (lower expression) is darker than purple (high expression). The authors should consider other color pallets (ie. Supp Data 3).
- The various scale color pallets throughout Figure 6 are confusing, try coordinating the colors to clearly represent the areas described. Additionally, the diagram in Figure 6D is helpful but crowded. Consider additional diagrams broken up by mesenchyme, epithelium, etc. or by distal and proximal compartments.
- Consider arranging dot (balloon) plots in cluster order, rather than grouping by cell type similarity.
- “Zonation patterns of mesenchymal...” section seems misplaced in the text. It should be included with the rest of the data describing the mesenchyme.
- The order of panels in figures are oftentimes not ordered properly. Some changes could easily be made to aid in reader ability (i.e., switching positions of Figure 2C and D and Figure 3C and D).
- Authors cite a stain for Figure 3C and D (line 225), but only computational data is shown.
- First sentence on line 212 needs rewording and comma should be removed.
- In Supplemental Data 3, alveolar markers are show (HOPX, PDPN), but these markers aren't expected to be expressed at the time-points analyzed. The authors could consider additional bud tip markers (NPC2, TESC, CA2). Also, proximal markers MUC5B or MUC5AC and SCGB1A1 are not shown. The authors should comment on how the epithelial clusters compare to published data.
- How were epithelial progenitors determined (line 252)?
- Missing end parentheses (line 86)
- Prerequisite is misspelled (line 268)
- “stronger expressed” is not grammatically correct (line 384)

Reviewer #2:

Remarks to the Author:

In this manuscript entitled "Developmental origins of cell heterogeneity in the human lung" by Sountoulidis et al., the authors created a comprehensive topographic atlas of early human lung development via integrating scRNA-Seq and spatial transcriptomics, showing that distinct secretory and neuroendocrine states activated either during lung fibrosis or small cell lung cancer progression. In addition, the authors also defined the origin of airway fibroblasts associated with airway smooth muscle in bronchovascular bundles and described a trajectory of Schwann cell progenitors to intrinsic parasympathetic neurons controlling bronchoconstriction.

It is a novel and insightful study to integrating scRNA-Seq and spatial transcriptomics analysis, which provides a rich resource for further research. Although extensive bioinformatics analysis are performed in the current manuscript, more experimental validation should be performed and the quality of some data should be improved. The bellows are my concerns.

1. The result of figure 4D and Suppl. Data 4I revealed that the cl-3 cells of epithelial cells showed the expression signature similar to fibrosis. I am wondering whether this pathogenic cell-state appeared during the development, and how this fibrosis-associated signature would change. More specifically, what is the mechanism by which regulates the fibrosis during embryonic development.
2. The authors found a communication between AF and ASM including IGF1 and WNT5A signaling. What is the exact dynamic process of their mutual signaling interactions ranging from 5 to 14 weeks post conception? When does this communication come up? Is there a peak or disappear of this interaction during the development?
3. The authors performed pseudo time analysis for the cluster of the mesenchymal cells, and divided the mesenchymal cells into several detailed subgroup. Does the marker genes corresponding to the subgroup are reported by other studies?
5. It will be better to perform several experimental validation of some cell marker genes, such as the cell proliferation marker Ki67 to evaluate whether the cells are proliferating cells. In addition, it is difficult to see the signal of SCPs and neuronal cells using H&E staining. Specific markers for staining SCPs and neuronal cells should performed in Figure 3C-E.
6. Some of the descriptions are unclear and simple. For example, in line 72, it written "Assuming that the two lungs are relatively bilaterally symmetrical, we used one half for scRNA-Seq and processed the other for spatial analyses". As the lung tissue is a highly polar organ with different cell composition and differentiation fate in different locations, are the areas for collecting samples for scRNA-seq and spatial analyses are symmetrical? Or the whole two lungs are subjected to scRNA-seq and spatial analyses? The authors need to describe the sampling and comparison strategies in detail. In addition, some incorrect writing in the whole text, such as in line 75 "A first clustering of 163.236, high-quality cDNA libraries", and line 551-552 "75.000 and 200.00 reads/cell". Should them be 163,236, 75,000 and 200,00?
7. It will be better to provide all the marker genes for each cell subgroup in Figure 1B by showing in heatmap or dot plot.
8. Scale bars are missing in Figure 1A, 1F, 2E-G, 3C-F, and extended Figure 2B, 3E, 4C.
9. From Supplementary Data 1B, the cell number reach to more than 20,000 (donor4, 5, 6, 14 and 16), even more than 40,000 (donor11 and 17-1), which is much higher than the standard

(https://assets.ctfassets.net/an68im79xiti/4tjk4KvXzTWgTs8f3tvUjq/2259891d68c53693e753e1b45e42de2d/CG000183_ChromiumSingleCell3__v3_UG_Rev_C.pdf) . This causes higher ratio of multicellular, which is difficult to remove using software. The authors need to focus on assessing whether clustering and trajectory for samples from donor4, 5, 6, 14 and 16, as well as donor11 and 17-1 are consistent with the conclusions.

10. According to Extend Figure 1G and H, a group of ciliated cells with high FOXJ1 expression seem to be missing in Figure 1B. Please pay attention to ensure the consistency of presentation and analysis data.

11. The author shows two time points (6pw, 10pw) of the spatial transcriptome samples in Figure 1A, while in Figure 2B, a comparative analysis of 6pw, 8pw and 11.5pw is performed. Why are the data of 10pw missing? It will be better to perform comparative analysis of 6pw, 8pw, 10pw and 11.5pw to reflect the changes in spatial structure during lung development.

Reviewer #3:
None

ABSTRACT AND MAIN TEXT – please follow the guidelines that are specific to the format of your manuscript, as listed in our Guide to Authors

(http://www.nature.com/ncb/pdf/ncb_gta.pdf) Briefly, Nature Cell Biology Articles, Resources and Technical Reports have 3500 words, including a 150 word abstract, and the main text is subdivided in Introduction, Results, and Discussion sections. Nature Cell Biology Letters have up to 2500 words, including a 180 word introductory paragraph (abstract), and the text is not subdivided in sections.

Methods should be written concisely, but should contain all elements necessary to allow interpretation and replication of the results. As a guideline, Methods sections typically do not exceed 3,000 words. The Methods should be divided into subsections listing reagents and techniques. When citing previous methods, accurate references should be provided and any alterations should be noted. Information must be provided about: antibody dilutions, company names, catalogue numbers and clone numbers for monoclonal antibodies; sequences of RNAi and cDNA probes/primers or company names and catalogue numbers if reagents are commercial; cell line names, sources and information on cell line identity and authentication. Animal studies and experiments involving human subjects must be reported in detail, identifying the committees approving the protocols. For studies involving human subjects/samples, a statement must be included confirming that informed consent was obtained. Statistical analyses and information on the reproducibility of experimental results should be provided in a section titled "Statistics and Reproducibility".

All Nature Cell Biology manuscripts submitted on or after March 21 2016 must include a

Data availability statement at the end of the Methods section. For Springer Nature policies on data availability see <http://www.nature.com/authors/policies/availability.html>; for more information on this particular policy see <http://www.nature.com/authors/policies/data/data-availability-statements-data-citations.pdf>. The Data availability statement should include:

- Accession codes for primary datasets (generated during the study under consideration and designated as "primary accessions") and secondary datasets (published datasets reanalysed during the study under consideration, designated as "referenced accessions"). For primary accessions data should be made public to coincide with publication of the manuscript. A list of data types for which submission to community-endorsed public repositories is mandated (including sequence, structure, microarray, deep sequencing data) can be found here <http://www.nature.com/authors/policies/availability.html#data>.
- Unique identifiers (accession codes, DOIs or other unique persistent identifier) and hyperlinks for datasets deposited in an approved repository, but for which data deposition is not mandated (see here for details <http://www.nature.com/sdata/data-policies/repositories>).
- At a minimum, please include a statement confirming that all relevant data are available from the authors, and/or are included with the manuscript (e.g. as source data or supplementary information), listing which data are included (e.g. by figure panels and data types) and mentioning any restrictions on availability.
- If a dataset has a Digital Object Identifier (DOI) as its unique identifier, we strongly encourage including this in the Reference list and citing the dataset in the Methods.

We recommend that you upload the step-by-step protocols used in this manuscript to the Protocol Exchange. More details can be found at www.nature.com/protocolexchange/about.

All imaging data should be accompanied by scale bars, which should be defined in the legend.

Cropped images of gels/blots are acceptable, but need to be accompanied by size markers, and to retain visible background signal within the linear range (i.e. should not be saturated). The boundaries of panels with low background have to be demarked with black lines.

Splicing of panels should only be considered if unavoidable, and must be clearly marked on the figure, and noted in the legend with a statement on whether the samples were obtained and processed simultaneously. Quantitative comparisons between samples on different gels/blots are discouraged; if this is unavoidable, it should only be performed for samples derived from the same experiment with gels/blots were processed in parallel, which needs to be stated in the legend.

The total number of Supplementary Figures (not including the “unprocessed scans” Supplementary Figure) should not exceed the number of main display items (figures and/or tables (see our Guide to Authors and March 2012 editorial <http://www.nature.com/ncb/authors/submit/index.html#suppinfo>; <http://www.nature.com/ncb/journal/v14/n3/index.html#ed>). No restrictions apply to Supplementary Tables or Videos, but we advise authors to be selective in including supplemental data.

GUIDELINES FOR EXPERIMENTAL AND STATISTICAL REPORTING

REPORTING REQUIREMENTS – To improve the quality of methods and statistics reporting in our papers we have recently revised the reporting checklist we introduced in 2013. We are now asking all life sciences authors to complete two items: an Editorial Policy Checklist (found here <https://www.nature.com/authors/policies/Policy.pdf>) that verifies compliance with all required editorial policies and a reporting summary (found

here <https://www.nature.com/authors/policies/ReportingSummary.pdf>) that collects information on experimental design and reagents. These documents are available to referees to aid the evaluation of the manuscript. Please note that these forms are dynamic 'smart pdfs' and must therefore be downloaded and completed in Adobe Reader. We will then flatten them for ease of use by the reviewers. If you would like to reference the guidance text as you complete the template, please access these flattened versions at <http://www.nature.com/authors/policies/availability.html>.

Author Rebuttal to Initial comments

Detailed responses to reviewers

In the revised manuscript "Developmental origins of cell heterogeneity in the human lung" and the present rebuttal letter, we have addressed all the concerns and suggestions from the referees, which we believe vastly improved our manuscript. We have added four validation experiments proposed by the referees to the revised figures. In addition, we have included a figure and panels of a meta-analysis of 3 different sc-mRNA seq data sets to address one of the points of reviewer#1. These are only included in this rebuttal letter together with a point-by-point description of all additions and modifications we have

made to the text. In the detailed response part, the referee's comments are in grey, in blue we explain our introduced changes and in orange we show the exact action we took to modify or introduce new text.

Detailed response to Reviewer1

Reviewer1 expressed concerns regarding our functional claims, that were based on computational predictions and supported by available information from previous functional studies.

In the following paragraphs, we address the concerns on functional claims, we initially made and explain how we have modified them, in the revised manuscript.

139 – 41 ZEB1 was the most up-regulated transcription factor (TF) in the immature ASM cl-12 (Extended Fig. 2G: module-6), suggesting that it activates the SM gene-program in ASM, similarly to VSM¹.

We referred to a paper, where both in vivo and in vitro assays showed that overexpressed Zeb1 directly binds to the alpha smooth muscle actin promoter and positively regulates its expression. Based on the gene-expression similarities of vascular and airway smooth muscle, we suggested that this transcription factor has the same function in the human embryonic lung.

These studies are suggestive but because they were only addressing ZEB1 function in vitro, we removed the text and reference.

157 – 61 The analysis of the ASM trajectory suggests that the spatially dispersed *FGF10^{pos} WNT2^{pos} RSP02^{pos}* progenitors (mes cl-0 and cl-6) enter the ASM program close to the distal airway buds (mes cl-8) and further proliferate (mes cl-20) and gradually differentiate along the distal to proximal epithelial axis (cl-12 and cl-13), presumably through interactions with neighboring epithelial cells and fibroblasts.

This part is an interpretation mainly supported by the spatial localization of the ASM, the epithelial, the adventitial and airway fibroblast clusters (Figure 6 A-B and Extended Figure 7). We now added the spatial distribution of the immature mesenchymal clusters mes cl-0, mes cl-2 and cl-6 in the Suppl. Data 3A and rephrased.

We referred to a paper addressing immature airway smooth muscle cells in developing mouse lung, analyzed by lineage-tracing experiments by Kumar and colleagues². Their findings are consistent with our description.

Since such an experiment is currently not feasible with human material, we removed the sentence according to the suggestion.

185 – 7 Our results identify AFs as a new cell-type in close contact with ASM and suggest their mutual signaling interactions, inducing cell differentiation and the organization of connective tissue around central airways.

We spatially identified and mapped the AF cluster at different developmental stages by unbiased and targeted transcriptomics methods. Antibody staining with a few selective markers showed the interdigitated spatial arrangement of the AFs with ASM. LUM expression in AFs was predicted by Nichenet³ and validated on tissue sections with IF (Figure 2E, G).

We modified the text:

Our results identify AFs as a new cell-type in close contact with ASM and suggest their mutual signaling interactions.

229 – 31 The selective expression of *JAG1* in SCPs suggests that it activates and maintains the SCP state and/or drives the formation of immature Schwann cells as proposed in studies of mouse limb nerves⁴.

Woodhoo and colleagues⁴ published gain-and loss-of-function experiments, to demonstrate that NOTCH-signaling activation controls various aspects of Schwann cell development, including their differentiation and proliferation. Importantly, they showed that neural crest-selective ablation of RBPJ, the key transcriptional mediator of canonical Notch signaling affected the development of satellite cells and the formation of ganglia and Schwann cells. Also, they showed that the Notch ligand, Jagged 1 was expressed by axons and glia, whereas its receptor, Notch 1 was detected only in glial cells. The similarities in expression pattern of Notch signaling components and downstream genes (Figure 3 H) suggest that Notch-signaling functions in a similar manner in the human lung.

We rephrased to:

The selective expression of *JAG1* in SCPs suggests that it activates NOTCH-signaling in parasympathetic ganglia, similarly to its role in mouse limb nerves, which also derive from the neural crest⁴.

232-45 The localization of parasympathetic ganglia in the bronchial interstitium and the NF-M^{pos} projections in 8.5 pcw lungs suggested the early onset of signaling events leading to ASM innervation by parasympathetic post-ganglionic neurons^{5,6}. We analyzed possible interactions between the neurons and ASM cell-identities focusing on ASM-derived signals with receptors and potential targets specifically expressed in neurons (Extended Fig. 3F, G). The ASM-produced chemokine *CXCL12* could activate its

receptor *CXCR4* in neurons (Extended Fig. 3H), regulating their migration, as previously described for midbrain dopaminergic neurons⁷. Similarly, THBS2 secreted by ASM may activate the neuronal migration through the CD47 receptor and induce the *TBX20* and *GFRA* targets, in mature neurons^{8,9} (Extended Fig. 3I). We conclude that pulmonary parasympathetic neurons derive from SCPs, as neurons in human sympathetic adrenal ganglia¹⁰ and mouse parasympathetic ganglia^{11,12}. Lineage trajectories and spatial analysis suggest that NOTCH-signaling, among SCPs and neurons, establishes and maintains ganglionic populations and that short- and long-range interactions of neurons with ASM may mediate bronchial innervation.

We had based our selection of ASM as potential targets for the parasympathetic neurons on the proximity of ganglionic structures to airway epithelial and airway smooth muscle cells in our spatial analysis. Additionally, we referred to mouse lineage labelling experiments with Wnt1-Cre which suggested that parasympathetic neurons touch the ASM⁶.

At present, we cannot experimentally address the direct contact between the parasympathetic neurons and ASM and therefore followed the reviewer's suggestion to remove the paragraph.

274-8 To further explore similarities of cl-3 and basaloid cells, we used the gene expression signature of selective basaloid genes¹³ to score all epithelial cells of our dataset. We found that cl-3 cells show the highest aggregated expression score for this "activated" cell state (Fig. 4D, individual genes in Suppl. Data 3E and Suppl. Data 4I), suggesting a shared epithelial genetic program in lung development and fibrosis.

The "basaloid" signature genes have been defined by Adams and colleagues¹³ and contains 117 genes statistically significant genes (Suppl. Data 6I). Of these, 96 are detected in embryonic epithelium (Average $\log_2(\text{UMIs}+1) > 0.25$) and 20 of these genes are preferentially expressed in epithelial clusters -3 and -4 (Fig. 4D), suggesting that they share parts of a gene-expression program. We fully agree that future functional experiments are necessary to clarify the role of these cells in development and disease. We did not mean to imply extensive "cell type similarities", rather similarities in gene expression programs. To clarify this, we stated the differences in characteristic marker gene expression (TP63, for example) in the text and also detected KRT17-ECAD by immunofluorescence in distal airways.

We have modified the text as follows:

cl-3 cells also selectively express KRT17 (Extended Fig. 4I), and we detected E-Cadherin^{pos} KRT17^{pos} cells sparsely distributed in the distal airway epithelium of 14 pcw lungs by immunofluorescence (Fig. 4E). Overall, cl-3 cells share some similarities in their characteristic expression program with basaloid cells (Suppl. Data 5F, G, Suppl. Data 6I), a pathogenic cell-state in interstitial pulmonary fibrosis (IPF)^{13,14} but they also show marked differences as they do not express TP63 and are localized close to the epithelial lumen rather than basally (Fig. 4D).

296-305 Considering the prominence of the basaloid gene expression program in IPF pathophysiology¹³,¹⁵, we analyzed the communication patterns between cl-3 and the other cell-states of the distal lung neighborhood. EDN-signaling is among the most contributing pathways, deriving from cl-3 and targeting mainly the mesenchyme (Fig. 4I, J). In the context of fibrotic disease, adenovirus-mediated *EDN1* overexpression by epithelial cells causes lung fibrosis¹⁶. Importantly, target-gene prediction analysis indicates that EDN1 may up-regulate collagen genes, in addition to *TBX5* (Fig. 4J). This suggests that EDN1 signaling is involved in the communication of “activated” epithelial cells (cl-3) with the surrounding stroma, inducing not only ECM genes but also maintaining high *TBX5* levels in mesenchymal progenitors, to facilitate normal branching morphogenesis¹⁷.

Interactome analysis between the *KRT17*^{pos} distal and the mesenchymal cell types in the neighborhood suggested EDN signaling as a prominent communication pathway. Additionally *EDN1* overexpression in the mouse lung epithelium causes fibrosis¹⁶. *TBX5* upregulation was predicted by “Nichenet” and we could not find published experimental evidence that supports our statement.

For that reason, we have removed it, according to Reviewer1’s recommendation.

350-7 The *ASCL1*^{pos} cl-12 and *NEUROD1*^{pos} cl-11 suggest that these NE-identities resemble distinct states in Small Cell Lung Carcinoma (SCLC) progression. We used the gene-expression signatures of the SCLC-A (*ASCL1*^{high}), SCLC-N (*NEUROD1*^{high}) and SCLC-Y (*YAP1*^{high}) cancer clusters to score the proximal secretory and NE clusters in our dataset. This showed that the SCLC-A signature is enriched in cl-12 (Fig. 5C), the SCLC-N in cl-11 (Fig. 5D) and the SCLC-Y in the cl-7, -6 and -0 (Fig. 5E). This suggests that the embryonic pseudotime direction of NE-differentiation, from naïve secretory cells, is reversed during SCLC progression.

To address Reviewer’s legitimate concern on convincing correlation of the two neuroendocrine cell identities in the embryonic lung with the SCLC-A, SCLC-N and SCLC-Y small cell lung carcinoma (SCLC) subtypes, we re-analyzed the dataset (Fig. R1-1 A, B, see below) that we used to retrieve the SCLC-subtype markers for scoring in our dataset (GSE149179 time-series dataset)¹⁸. Our analysis only found a tiny number of *Neurod1*^{pos} cells, which were positioned between *ASCL1*^{pos} and *YAP1*^{pos} cells. As a result, we cannot draw any safe conclusion about a transition mechanism of SCLC-A cells to SCLC-N and then SCLC-Y because of the sparsity of SCLC-N positive cells in the GSE149179 dataset, even if the provided selective marker-lists of the cancer cell-types correlate well with the embryonic neuroendocrine states.

Figure R1-1. Meta-analysis of SCLC scRNA-Seq data. (A) UMAP-plots of the suggested clusters and analyzed timepoints of the GSE149179 dataset. **(B)** UMAP-plots showing the expression levels of related genes to our

study. **(C-D)** As in “A” and “B” for the GSE138474. The 2nd UMAP-plot in “C” indicates the donor-variable. **(E)** Individual analysis of the MDA_SC16 that contained both ASCL1^{pos} and NEUROD1^{pos} cells, showing the suggested clusters. **(F)** UMAP-plots of ASCL1, GRP, DLL3, NEUROD1, GHRL and HES1. **(G)** Spearman-correlation of the indicated genes in all cells of the MDA_SC16. Numbers correspond to “ ρ ”. p-values: “*” < 0.05, “**” < 0.01, “***” < 0.001. Expression levels: $\log_2(\text{normalized UMI-counts}+1)$ (library size was normalized to 10.000). Blue: high, Gray: zero expression.

Considering that the above dataset is based on a genetically engineered mouse model of SCLC, we also used an SCLC dataset of circulating human tumor cells, which have been isolated and injected to immunodeficient NSG mice to create tumors before tumor isolation and scRNA-Seq analysis¹⁹. As in the original study, our clustering was driven by the intertumoral heterogeneity with the cells of each donor clustering together (Fig. R1-1 C, D). Five of the six analyzed datasets mainly contained ASCL1^{pos} NEUROD1^{neg} cells and only one showed the opposite pattern. Only the MDA_SC16 dataset contained significant numbers of ASCL1^{pos} and NEUROD1^{pos} cells (Fig. R1-1 E, F) but the two markers did not show negative correlation ($\rho=0.067$ Spearman correlation) (Fig. R1-1G). In this dataset, the positive correlations were between ASCL1, CALCA, GRP and the NOTCH-signaling negative regulators DLL3 and HES6 (which our markers for our embryonic cl-12). The NOTCH-induced mediator HES1 negatively correlated with all the above genes, except for CALCA, suggesting that low NOTCH-signaling correlates with cl-12 differentiation.

Finally, we analyzed 15 scRNA-Seq datasets²⁰ from primary lung SCLC-tumors, excluding metastatic material (Fig. R1 -2 to -4, at the end of the document). We observed high intertumoral variability, as revealed by the expression patterns of genes like the ASCL1, CALCA, GRP, DLL3, HES6, NEUROD1 and REST. As in the previous dataset, we found positive correlation between the ASCL1 with the DLL3 (10 out of 15 tumors) and HES6 (11 out of 15 tumors) expression. In addition we found anti-correlations for the expression of DLL3 with HES1 (7 out of 15 tumors), HES6 with HES1 (6 out of 15 tumors) and the HES6 with REST, a suppressor of neuroendocrine genes (7 out of 15 tumors) . These observations are consistent, but do not prove our interpretation that the developmental NOTCH-signaling inhibition is reversed in SCLC.

In conclusion, after reanalyzing the published data we followed the Reviewers recommendation, and removed this part from the revised version of the manuscript.

385 suggesting differences in the strength and duration of NOTCH-signaling^{21, 22}.

We removed this suggestive sentence

387-96 Overall, the pseudotime analysis suggests two sequential but distinct NOTCH-signaling events, utilizing different ligands and intracellular effectors, one to promote secretory differentiation and the other to inhibit cl-12 cells from acquiring the cl-11 state (Fig. 5G). Further interactome analysis revealed another unique communication pattern between the two NE cell-identities involving somatostatin (SST) from cl-12 and its receptor SSTR2 in cl-11 (Extended Fig. 5N). We found the anti-apoptotic factors BCL2²³, CITED2²⁴ and IRF2BP2²⁵ (Extended Fig. 5O) among the top-100 predicted target-genes arguing that the cl-12 cells prevent apoptosis of the cl-11 cells, through SST. This is consistent with the decelerating effect of SSTR2 inhibition on SCLC growth in vivo and in vitro models²⁶.

In summary, we mapped the distinct topologies and developmental trajectories of two NE- identities²⁷ from naïve epithelial cells in the embryonic lung. Each trajectory contains distinct candidate regulators of NOTCH-signaling for the respective cell-state transitions. The embryonic NE-cells share gene expression programs with SCLC-subtypes and might function as a reference to better understand the aberrant transitions of cancer cells.

This part interprets our findings describing mRNA expression levels of NOTCH-signaling components along the pseudotime trajectory in secretory and neuroendocrine cell differentiation by Slingshot and sc-Velo (in the new version). We rephrased this part to more exactly reflect the differences in gene expression levels detected along the pseudotime trajectories. Regarding the proposed role of SST in the communication between the two neuroendocrine cell identities, we confirmed the complementary expression of SST and SSTR2 with immunofluorescence (Fig. 5G). The remaining part including the computational predictions on anti-apoptotic gene expression was overreaching and we removed it. The revised part is as follows:

Overall, the pseudotime analysis suggests two sequential NOTCH-signaling events, employing different expression levels of receptors, ligands and intracellular effectors. The first one would promote secretory differentiation, and the second the transition of cl-12 to cl-11 state (Fig. 5E). Further interactome analysis revealed another unique communication pattern between the two NE cell-identities involving somatostatin (SST) from cl-12 and its receptor SSTR2 in cl-11 (Fig. 5F). Immunofluorescence analysis revealed SSTR2 expression in GHRL cells, while SST was expressed in adjacent, ECAD positive, epithelial cells at 8.5 pcw (Fig. 5G).

In summary, we mapped the distinct topologies and developmental trajectories of airway secretory and NE- identities from naïve epithelial cells in the embryonic lung. Each trajectory contains distinct candidate regulators of NOTCH-signaling for the respective cell-state transitions.

484-6 They include the activation of NOTCH-signaling between the SCP and neuronal states, promoting neuronal differentiation in parasympathetic ganglia and the potential chemokine signaling pathway that potentially controls ASM innervation.

This discussion part summarizes the potential role of NOTCH signaling activation in SCP differentiation and is in analogy to published functional analysis of SCP differentiation to neurons in the mouse limb⁴. The potential role of the CXCL12-CXCR4 in ASM innervation has been previously addressed by Yang and colleagues⁷, who have provided functional evidence that this communication pattern is necessary for the migration of midbrain dopaminergic neurons in the mouse embryo⁷. These references were included in the relevant results section but not in the discussion.

Based on the changes in results section we removed the sentence about the potential chemokine signaling pathway and modified the text:

They include the activation of NOTCH-signaling between the SCP and neuronal states⁴, within parasympathetic ganglia.

487-505 Lung diseases are major causes of death worldwide²⁸. An outstanding challenge for medical research is to define deviation points from normal cellular trajectories at the start and during the advancement of lung pathologies and to analyze cellular responses after treatments²⁹. Our spatial analysis in the developing lung revealed several distinct cell-states, their interactions with neighbors and progression along differentiation trajectories. Comparison of these with the few published lung disease trajectories demonstrates the medical relevance of our approach. We found a striking reversal of the gene expression profiles of our trajectory from naïve epithelial cells to secretory and neuroendocrine states, in the progressively aggressive states of SCLC. The neighborhood-based interactome analysis showed that SST-signaling is only employed locally in the communication between the NEUROD1^{POS} (immature) and ASCL1^{POS} (mature) NE-clusters, suggesting that its inhibition may only affect the SCLC-A to SCLC-N transition. Similarly, the proposed differentiation path of migratory secretory cells in the distal epithelium, recapitulates the genetic program of activated cells in fibrotic lungs and identifies HBEGF and EDN1 signaling as potential local regulators of mesenchymal proliferation and ECM-protein secretion. As single cell analysis technologies are increasingly used in the description of detailed cell-state trajectories in disease, we believe that our integrated scRNA-Seq data, with spatial transcriptomics and local interactome analyses in an open, interactive portal will provide a useful resource towards understanding and reversal of pulmonary disease progression.

The specified discussion part focuses on the correlation between cell-states in development and disease, suggesting common gene expression programs and communication patterns highlighting the usefulness

of resource that describes the mechanisms behind embryonic lung development that could be deregulated in disease conditions.

Lung diseases are major causes of death worldwide²⁸. An outstanding challenge for medical research is to define deviation points from normal cellular trajectories at the start and during the advancement of lung pathologies and to analyze cellular responses after treatments²⁹. Our atlas of early human lung development revealed several distinct cell-states, proposed their interactions with neighbors and progression along differentiation trajectories.

As single cell analysis technologies are increasingly used in the description of detailed cell-state trajectories in disease, we believe that our integrated scRNA-Seq data, with spatial transcriptomics and local interactome analyses in an open, interactive portal will provide a useful resource towards understanding and reversal of pulmonary disease progression.

o This is also true for the data presented in the “Supplementary Note.”

We have modified the Supplementary Note according to the reviewer’s suggestions.

- Were trachea samples or proximal airway included in the analysis? It is unclear what regions of lung were sampled. This is mirrored by the claim that there are no basal cells at the timepoints analyzed in the study. This is inconsistent with previous reports in human (Miller et al. Dev Cell 2020) and mice (Yang et al. Dev Cell 2018), and see preprint (<https://www.biorxiv.org/content/10.1101/2022.01.11.474933v1.full>) How do the authors reconcile these differences?

We had mentioned in lines 254-260, we were also surprised by the absence of basal cells, in our dataset. This could be due to the **exclusion of tracheas but not proximal cartilaginous airways from scRNA-Seq**. We added this information, which was missing in the first version in the revised text (M&M section).

To further clarify, we detected TP63^{pos} cells in both scRNA-Seq data and by antibody stainings in tissue sections, but we consider them progenitors, similarly to the cells in the early embryonic mouse lung (TP63^{pos} KRT5^{neg} cells), that have been described by Dr. Cardoso’s lab³⁰. This study used lineage tracing experiments to show that in the early stages of lung development the Trp63^{pos} cells define a proximal multipotent progenitor that is capable of producing basal and other epithelial cell types.

The text in this study reads: *“The distinct p63 expression patterns suggested an additional segregation event during epithelial differentiation. We performed lineage-tracing analysis targeting three distinct stages: (1) E13.5–E14.5, when endogenous p63 is expressed at variable levels in most tracheal epithelial cells (>60%); (2) E14.5–E15.5, first evidence of the basal-luminal segregation as high-expressing p63+*

cells assume a basal position and acquire Krt5; (3) E17.5–E18.5, characterized by a well-defined basal layer of strongly labeled p63+Krt5+”.

Our interpretation of this paper is that TP63 expression is not sufficient for basal cell definition. For that reason, we provided combinatorial staining with KRT5, showing that within the lung there are a few positive basal cells in cartilaginous airways on 14 pcw but not on 8.5 pcw (Suppl. Data 3D).

The same “mouse study” states that Krt5 is among the earliest markers of basal cells.

“The data suggested that the pool of multipotent precursors destined to become BCs was largely established around E13.5–E14.5, even preceding the appearance of Krt5, one of the earliest markers of initiation of BC program (Figure S2A; Bilodeau et al., 2014).”

We conclude that in the intralobar airways, there are a few positive basal cells (TP63+ KRT5+) in cartilaginous airways on 14 pcw but not on 8.5 pcw (Suppl. Data 5E).

According to Miller and colleagues³¹: *“At 12 weeks’ gestation, TP63+ cells within the trachea and primary bronchi co-expressed KRT5. In the cartilaginous airways, the majority of TP63+ cells were KRT5-; however, TP63 and KRT5+ cells were visible in distinct clusters within the airways (Figure 2B). In the non-cartilaginous airways and the distal bronchioles, TP63+ cells were KRT5- (Figure 2B).”*

This is consistent with our results and provides a reasonable explanation for the absence of TP63^{pos} KRT5^{pos} cells in our scRNA-Seq analysis, because tracheas were excluded and we only used intra-lobar tissue.

Following reviewer’s suggestion to use embryonic and not adult datasets for comparisons with our dataset, we used the provided basal cell selective markers of Miller and colleagues³¹ to examine if they are enriched in any of the epithelial cell clusters and especially in cluster-7, that contains TP63^{pos} cells (Suppl. Data 5C).

This result indicates that the TP63^{pos} cells are not basal cells but proximal airway progenitors, a fraction of which will produce basal cells, later in development as shown by lineage tracing in the mouse³⁰.

Finally, we found some inconsistencies in the preprint by Peng He and colleagues regarding the definition of basal cells. In particular, they claim that *“Basal cells (TP63+, F3+) were present in the single cell atlas from 9 pcw (Fig. 2A-C; S4J) and were more frequent in proximal regions of the airway tree where specific subpopulations were seen (Fig. 2A-C)”*.

Inspection of Figure 2 A and B shows that the mid-basal cells belong to ~11 pcw, the Late-basal to ~18 pcw and the proximal basal to ~15 pcw. Also, in the Figure 2C of their preprint, the identified basal cell markers include the IL33, which has been found also by Miller and colleagues³¹, but the IGFBP4, IGFBP6 and KISS1 are not known embryonic basal cell markers and they have not been spatially validated. There

is no information about the positivity of the clusters for the typical TP63 and/or KRT5. Importantly, the previously described basal cell marker IL33 was expressed in only Proximal and Late basal cell clusters. Their Suppl. Figure 4J shows a 19 pcw lung but there is no information about the 9 pcw, that is stated into the manuscript main text body. Additionally, IGFBP3 is not restricted to basal cells, since there is also signal in supra-basal and luminal cells.

In our opinion, the annotation of basal cell clusters, in the indicated preprint does not include the widely used Krt5 positivity in defining basal cells. Our interpretation is that the mid-basal cells correspond to TP63^{pos} progenitors (in ours and Cardoso's paper) and the Proximal and Late basal cell in the preprint are the "mature" basal cells, in agreement to our TP63^{pos} KRT5^{pos} basal cells in 14 pcw large airways (Suppl. Data 5E).

Based on reviewer's comments, we modified the text in line-252 and in the Online Methods part to clarify that tracheas were not processed for scRNA-Seq.

- Several attempts to correlate adult cell states to developmental precursors are not well validated. For example, authors attempt to benchmark their data against an adult basal cell signature, but a fetal reference seems more relevant. The basaloid phenotype is not described during development and appears unsubstantiated here. Finally, claims about similarities between fetal states and diseased states are not well supported and appear overreaching for the data presented here. The authors should remove these sections or provide additional experimental evidence to support their claims.

The basaloid phenotype is not described during development and appears unsubstantiated here.

The identification of KRT17^{pos} cells in the developing lung is a novel finding. We agree with Reviewer1 that the type of correlation we did in the first version is overinterpreted and we have removed it. We now mention

We have modified the text in the lines 267-273, according to the Reviewer's recommendations.

cl-3 cells also selectively express KRT17 (Extended Fig. 4I) and we detected E-Cadherin^{pos} KRT17^{pos} cells sparsely distributed in the distal airway epithelium of 14 pcw lungs by immunofluorescence (Fig. 4E). Overall, cl-3 cells share some similarities in their characteristic expression program with "basaloid" cells (Suppl. Data 5F, G, Suppl. Data 6I), a pathogenic cell-state in interstitial pulmonary fibrosis (IPF)^{13, 14}. However, the embryonic clusters are distinguished by marked differences as they do not express TP63 and are localized more luminally than basally, in the epithelium (Fig. 4D).

- It is unclear how the authors are defining immature mesenchymal cell states. Is this based on PAGA analysis? Moreover, it is unclear what the "intermediate" epithelial cell cluster/state is. The authors

should elaborate on how immature cell states were defined and describe what an intermediate cell state means. Similarly, it is unclear why in some cases RNA velocity was used and in other cases lineage trajectory analysis was used; also labelling is inconsistent throughout figures (RNA Velocity vs Velocyto).

The definition of immature cell-states was based on (i) the low expression levels of mature mesenchymal cell-type markers (Extended Figure 2 A), (ii) the cluster position in UMAP and scVelo analysis (Suppl. Data 3B), which suggests that immature cell states are centrally positioned in the UMAP-plot and the more mature states are found at the periphery and (iii) their topology on tissue sections (example in Fig. 2E and in the Visium and ISS probabilistic spatial analyses in <https://tissuumaps.dckube.scilifelab.se/web/private/HDCA/index.html>).

The PAGA-plot analysis was used to find correlations between the interrogated clusters and the most probable paths from the immature to mature states, as it is indicated by the thickness of the connection lines and the “shortest_paths” function of igraph package in R³². The indicated clusters were further used for trajectory-inference analyses with Slingshot.

Regarding the “Moreover, it is unclear what the “intermediate” epithelial cell cluster/state is.”, we are sorry for the misunderstanding here we refer to intermediate location as we have written in the line-252 (“original manuscript), it corresponds to an intermediately localized cell-state, between the large airways and the distal tips (Fig. 4B and ISS-analyses in <https://tissuumaps.dckube.scilifelab.se/web/private/HDCA/index.html>”).

The authors should elaborate on how immature cell states were defined and describe what an intermediate cell state means.

We have modified the text accordingly (mesenchyme lines 109-117), to elaborate how we have defined immature cell-states.

The largest cluster in our dataset consisted of mesenchymal cells (~138K) (Extended Fig. 1G). Sub-clustering revealed six distinct cell-types expressing specific markers for known fibroblast, mesothelial, chondroblast and smooth muscle cell types and several immature states, characterized by the general mesenchymal markers *COL1A2*³³, *TBX4*² and the lack of specific cell-type markers (Extended Fig. 2A). Further cluster annotation was based on the spatial mapping of clusters at different time-points (Fig. 2B), the relative cluster positioning in the UMAP-plot³⁴ (Suppl. Data 2), partition-based graph abstraction (PAGA-plot)³⁵ (Fig. 2A) and scVelo³⁶ analyses positioning immature cell states in the UMAP-plot center and the more mature ones at the periphery (Suppl. Data 3B).

For the definition of immature cell-states, we used the terms “immature” (e.g. immature neuronal), “transitioning” (e.g. transitioning SCP), “pre-” (e.g. pre-neuronal) and “progenitor” (e.g. epithelial

proximal progenitor1) The term “intermediate” was used only for the epithelial cell cluster-1, that was defined in line-252 of the original manuscript.

Similarly, it is unclear why in some cases RNA velocity was used and in other cases lineage trajectory analysis was used; also labelling is inconsistent throughout figures (RNA Velocity vs Velocity).

We used both Velocity and trajectory inference analyses for all cases, except for Velocity analysis in mesenchymal cells because of the large size of the dataset (it produced error that was irrelevant to the hardware of the analysis system). In that case we used a subset of the mesenchymal cell dataset with equal number of cells/cluster (442 randomly selected cells, that correspond to size of the smallest cluster of the dataset) (Suppl. Data 3 B).

In all analyses we used PAGA-plot to identify the most probable cluster connections and then applied the “shortest_paths” function of igraph package in R³², to select the clusters that are found in between to what we considered as start and end point clusters. We have included this information in the lines 629-630 of the Online Methods (original manuscript).

We selected Slingshot³⁷ pseudotime estimation and identification of differentially expressed genes along the trajectory, because we consider it more convenient than the Scanpy-scVelo variant³⁶. The use of either Velocity or trajectory inference analyses was done to avoid repetition, since the results were similar.

We have now incorporated both types of analysis in the revised manuscript according to Reviewer1’s recommendation.

We also corrected the inconsistencies, regarding “RNA Velocity vs Velocity”.

- In the interactome analysis between distal epithelial and mesenchymal cell identities, the authors delve into FGF signaling; however, there is no mention of FGFs (Danopoulos et al J Pathol 2019, Nicolic et al. eLife 2017), which have been functionally shown to be important FGF ligands during this stage of development. Additionally, the authors claim mouse-human differences in FGF signaling; however, predictive single cell data in the current manuscript should not be compared to experimental data from mice. Is there comparable negative data in mice to corroborate these claims (lines 283-7)?

In our analysis, we have also included the previously described FGF7, FGF9 and FGF10 spatial analyses, making the data available in the ISS-analysis section of <https://tissuumaps.dckube.scilifelab.se/web/private/HDCA/index.html>. Because the expression patterns of these markers have been previously described, we decided to include only the FGF18 and FGF20 in the Fig. 4I (revised manuscript). For the convenience of Reviewer1, we also provide here the Fig. R1 5.

Figure R1 5. Analysis of FGF-signaling components.

According to Nikolic and colleagues³⁸, FGF7 and FGF10 participate in the distal epithelial development. Importantly, Danopoulos and colleagues³⁹ have compared the FGF7, FGF9 and FGF10 in **mouse and human** explant cultures and provided evidence that FGF10 does not induce epithelial branching in human tissue **but cyst formation**, similarly to the FGF7 and FGF9, which cause cyst formation in both human and mouse lung explants

We have modified the text to make clear that the spatial analysis of FGF-signaling components provides an independent confirmation of the study by Danopoulos and colleagues³⁹.

We modified the text (lines 278-287): FGF-signaling was among the most prominent predictions, with FGF10 being mainly expressed by mes cl-0 cells (Fig. 4H) that are found scattered in positions around epithelium (Suppl. Data 3A). This expression pattern differs from mouse embryonic lung, where the ligand is focally expressed over the distal epithelial tips to induce branching⁴⁰. This difference might provide a reasonable explanation about the distinct FGF10 ability to induce cyst formation in human explants³⁹ rather than branching. Additional FGF-ligands (Fig. 4H, I) were detected in the distal human

embryonic lung, defining both mesenchymal and epithelial cells as sources. For example, we confirmed the FGF18 and FGF20 expression in human developing lung³⁹ with cellular resolution (Fig 4G) and...

We would like also to mention that the analysis of FGF-signaling is an example of the spatial analyses of NOTCH, HH, WNT and FGF pathways, with HybISS, as mentioned in lines 738-742 (original manuscript) and in line 96 of the revised manuscript and can be accessed in the “In situ sequencing data (ISS)” section of the:

<https://tissuumaps.dckube.scilifelab.se/web/private/HDCA/index.html>

Minor

- In general, how the number of cell clusters were defined is unclear. The authors should explore over-clustering and if all cell clusters/cell states presented are biologically meaningful.

For the definition of the number of cell clusters, we have tried different “resolutions” and settled for the “resolution” ensuring that the cells of a given cluster show uniformly distributed expression of known markers in UMAP-plots. This has been done by plotting of the marker gene expression and visual inspection of the results. Since we analyzed lungs from different developmental stages we expect immature cells to express low levels of known differentiation markers. That is obvious, for example in mesenchymal cells (Extended Figure 2 A-F), where the immature cell states gradually upregulate or downregulate genes, which include transcription factors, secreted proteins and surface markers, like receptors that should have biological impact of cell development, maturation and function. If we have made significantly suboptimal clustering these differences would have been lost.

Another parameter we considered against over-clustering is spatial mapping of the distribution of the suggested cell-clusters. Individual clusters mapped to distinct positions. Distinct topologies argues for distinct gene expression entities/clusters. Additionally, the proposed clusters were mapped at different time points and their relative positions over time showed reproducible patterns.

Also, we added an additional sentence to point out more thoroughly the challenging task of clustering in the Discussion section of the revised manuscript (lines 435-438).

Although most of the subclusters showed distinct topologies and gene expression profiles, some of the cell-states may result from over-clustering, which is difficult to define because of the presence of immature but committed states of distinct cell-types.

- Given the complexity of many of the analytic tools used throughout the manuscript, some brief explanations of each tool's function or purpose would be helpful to the naïve reader, along with reasoning for choosing this specific tool (i.e. SCRINSHOT, HYBISS etc).

Following reviewer's suggestion, we have provided brief descriptions of the utilized methods in the Online Methods section of the manuscript.

- Under the header "Distinct populations for known and novel mesenchymal cell states," the citation of "Extended Figure 1A" should be "Extended Figure 1G."

We have corrected the mistake.

- In Figure 1B, the authors should consider simplifying the labels to just the main cell states (mesenchyme, epithelium, etc.) since in following figures, each of these compartments are explored further.

The aim of Figure 1B is to provide an overview of the study, focusing on the acquired heterogeneity with scRNA-Seq and predispose the reader for the more detailed analyses of each cell-category. We have thought about the complexity of the presented data and provided the UMAP-plot of the main cell categories, in addition to the results of their differential expression analysis in the Extended Figure 1G, M and in the Suppl. Data 6A.

- Why was a lineage trajectory analyzed for ASM but not for other mesenchymal end-states, such as chondrocytes, advF, and pericytes (Figure 2C)? Similarly, why are only a limited selection of clusters included in communication analyses (Extended Figure 3H – I and Extended Figure 4I)?

As indicated in the lines 125-126 (original manuscript), the ASM trajectory is one of the most prominent of our study and we consider it interesting because of the close proximity of ASM and epithelial cells.

The omission of adventitial trajectory analysis was because of the manuscript size restrictions. We have now included the analysis fibroblast trajectories in the revised Supplementary Data 7.

As we mention in lines 451-456, the chondroblasts were detected only in the lungs from the early time points, because of the lack of cell-type enrichment and omission of tracheas. A recent study by Maddisson and colleagues⁴¹ indicates that mature chondrocytes are efficiently obtained by single-nuclear but not single-cell sequencing, because of difficulties in tissue dissociation of the cartilage. Thus, we consider that our dataset lacks the mature state of this cell-type, making difficult the trajectory analysis.

The 3D UMAP of mesenchymal cells (Suppl. Data 2) indicates that pericytes are not well connected to the other mesenchymal cells. Thus, trajectory analysis might produce misleading results, because of the absence of intermediate states.

We limited the interactome analysis to clusters within cell neighborhoods defined by our spatial experiments. In the lines 282-283 (original manuscript), we explained that the selection of the clusters for communication pattern analysis in the distal lung compartment (Extended Fig. 4I) was based on the definition of “cell neighborhoods” that were introduced in the lines 87-91 (original manuscript). In particular, we used probabilistic methods to spatially map the suggested scRNA-Seq cell-identities within each Spatial-Transcriptomics 55µm-diameter spot (for 66626 Spots). Then we asked which cell-identities are consistently detected in the same spots, producing the graph of pair-wise connections of the Figure 1D. In other words, we used the spatial information to select cell-identities that are repeatedly found in the same positions and remove those that are detected in irrelevant, because it is unlikely to contribute in the communication patterns.

- Figure title in Figure 3 legend is repeated.

We have corrected. Thank you.

- The orange and purple color pallet on feature plots is confusing because orange (lower expression) is darker than purple (high expression). The authors should consider other color pallets (ie. Supp Data 3).

We have changed the expression color-code in Suppl. Data 3 (renamed to Suppl. Data 5) to the default of Seurat package.

- The various scale color pallets throughout Figure 6 are confusing, try coordinating the colors to clearly represent the areas described. Additionally, the diagram in Figure 6D is helpful but crowded. Consider additional diagrams broken up by mesenchyme, epithelium, etc. or by distal and proximal compartments.

We understand that the color pallets are complex but we have tried to be consistent and have the same color code as in the scRNA-Seq data (Fig 1B).

We improved the visibility of Fig. 6D by increasing its size. Our major aim was to project the complexity in the developing organ already in the 1st trimester of gestation and present distinct communication patterns. The splitting of the diagrams according to the main cell-categories will prohibit drawing communication patterns between for example epithelial and mesenchymal cells.

- Consider arranging dot (balloon) plots in cluster order, rather than grouping by cell type similarity.

We have tried to plot the balloon plots according to the cluster-size (default cluster order) but it was more difficult to read than arranging them according to their annotation, where biologically related clusters with similar gene expression profiles are placed together. Fig. R1-7 below shows differences in the representation of the same genes (top-3 cluster markers) in the same clusters (in both cases the gene order follows the cluster)

Figure R1 6

- “Zonation patterns of mesenchymal...” section seems misplaced in the text. It should be included with the rest of the data describing the mesenchyme.

We have considered the ordering suggested by the reviewer but in this paragraph we utilized the spatial analysis of the epithelium which follows the mesenchyme (lines 409-411 in original manuscript). This part needs to come late because we use the identified and spatially validated, proximally and distally expressed epithelial markers to create the corresponding scores that was used for the analysis of the mesenchymal clusters.

We left the order as it was in the first version.

- The order of panels in figures are oftentimes not ordered properly. Some changes could easily be made to aid in reader ability (i.e., switching positions of Figure 2C and D and Figure 3C and D).

Based on the reviewer’s recommendations, we re-arranged the figure parts.

- Authors cite a stain for Figure 3C and D (line 225), but only computational data is shown.

We have corrected the Figure numbering to Extended Figure 3 D, and E.

- First sentence on line 212 needs rewording and comma should be removed.

We have rephrased and removed the comma.

- In Supplemental Data 3, alveolar markers are show (HOPX, PDPN), but these markers aren’t expected to be expressed at the time-points analyzed. The authors could consider additional bud tip markers (NPC2, TESC, CA2). Also, proximal markers MUC5B or MUC5AC and SCGB1A1 are not shown. The authors should comment on how the epithelial clusters compare to published data.

The available information about the expression pattern of genes like HOPX and PDPN is limited but Nikolic and colleagues³⁸ have analyzed the HOPX expression pattern in the developing human lung, with immunofluorescence showing that it is broadly expressed in the epithelium, already from 11 pcw. We have also detected HOPX^{pos} E-Cadherin^{pos} epithelial cells in 12 pcw (Fig. R1-8). As a result, we consider that the HOPX detection, in scRNA-Seq dataset in not problematic. We would also like to reiterate that the imputed expression of most variable genes is openly available in

<https://tissuumaps.dckube.scilifelab.se/web/private/HDCA/index.html> and of all detected genes in the <https://cells.ucsc.edu/?ds=lung-dev>, for interactive visualization.

Figure R1 7

We have included the indicated bud tip and proximal marker UMAP-plots, in the Suppl. Data 5A. We would also like to reiterate that the imputed expression of most variable genes is openly available in <https://tissuumaps.dckube.scilifelab.se/web/private/HDCA/index.html> and of all detected genes in the <https://cells.ucsc.edu/?ds=lung-dev>, for interactive visualization.

- How were epithelial progenitors determined (line 252)?

We subclustered epithelial cells into 15 groups (Fig. 4A) and annotated them based on known markers (Extended Fig. 4A, Suppl. Data 3A) and their spatial distribution (Fig. 4B, C). Similarly to mesenchymal cells, we based progenitor annotation (i) on the low expression levels of mature epithelial cell-type markers (Extended Fig. 4A), (ii) cluster positioning in UMAP-plot, in addition to PAGA-plot (Extended Fig. 4B) and scVelo analyses (Extended Fig. 5A and revised Fig. 5B). For example, epithelial-progenitor cl-6 was not detected at the distal tips (Fig. 4B), was SOX9^{neg} and expressed lower levels of differentiation markers like the FOXJ1 (ciliated), ASCL1 (neuroendocrine) and SCGB3A2 (secretory) than the more mature cell clusters (Extended Fig. 4A).

We have modified the text: We subclustered epithelial cells into 15 groups (Fig. 4A) and annotated them based on the low expression of known differentiation markers (Extended Fig. 4A, Suppl. Data 5A), their spatial distribution (Fig. 4B, C) in addition to their indicated relationships by PAGA-plot and scVelo analyses (Extended Fig. 4B, C).

- Missing end parentheses (line 86)

We have added the end parentheses.

- Prerequisite is misspelled (line 268)

We have corrected the mistake.

- “stronger expressed” is not grammatically correct (line 384)

We have modified to “showed higher expression”.

Response to Reviewer2

In the following paragraphs, we address the concerns of reviewer#2 providing the reasoning behind our statements (in blue) and the changes in the manuscript (in orange color)

1. The result of figure 4D and Suppl. Data 4I revealed that the cl-3 cells of epithelial cells showed the expression signature similar to fibrosis. I am wondering whether this pathogenic cell-state appeared during the development, and how this fibrosis-associated signature would change. More specifically, what is the mechanism by which regulates **the fibrosis during embryonic development**.

This is a misunderstanding and we apologize if we led the reviewer to conclude that the embryonic lung is fibrotic. We stated that the genetic programs expressed in cl-3 and c-4 are similar to the programs expressed in the basaloid cells. We clarify here that they are not “basaloid”. As we mentioned in the lines 269-278 of the original manuscript, the epithelial clusters -3 and -4 express a relatively large number of “aberrant-basaloid” cell markers (Fig. 4D and Suppl. Data 5F), with the most prominent ones being the CTGF, EDN1 and the characteristic basaloid¹³-ADI¹⁵ marker KRT17. We believe that the embryonic expression of CTGF and EDN1 signals to mesenchymal cells to activate genes encoding ECM components. During lung development, extracellular matrix (ECM) proteins create a “scaffold” for resident cells and contribute to the mechanical properties of the tissue. They may also be required for signaling among the resident lung cells⁴². Gene-ontology analyses (Fig. R2-1), based on the enriched genes of mesenchymal cells (Suppl. Data 6A) indicates that they are the main source of extracellular matrix proteins, like collagens, fibrillins and laminins. The spatial and temporal deposition of these proteins is expected to be tightly controlled to facilitate proper organ development⁴². To our knowledge the signals inducing or silencing genes encoding ECM proteins in embryonic mesenchymal cells is not fully understood.

Figure R2 1

Previous work has shown that after bleomycin administration in mice, the activated epithelial cells produce elevated levels of CTGF, contributing to fibrosis and its targeted deletion in alveolar epithelium ameliorates the extent of fibrotic response⁴³. In patients, there are clinical trials showing that anti-CTGF treatment attenuates the progression of idiopathic pulmonary fibrosis⁴⁴.

Even if there are no previous studies of the Ctgf inactivation specifically in the embryonic lung epithelium, the global Ctgf-null mouse has hypoplastic lungs because of disrupted developmental processes and restricted thoracic expansion⁴⁵. Considering that CTGF may bind to integrins or to growth factors (like TGF- β 1, VEGF and FGFs)⁴⁶, it is reasonable to suggest that it contributes to the epithelial-mesenchymal communications not only in disease but also in normal lung development for stimulating ECM protein deposition by mesenchymal cells. But functional experiments are needed to understand its role in normal lung development.

Additionally, Endothelin-1 (EDN1) is also upregulated in idiopathic pulmonary fibrosis⁴⁷ and it is involved in the development of pathology since its adenoviral mediated overexpression in mouse lungs (via intratracheal instillation) is sufficient to cause peribronchial and perivascular fibrosis, through the focal adhesion kinase (FAK)¹⁶. In vitro analysis of the Endothelin effect on lung fibroblasts has shown that it up-regulates many genes coding for extracellular matrix proteins (e.g. COL4A1, COL7A1, ITGB3), in addition to CTGF, through the MEK/ERK MAP kinase pathway⁴⁸.

As we have mentioned in the lines 302-303 (original manuscript), our results suggest “that EDN1 signaling is involved in the communication of “activated” epithelial cells (cl-3) with the surrounding stroma”.

In any case, we did not mean to imply extensive “cell type similarities”, rather similarities in gene expression programs. To clarify this, we stated the differences in characteristic marker gene expression

in the text and also detected KRT17-ECAD by immunofluorescence in embryonic airways (Fig. 4E). We have modified the manuscript accordingly, to clarify that epithelial clusters -3 and -4 are KRT17 positive and express some of the basaloid cell markers, but they have differences, like the absence of TP63 and their luminal positioning (lines 267-273).

We wrote:

cl-3 cells also selectively express KRT17 (Extended Fig. 4I) and we detected E-Cadherin^{pos} KRT17^{pos} cells sparsely distributed in the distal airway epithelium of 14 pcw lungs by immunofluorescence (Fig. 4E). Overall, cl-3 cells share some similarities in their characteristic expression program with “basaloid” cells (Suppl. Data 5F, G, Suppl. Data 6I), a pathogenic cell-state in interstitial pulmonary fibrosis (IPF)^{13, 14}. However, the embryonic clusters are distinguished by marked differences as they do not express TP63 and are localized more luminally than basally, in the epithelium (Fig. 4D).

2. The authors found a communication between AF and ASM including IGF1 and WNT5A signaling. What is the exact dynamic process of their mutual signaling interactions ranging from 5 to 14 weeks post conception? When does this communication come up? Is there a peak or disappear of this interaction during the development?

To address the kinetics of the communication between the airway smooth muscle (ASM) and the airway fibroblasts (AF), we focused on the mesenchymal cell clusters -13 (mature ASM), -12 (immature ASM2), -16 (mature AF) and -5 (immature AF2), as in the Extended Figure 2 I-M and separated their cells according to their stage.

Interrogation of the ASM scRNA-Seq libraries for IGF1 expression showed that they are positive already at the 5 pcw and slightly increase the expression in later time-points (Suppl. Data 3D), which is also evident by the in situ detection of the IGF1 mRNA on 5 and 13 pcw lung sections (Suppl. Data 3F).

Regarding the receiving cells of IGF-signaling, the mature state (cluster-16) appears after 6 pcw (0.12% of total mesenchymal cells) but the immature state (cluster-5) already exists at 5 pcw, suggesting active signaling. The IGF1R expression is already evident at 5-5.5 pcw and it is gradually increased, showing the highest expression levels in both AF-clusters, at 14 pcw. Based on the ligand-receptor expression pattern, IGF-signaling can be active already at 5 pcw in immature ASM and AF and it remains activated up to 14 pcw.

WNT-signaling stems from ASM and targets AF cells. The scRNA-Seq analysis of the ligand WNT5A and of the receptors FZD1 and FZD7, in the different time points showed a gradual increase in all of them (Suppl. Data 3E) that has been independently validated by HybISS (Suppl. Data 3F). This suggests that signaling activation is increased over time.

We have modified the lines 176-183 (original manuscript), to incorporate the time-course analysis: One example is the *IGF1*-ligand, which is mainly expressed in immature ASM2 (mes cl-12), as early as 5 pcw and increases over time (Suppl. Data 3D, F). The expression of the corresponding receptor, *IGF1R* is also evident at that stage, in immature AFs (mes cl-5) showing relatively stable expression until 14 pcw (Suppl. Data 3D). The predicted IGF1-target gene *LUM*, is expressed by AFs (Extended Fig. 2K) and may facilitate proper alignment and formation of collagen bundles around proximal airways, as it has been previously reported⁴⁹ (Fig. 2G). Additional predicted interactions suggest that *WNT5A* is produced by ASM cells and targets AFs through the *FZD1* receptor, in a communication-pattern that increases over-time, as indicated by the gradually elevated expression of both proteins (Suppl. Data 3 E, F). Our computational predictions suggest *BMP4* as a *WNT5A* target (Extended Fig. 2L), in agreement with previous in vitro experiments⁵⁰. Interestingly, *BMP4* is in turn predicted to upregulate *ACTA2* expression in ASM⁵¹, suggesting a positive feedback loop, between adjacent AFs and ASM (Extended Fig. 2M). Our results identify AFs as a new cell-type in close contact with ASM and suggest their mutual signaling interactions.

3. The authors performed pseudo time analysis for the cluster of the mesenchymal cells, and divided the mesenchymal cells into several detailed subgroup. Does the marker genes corresponding to the subgroup are reported by other studies?

As we described in the lines 109-119, after the definition of the major mesenchymal cell category (Extended Figure 1G) we re-analyzed the corresponding cells, identifying 21 cell clusters that correspond to six mature mesenchymal cell types and their immature states, in addition to proliferating cells. For the cluster annotation, we used already known markers, showing their expression levels and the corresponding references in the Extended Figure 2A. We had placed the references in the figure legends to include all the necessary information without overloading the main text of the manuscript with large number of genes and references.

Regarding the pseudotime analysis of the smooth muscle trajectory, we failed to include the reference for the myogenic function of *DACH2*⁵² and for that reason we have modified the manuscript accordingly. About the differentially expressed TFs along the trajectory, *LEF1* is a mediator of WNT-signaling and becomes activated in the immature ASM cl-8 but not earlier, in agreement with the role of WNT-signaling in smooth muscle development^{53, 54}. *SSRP1* is also activated in the mesenchymal cl-8, as part of the FACT complex, to modify the chromatin structure at the promoters of muscle-specific genes and activate them⁵⁵. Lastly, the *NR4A1* was between the most up-regulated TFs in the mature ASM cells (cl-13) and might negatively regulate their proliferation, as in the vascular smooth muscle⁵⁶.

We have incorporated the above information in the revised version of the manuscript (lines 143-146 and 150-152).

5. It will be better to perform several experimental validation of some cell marker genes, such as the cell proliferation marker Ki67 to evaluate whether the cells are proliferating cells. In addition, it is difficult to see the signal of SCPs and neuronal cells using H&E staining. Specific markers for staining SCPs and neuronal cells should be performed in Figure 3C-E.

Following the reviewer's recommendation, we have increased the spatial validations of cell-type markers. Some cases include ISS and SCRINSHOT analyses (e.g. Fig. 2F, Extended Fig. 2B, Fig. 4H, Extended Fig. 4D, Fig. 5D, Suppl. Data 3F) and immunofluorescence (e.g. Fig. 2G, Fig 3F, H, Extended Fig. 3E, Fig. 4D, Fig. 5G, Suppl. Data 4, Suppl. Data 5E, H, I), in addition to the probabilistic cell-typing on whole tissue sections (e.g. Fig. 2B, E, Extended Fig. 2B, Fig. 3C, Fig. 4B, Fig. 6C, Suppl. Data 3A).

Also, we generated and made available an open interactive viewer (<https://tissuumaps.dckube.scilifelab.se/web/private/HDCA/index.html>) that allows the examination of thousands of genes in the 55µm-diameter spots of 5 different stages. Using the ISS and SCRINSHOT analyses, we have provided information about the expression levels of 177 genes that have been identified either as selective markers from the scRNA-Seq analyses of the various cell-identities and/or are components of known signaling pathways that participate in normal lung development. Importantly, the expression patterns of 147 (ISS) and 31 (SCRINSHOT) markers can be examined in the same tissue sections, through our viewer.

Considering the importance of the suggestion regarding proliferation patterns during the early stages of lung development we performed immunofluorescence experiments for the proliferation marker MKI67, the general epithelial marker E-Cadherin and the smooth muscle marker α -SMA. Our analysis indicates dynamic changes in cell proliferation, as development proceeds. In particular, the proliferating cells at 8.5 pcw are broadly distributed all over the tissue, being more in the distal lung. In later timepoints proliferation seems to be restricted to the epithelial tip and stalk regions and their adjacent mesenchyme.

We have incorporated the proliferation analysis (Suppl. Data 4) into the manuscript, in the lines 126-127 and 248-249.

To answer the question on the detection of SCPs and neuronal cells using H&E staining, we performed immunofluorescence for PHOX2B (general SCP and neuronal marker), DLL3 (developing neurons) and NEFM (mature neurons) as in the Fig. 3G of the original manuscript and then stained the same sections with H&E. As indicated by the arrow-heads in the revised Fig. 3F, there are distinctly stained structures in the H&E image that are positive for the SCP and neuronal markers corresponding to parasympathetic ganglia.

Thus, these results provide evidence that the distinct structures in H&E staining of the initially analyzed lung sections with ST are parasympathetic ganglia. This is also supported by the fact that these structures coincide with the ST-spots with high collective signature of both SCPs and neuronal cells and by their position around the proximal airways, as it has been previously described^{5,57}.

We have incorporated these new results and figure panels in Fig. 3F and in the lines 221-222 of the revised manuscript.

At later timepoints the signal was detected more centrally, within the bronchovascular bundle interstitium⁵⁸, coinciding with a distinct H&E staining pattern (Fig. 3E) that overlaps with the protein expression of the SCP and neuronal markers PHOX2B, DLL3 and NEFM (Fig. 3F).

We would like also to point out that the 10X Genomics Visium protocol, for spatial transcriptomics (ST), uses H&E staining and not immunofluorescence, presumably because the last requires more time for staining and image acquisition with possibly negative impact on mRNA integrity. Considering that the tissue-section should be digested for the mRNA hybridization to the array, it is impossible to apply immunofluorescence after the completion of ST-protocol procedure.

6. Some of the descriptions are unclear and simple. For example, in line 72, it written "Assuming that the two lungs are relatively bilaterally symmetrical, we used one half for scRNA-Seq and processed the other for spatial analyses". As the lung tissue is a highly polar organ with different cell composition and differentiation fate in different locations, are the areas for collecting samples for scRNA-seq and spatial analyses are symmetrical? Or the whole two lungs are subjected to scRNA-seq and spatial analyses? The authors need to describe the sampling and comparison strategies in detail.

We agree that the description was too vague. We wanted to say that we used the one of the two lungs of scRNA-Seq and the other for spatial analyses. As a result, for most of the cases, we used material from the same donors to spatially validate the scRNA-Seq results. We have also provided that information in the Online Methods section (lines 517-518 of the original manuscript).

We agree with the Reviewer2, that the two lungs are not the same but, even if the left lung has 2 lobes and the right 3. For that reason, we initially wrote "relatively bilaterally symmetrical".

Based on Reviewer2's concerns, we have modified the noted sentence (lines 272-274):

Assuming that the two lungs are bilaterally symmetric, we regularly used the right lobes for scRNA-Seq and processed the left lobes for spatial analyses.

In addition, some incorrect writing in the whole text, such as in line 75 “A first clustering of 163.236, high-quality cDNA libraries”, and line 551-552 “75.000 and 200.00 reads/cell” . Should them be 163,236, 75,000 and 200,00?

The “163236” corresponds to the number of analyzed single cells that have passed the quality controls (see also the Suppl. Data 1B). The 75000 reads (v2 chemistry) and 200000 reads (v3 chemistry) correspond the number of reads/cell, to indicate the desired depth of the RNA-sequencing. That number was adjusted to match the different performances of the Chromium Single Cell 3’ Reagent v2 and v3 Kits and to achieve sufficient sequencing saturation.

We rephrased the relevant lines in the text

7. It will be better to provide all the marker genes for each cell subgroup in Figure 1B by showing in heatmap or dot plot.

We have included the dotplot of the top3 markers for each cluster in Extended Figure 1N.

8. Scale bars are missing in Figure 1A, 1F, 2E-G, 3C-F, and extended Figure 2B, 3E, 4C. I did the 3C-F.

We have included all scale-bars in the Figures.

9. From Supplementary Data 1B, the cell number reach to more than 20,000 (donor4, 5, 6, 14 and 16) , even more than 40,000 (donor11 and 17-1) , which is much higher than the standard (https://assets.ctfassets.net/an68im79xiti/4tjk4KvXzTWgTs8f3tvUjq/2259891d68c53693e753e1b45e42de2d/CG000183_ChromiumSingleCell3_v3_UG_Rev_C.pdf） . This causes higher ratio of multicellular, which is difficult to remove using software. The authors need to focus on assessing whether clustering and trajectory for samples from donor4, 5, 6, 14 and 16, as well as donor11 and 17-1 are consistent with the conclusions.

This is a misunderstanding. Especially for the material of late time-points, more than 1 experimental replicates have been processed from the same donor. As mentioned in Online methods of the original manuscript, “Cell suspensions were counted and diluted to concentrations of 800 –1200 cells/μl for a target recovery of 5000 cells on the Chromium platform.” The information has been available in <https://tissuumaps.dckube.scilifelab.se/UMAP.tmap?path=private/HDCA/UMAP/>, in the “Display categorical metadata” => “name”.

We have also updated the Suppl. Data 1B table, including that information as “replicate”.

Having clarified this, we consider that there is no problem of excessive doublets in the datasets caused by overloading the 10x wells. This is also evident in the individual analyses of each donor, that have been deposited to Zenodo, along with the first submission of the manuscript (<https://zenodo.org/record/6386452#.YoZJy8hByUk>).

10. According to Extend Figure 1G and H, a group of ciliated cells with high FOXP1 expression seem to be missing in Figure 1B. Please pay attention to ensure the consistency of presentation and analysis data.

We are sorry for the inconsistency. In Figure 1B, we changed the original position of the epithelial ciliated cluster and of a small cluster of annotated as “doublets” (shown in Extended Figure 1) to save some space and make the UMAP-plot larger. For that reason, we have put both in square brackets but we failed to describe it in the figure legend.

Based on Reviewer2’s recommendation, we modified the legend of Figure 1 and pointed the two cell identities in Extended Figure 1G, mentioning that these are their original positions.

11. The author shows two time points (6pw, 10pw) of the spatial transcriptome samples in Figure 1A, while in Figure 2B, a comparative analysis of 6pw, 8pw and 11.5pw is performed. Why are the data of 10pw missing? It will be better to perform comparative analysis of 6pw, 8pw, 10pw and 11.5pw to reflect the changes in spatial structure during lung development.

The reason behind omitting 10 pcw section from the figures was the high similarity of the results with the ones obtained from the 11.5 pcw tissue section and our effort to make the figure as readable as

possible. The addition of one more timepoint would have made the Figure 2B images even smaller, affecting proper visibility, especially in printouts.

But to allow comparisons between the 6pw, 8pw, 10pw and 11.5pw, as the reviewer suggested, we had already included all four timepoints in the interactive viewer (<https://tissuumaps.dckube.scilifelab.se/web/private/HDCA/index.html>).

Analyses of individual donor SCLC scRNA-Seq datasets²⁰, derived from primary lung tumors.

Figure R1 2

HTA8-2002

gene correlation within cancer-cell clusters

HTA8-2003

HTA8-2004

HTA8-2005

HTA8-2007

Figure R1 3

HTA8-2008

gene correlation within cancer-cell clusters

HTA8-2009

HTA8-2010

HTA8-2011

HTA8-2012

Figure R1 4

HTA8_2013

gene correlation within cancer-cell clusters

HTA8_2014

HTA8_2017

HTA8_2018

HTA8_2019

References

1. Nishimura, G. *et al.* DeltaEF1 mediates TGF-beta signaling in vascular smooth muscle cell differentiation. *Dev Cell* **11**, 93-104 (2006).
2. Kumar, M.E. *et al.* Mesenchymal cells. Defining a mesenchymal progenitor niche at single-cell resolution. *Science* **346**, 1258810 (2014).
3. Browaeys, R., Saelens, W. & Saeys, Y. NicheNet: modeling intercellular communication by linking ligands to target genes. *Nat Methods* **17**, 159-162 (2020).
4. Woodhoo, A. *et al.* Notch controls embryonic Schwann cell differentiation, postnatal myelination and adult plasticity. *Nat Neurosci* **12**, 839-847 (2009).
5. Burns, A.J., Thapar, N. & Barlow, A.J. Development of the neural crest-derived intrinsic innervation of the human lung. *Am J Respir Cell Mol Biol* **38**, 269-275 (2008).
6. Freem, L.J. *et al.* The intrinsic innervation of the lung is derived from neural crest cells as shown by optical projection tomography in Wnt1-Cre;YFP reporter mice. *J Anat* **217**, 651-664 (2010).
7. Yang, S. *et al.* Cxcl12/Cxcr4 signaling controls the migration and process orientation of A9-A10 dopaminergic neurons. *Development* **140**, 4554-4564 (2013).
8. Song, M.R. *et al.* T-Box transcription factor Tbx20 regulates a genetic program for cranial motor neuron cell body migration. *Development* **133**, 4945-4955 (2006).
9. Sergaki, M.C. & Ibanez, C.F. GFRalpha1 Regulates Purkinje Cell Migration by Counteracting NCAM Function. *Cell Rep* **18**, 367-379 (2017).
10. Kameneva, P. *et al.* Single-cell transcriptomics of human embryos identifies multiple sympathoblast lineages with potential implications for neuroblastoma origin. *Nat Genet* **53**, 694-706 (2021).
11. Dyachuk, V. *et al.* Neurodevelopment. Parasympathetic neurons originate from nerve-associated peripheral glial progenitors. *Science* **345**, 82-87 (2014).
12. Espinosa-Medina, I. *et al.* Neurodevelopment. Parasympathetic ganglia derive from Schwann cell precursors. *Science* **345**, 87-90 (2014).
13. Adams, T.S. *et al.* Single-cell RNA-seq reveals ectopic and aberrant lung-resident cell populations in idiopathic pulmonary fibrosis. *Sci Adv* **6**, eaba1983 (2020).
14. Kathiriya, J.J. *et al.* Human alveolar type 2 epithelium transdifferentiates into metaplastic KRT5(+) basal cells. *Nat Cell Biol* **24**, 10-23 (2022).
15. Strunz, M. *et al.* Alveolar regeneration through a Krt8+ transitional stem cell state that persists in human lung fibrosis. *Nat Commun* **11**, 3559 (2020).
16. Lagares, D., Busnadiego, O., Garcia-Fernandez, R.A., Lamas, S. & Rodriguez-Pascual, F. Adenoviral gene transfer of endothelin-1 in the lung induces pulmonary fibrosis through the activation of focal adhesion kinase. *Am J Respir Cell Mol Biol* **47**, 834-842 (2012).
17. Arora, R., Metzger, R.J. & Papaioannou, V.E. Multiple roles and interactions of Tbx4 and Tbx5 in development of the respiratory system. *PLoS Genet* **8**, e1002866 (2012).

18. Ireland, A.S. *et al.* MYC Drives Temporal Evolution of Small Cell Lung Cancer Subtypes by Reprogramming Neuroendocrine Fate. *Cancer Cell* **38**, 60-78 e12 (2020).
19. Stewart, C.A. *et al.* Single-cell analyses reveal increased intratumoral heterogeneity after the onset of therapy resistance in small-cell lung cancer. *Nat Cancer* **1**, 423-436 (2020).
20. Chan, J.M. *et al.* Signatures of plasticity, metastasis, and immunosuppression in an atlas of human small cell lung cancer. *Cancer Cell* **39**, 1479-1496 e1418 (2021).
21. Liu, Z. *et al.* The intracellular domains of Notch1 and Notch2 are functionally equivalent during development and carcinogenesis. *Development* **142**, 2452-2463 (2015).
22. Liu, Z. *et al.* The extracellular domain of Notch2 increases its cell-surface abundance and ligand responsiveness during kidney development. *Dev Cell* **25**, 585-598 (2013).
23. Singh, R., Letai, A. & Sarosiek, K. Regulation of apoptosis in health and disease: the balancing act of BCL-2 family proteins. *Nat Rev Mol Cell Biol* **20**, 175-193 (2019).
24. Mattes, K., Berger, G., Geugien, M., Vellenga, E. & Schepers, H. CITED2 affects leukemic cell survival by interfering with p53 activation. *Cell Death Dis* **8**, e3132 (2017).
25. Pastor, T.P., Peixoto, B.C. & Viola, J.P.B. The Transcriptional Co-factor IRF2BP2: A New Player in Tumor Development and Microenvironment. *Front Cell Dev Biol* **9**, 655307 (2021).
26. Lehman, J.M. *et al.* Somatostatin receptor 2 signaling promotes growth and tumor survival in small-cell lung cancer. *Int J Cancer* **144**, 1104-1114 (2019).
27. Cao, J. *et al.* A human cell atlas of fetal gene expression. *Science* **370** (2020).
28. Gibson, G.J., Loddenkemper, R., Lundback, B. & Sibille, Y. Respiratory health and disease in Europe: the new European Lung White Book. *Eur Respir J* **42**, 559-563 (2013).
29. Rajewsky, N. *et al.* Publisher Correction: LifeTime and improving European healthcare through cell-based interceptive medicine. *Nature* **592**, E8 (2021).
30. Yang, Y. *et al.* Spatial-Temporal Lineage Restrictions of Embryonic p63(+) Progenitors Establish Distinct Stem Cell Pools in Adult Airways. *Dev Cell* **44**, 752-761 e754 (2018).
31. Miller, A.J. *et al.* In Vitro and In Vivo Development of the Human Airway at Single-Cell Resolution. *Dev Cell* **53**, 117-128 e116 (2020).
32. Csardi, G. & Nepusz, T. The igraph software package for complex network research. *InterJournal, complex systems* **1695**, 1-9 (2006).
33. Travaglini, K.J. *et al.* A molecular cell atlas of the human lung from single-cell RNA sequencing. *Nature* **587**, 619-625 (2020).
34. McInnes, L., Healy, J. & Melville, J. arXiv:1802.03426 (2018).
35. Wolf, F.A. *et al.* PAGA: graph abstraction reconciles clustering with trajectory inference through a topology preserving map of single cells. *Genome Biol* **20**, 59 (2019).
36. Bergen, V., Lange, M., Peidli, S., Wolf, F.A. & Theis, F.J. Generalizing RNA velocity to transient cell states through dynamical modeling. *Nat Biotechnol* **38**, 1408-1414 (2020).
37. Street, K. *et al.* Slingshot: cell lineage and pseudotime inference for single-cell transcriptomics. *BMC Genomics* **19**, 477 (2018).
38. Nikolic, M.Z. *et al.* Human embryonic lung epithelial tips are multipotent progenitors that can be expanded in vitro as long-term self-renewing organoids. *Elife* **6** (2017).
39. Danopoulos, S. *et al.* Discordant roles for FGF ligands in lung branching morphogenesis between human and mouse. *J Pathol* **247**, 254-265 (2019).

40. Bellusci, S., Grindley, J., Emoto, H., Itoh, N. & Hogan, B.L. Fibroblast growth factor 10 (FGF10) and branching morphogenesis in the embryonic mouse lung. *Development* **124**, 4867-4878 (1997).
41. Madisson, E. *et al.* A spatial multi-omics atlas of the human lung reveals a novel immune cell survival niche. *bioRxiv*, 2021.2011.2026.470108 (2021).
42. Zhou, Y. *et al.* Extracellular matrix in lung development, homeostasis and disease. *Matrix Biol* **73**, 77-104 (2018).
43. Yang, J., Velikoff, M., Canalis, E., Horowitz, J.C. & Kim, K.K. Activated alveolar epithelial cells initiate fibrosis through autocrine and paracrine secretion of connective tissue growth factor. *Am J Physiol Lung Cell Mol Physiol* **306**, L786-796 (2014).
44. Richeldi, L. *et al.* Pamrevlumab, an anti-connective tissue growth factor therapy, for idiopathic pulmonary fibrosis (PRAISE): a phase 2, randomised, double-blind, placebo-controlled trial. *Lancet Respir Med* **8**, 25-33 (2020).
45. Baguma-Nibasheka, M. & Kablar, B. Pulmonary hypoplasia in the connective tissue growth factor (Ctgf) null mouse. *Dev Dyn* **237**, 485-493 (2008).
46. Leask, A. & Abraham, D.J. All in the CCN family: essential matricellular signaling modulators emerge from the bunker. *J Cell Sci* **119**, 4803-4810 (2006).
47. Ugucioni, M. *et al.* Endothelin-1 in idiopathic pulmonary fibrosis. *J Clin Pathol* **48**, 330-334 (1995).
48. Xu, S.W. *et al.* Endothelin-1 induces expression of matrix-associated genes in lung fibroblasts through MEK/ERK. *J Biol Chem* **279**, 23098-23103 (2004).
49. Godoy-Guzman, C., San Martin, S. & Pereda, J. Proteoglycan and collagen expression during human air conducting system development. *Eur J Histochem* **56**, e29 (2012).
50. Diederichs, S. *et al.* Regulation of WNT5A and WNT11 during MSC in vitro chondrogenesis: WNT inhibition lowers BMP and hedgehog activity, and reduces hypertrophy. *Cell Mol Life Sci* **76**, 3875-3889 (2019).
51. Wang, C. *et al.* Differentiation of adipose-derived stem cells into contractile smooth muscle cells induced by transforming growth factor-beta1 and bone morphogenetic protein-4. *Tissue Eng Part A* **16**, 1201-1213 (2010).
52. Heanue, T.A. *et al.* Synergistic regulation of vertebrate muscle development by Dach2, Eya2, and Six1, homologs of genes required for Drosophila eye formation. *Genes Dev* **13**, 3231-3243 (1999).
53. Aros, C.J., Pantoja, C.J. & Gomperts, B.N. Wnt signaling in lung development, regeneration, and disease progression. *Commun Biol* **4**, 601 (2021).
54. Cohen, E.D. *et al.* Wnt signaling regulates smooth muscle precursor development in the mouse lung via a tenascin C/PDGFR pathway. *J Clin Invest* **119**, 2538-2549 (2009).
55. Lolis, A.A. *et al.* Myogenin recruits the histone chaperone facilitates chromatin transcription (FACT) to promote nucleosome disassembly at muscle-specific genes. *J Biol Chem* **288**, 7676-7687 (2013).
56. Liu, Y. *et al.* Nur77 suppresses pulmonary artery smooth muscle cell proliferation through inhibition of the STAT3/Pim-1/NFAT pathway. *Am J Respir Cell Mol Biol* **50**, 379-388 (2014).
57. Cho, K.H. *et al.* Ganglia in the Human Fetal Lung. *Anat Rec (Hoboken)* **302**, 2233-2244 (2019).

58. Dalpiaz, G., Cancellieri, A. & SpringerLink (Online service) XVII, 295 p. 569 illus., 522 illus. in color (2017).

Decision Letter, first revision:

Date: 12th August 22 05:01:46
Last Sent: 12th August 22 05:01:46
Triggered By: Stylianos Lefkopoulos
From: stylianos.lefkopoulos@springernature.com
To: christos.samakovlis@su.se
CC: ncb@springernature.com
Subject: Your manuscript, NCB-RS48074A
Message: Our ref: NCB-RS48074A

12th August 2022

Dear Christos,

Thank you for submitting your revised manuscript "Developmental origins of cell heterogeneity in the human lung" (NCB-RS48074A). It has now been seen by the original referees and their comments are below. The reviewers find that the paper has improved in revision, and therefore we'll be happy in principle to publish it in Nature Cell Biology, pending minor revisions to comply with our editorial and formatting guidelines.

If the current version of your manuscript is in a PDF format, please email us a copy of the file in an editable format (Microsoft Word or LaTeX) as soon as possible -- we can not proceed with PDFs at this stage.

We are now performing detailed checks on your paper and will send you a checklist detailing our editorial and formatting requirements in about a week after you provide us with the editable file. Please do not upload the final materials and make any revisions until you receive this additional information from us.

Thank you again for your interest in Nature Cell Biology. Please do not hesitate to contact me if you have any questions.

Best wishes,

Stelios

Stylianos Lefkopoulos, PhD
He/him/his

Associate Editor
Nature Cell Biology
Springer Nature
Heidelberger Platz 3, 14197 Berlin, Germany

E-mail: stylianos.lefkopoulos@springernature.com
Twitter: @s_lefkopoulos

Reviewer #1 (Remarks to the Author):

In this revision, the authors have addressed many of the original concerns with adequate changes/modification. Moreover, some claims about functionality and some of the comparisons to the adult lung and disease states were tempered. The manuscript has improved.

Reviewer #2 (Remarks to the Author):

In this revised manuscript, the authors have addressed the major concerns of reviewers. It is suitable for publication in Nature Cell Biology.

Final Decision Letter:

Date: 23rd November 22 12:33:21

Last Sent: 23rd November 22 12:33:21

Triggered By: Stylianos Lefkopoulos

From: stylianos.lefkopoulos@springernature.com

To: christos.samakovlis@su.se

CC: ncb@springernature.com

BCC: rjsproduction@springernature.com; rjsart@springernature.com

Subject: Decision on Nature Cell Biology submission NCB-RS48074B

Message: Dear Christos,

I am pleased to inform you that your manuscript, "A topographic atlas defines developmental origins of cell heterogeneity in the human embryonic lung", has now been accepted for publication in Nature Cell Biology. Congratulations to you and the whole team!

Over the next few weeks, your paper will be copyedited to ensure that it conforms to

Nature Cell Biology style. Once your paper is typeset, you will receive an email with a link to choose the appropriate publishing options for your paper and our Author Services team will be in touch regarding any additional information that may be required.

Please note that *Nature Cell Biology* is a Transformative Journal (TJ). Authors may publish their research with us through the traditional subscription access route or make their paper immediately open access through payment of an article-processing charge (APC). Authors will not be required to make a final decision about access to their article until it has been accepted. Find out more about Transformative Journals

If you have not already done so, we strongly recommend that you upload the step-by-step protocols used in this manuscript to the Protocol Exchange (www.nature.com/protocolexchange), an open online resource established by Nature Protocols that allows researchers to share their detailed experimental know-how. All uploaded protocols are made freely available, assigned DOIs for ease of citation and are fully searchable through nature.com. Protocols and Nature Portfolio journal papers in which they are used can be linked to one another, and this link is clearly and prominently visible in the online versions of both papers. Authors who performed the specific experiments can act as primary authors for the Protocol as they will be best placed to share the methodology details, but the Corresponding Author of the present research paper should be included as one of the authors. By uploading your Protocols to Protocol Exchange, you are enabling researchers to more readily reproduce or adapt the methodology you use, as well as increasing the visibility of your protocols and papers. You can also establish a dedicated page to collect your lab Protocols. Further information can be found at www.nature.com/protocolexchange/about

With kind regards,
Stelios

Stylios Lefkopoulos, PhD
He/him/his
Associate Editor
Nature Cell Biology
Springer Nature
Heidelberger Platz 3, 14197 Berlin, Germany

E-mail: stylios.lefkopoulos@springernature.com
Twitter: @s_lefkopoulos
